# A mountain ridge model for quantifying oblique mountain wave propagation and distribution

Sebastian Rhode[1], Peter Preusse[1], Manfred Ern[1], Jörn Ungermann[1], Lukas Krasauskas[1], Julio Bacmeister[2], and Martin Riese[1]

[1]Institute of Energy and Climate Research, Stratosphere (IEK-7), Forschungszentrum Jülich, Jülich, Germany
[2]National Center for Atmospheric Research (NCAR), Boulder, CO, USA

**Correspondence:** Sebastian Rhode (s.rhode@fz-juelich.de)

**Abstract.**

Following the current understanding of gravity waves (GWs) and especially mountain waves (MWs), they have a high potential for horizontal propagation from their source. This horizontal propagation and therefore the transport of energy is usually not well represented in MW parameterizations of numerical weather prediction and general circulation models. In this study, we present a mountain wave model (MWM) for the quantification of horizontal propagation of orographic gravity waves. This model determines MW source location from topography data and estimates MW parameters from a fit of idealized Gaussian-shaped mountains to the elevation. Propagation and refraction of these MWs in the atmosphere are modeled using the ray tracer GROGRAT. Ray-tracing each MW individually allows for an estimation of momentum transport due to both vertical and horizontal propagation. The MWM is a capable tool for the analysis of MW propagation and global MW activity and can support the understanding of observations and improvement of MW parameterizations in GCMs. This study presents the model itself and gives validations of MW-induced temperature perturbations to ECMWF IFS numerical weather prediction data and estimations of GW momentum flux (GWMF) compared to HIRDLS satellite observations. The MWM is capable of reproducing the general features and amplitudes of both of these data sets and, in addition, is used to explain some observational features by investigating MW parameters along their trajectories.

## 1 Introduction

Gravity waves (GWs) are atmospheric waves for which gravity or buoyancy acts as the restoring force (Fritts, 1984). They are an important dynamical process and interact with the large-scale flow (e.g. Holton, 1983; Andrews et al., 1987) and contribute to the generation of clouds (e.g. Thayer et al., 2003; Saha et al., 2020). Since they propagate through the atmosphere, both vertically and horizontally, they transport momentum from the lower atmosphere, or even the ground, to higher levels such as the stratosphere, mesosphere, and lower thermosphere. This relocation of momentum is one of the drivers of the Brewer-Dobson-Circulation (BDC) and the main driver of the upper, mesospheric branch of the residual circulation (e.g. McIntyre, 1998; Fritts and Alexander, 2003). Various studies also argue for a significant role of gravity waves in the occurrence of Sudden

Stratospheric Warming (SSW) events (e.g. Whiteway et al., 1997; Kidston et al., 2015) and even their shape, i.e. whether the polar vortex splits or displaces (Albers and Birner, 2014; Ern et al., 2016; Song et al., 2020).

In addition, the effects of GWs are a major uncertainty in climate projections and numerical weather forecasts (Shepherd, 2014). On the one hand, GWs have a significant impact on the dynamics of the atmosphere and even larger-scale climate phenomena, on the other hand, they have to be parameterized within general circulation models (GCMs). While larger-scale GWs are resolved by the models, small- and medium-scale GWs caused by the sub-grid-scale orography and convection are typically approximated by a parameterization scheme (e.g. Lott and Miller, 1997; Kim et al., 2003; Xie et al., 2020). Following
(Skamarock, 2004), the shortest resolved scales are about 4 times the grid resolution, which translates to about $500 - 1000\,\mathrm{km}$ for long-term GCM simulations. Shorter GWs are considered small-scale in this study. One particular shortcoming of these parameterizations employed in GCMs is the commonly used column-wise calculation approach, which does not allow for the GW's momentum to be transported horizontally, whereby the corresponding GW drag will be exerted above the mountains. However, a high potential of horizontal GW propagation has been found both in observations and model studies (e.g. Preusse
et al., 2002; Sato et al., 2012; Krisch et al., 2017; Ehard et al., 2017; Strube et al., 2021).

Although GWs can be excited by various processes, e.g. convection, jet imbalances, and even volcanic eruptions (Wright et al., 2022; Ern et al., 2022), one of the major and most predictable sources of gravity waves is wind flow over orography, by which air parcels are displaced vertically. These stationary (with respect to the ground) mountain waves (MWs) propagate through the atmosphere, both vertically and horizontally. In the middle atmosphere, they can be measured from satellites (e.g.
Eckermann and Preusse, 1999; Preusse et al., 2002; Jiang et al., 2004; Eckermann and Wu, 2012). The strength of the excited MWs depends on the height and shape of the orography, the velocity of the low-level flow, and the propagation conditions due to the winds above. At mid-latitudes, westerly winds prevail in the troposphere and north-south oriented ridges are expected to be particularly effective MW sources. Accordingly, the Rocky Mountains and the Andes are regions in which particularly high MW activity is expected. Indeed, for both mountain chains severe clear air turbulence due to mountain waves has caused
major aviation incidents (e.g. Smithsonian Magazine, 2005; Boldmethod, 2016; Aviación Global, 2019).

Strong GW activity in the troposphere, however, does not always directly translate to high GW activity in the stratosphere, although these MWs should propagate upwards. Climatologies of GWs for the mid-stratosphere show only moderate GW activity over both the Rocky Mountains and the Andes north of $40°$S (e.g. Geller et al., 2013; Ern et al., 2018; Hindley et al., 2020). Likewise, the highest mountains on Earth, the Himalaya, have only moderate impact on middle atmosphere GW
distributions. In order to understand this comparatively low stratospheric GW activity, observations might be aided by model studies focusing on specific types of GWs. A model describing orographic GW propagation from source to breaking, for example, could shed light on the orographic part of the measured GW spectrum and help disentangle it from other sources. The combination of model and observation data enriches the analysis by providing more data to base the conclusions on but does also provide an opportunity to probe the underlying theory in a real-world application.

In this study, we present a mountain wave model (MWM) capable of estimating the sources of orographic gravity waves similar to the approach by Bacmeister (1993) and Bacmeister et al. (1994). The propagation of the so-determined MWs throughout the atmosphere is modeled using the ray-tracer GROGRAT (Marks and Eckermann, 1995). Refraction of GWs due to horizon-

tal gradients and time dependence of the background fields is considered within the ray-tracer. Results include 4-dimensional momentum flux distributions, drag exerted on the background winds, and estimations of the residual temperature perturbations caused by the MWs. Compared to previous studies, we aim for higher accuracy in terms of mountain source detection by a fit of an idealized mountain shape at arbitrary angles. Additionally, we are considering flow blocking and surface friction effects in the low-level winds to improve modeled MW amplitudes and field characteristics. We are validating our model against satellite and model data to a higher degree of detail and emphasis on the MWM than in previous studies (e.g. Jiang et al., 2002, 2004). For this, we consider data at altitudes of 15-25 km, which is low enough for a comparison of the effect of primary MWs before and after wind filtering in the atmosphere. Satellite data at these low altitudes have not been used for such a comparison before.

The described MWM is used to explain GW features in satellite data by investigating their wave characteristics from source to observation altitude. MW propagation patterns throughout the year are predicted and agree with previous studies of oblique MW spread. Therefore this model might be used for identifying MW propagation patterns in further studies for improving MW parameterizations in GCMs by approximating their horizontal spread.

This article is organized as follows: First, Sect. 2 introduces the data used in this study. A general description of the mountain wave model is given in Sect. 3, which describes the detection of mountain wave sources, estimation of launch parameters, and modeling of the propagation. In addition, the post-processing of MWM data resulting in reconstructions of residual temperature and gravity wave momentum flux (GWMF) distributions is discussed. Following this, a brief validation of the model is given in Sect. 4 including an investigation of the detected scales and a comparison to ECMWF operational analysis temperature data. In Sect. 5 the model's capability to predict horizontal GW propagation is shown by comparison with HIRDLS satellite data. Calculated GW parameters along the ray paths and their change with altitude and critical level filtering are considered as possible causes of some observational features. In addition, predictions of horizontal GWMF patterns throughout the year are shown, which give first insight into the universality of horizontal propagation. Finally, the results are summarized in Sect. 6 and concluding remarks are given.

## 2   Data

This study uses the ETOPO1 topography data (Amante and Eakins, 2009) for the ridge finding as well as atmospheric background winds and temperature from ERA5 reanalysis (Hersbach et al., 2020) for MW propagation modeling via ray-tracing. HIRDLS satellite observations of gravity wave momentum fluxes (GWMFs) are used for validation and comparison. The used data sets are presented in the following sections.

### 2.1   Topography data

The underlying topographic elevation data used in this study is taken from the ETOPO1 GLOBAL RELIEF MODEL (Amante and Eakins, 2009; Center, 2009) data set. This data is available in two versions: one describes the bedrock elevation only, and the other considers also the ice surface, i.e. glaciers and ice sheets. Since we are interested in the elevation encountered by the low-level flow, we use the data including the ice surface. This data set models the earth's surface, including ocean bathymetry,

on a resolution of 1 arc-minute, or about 1.85 km at the equator, and is combined from multiple global and regional data sets. However, we set all regions below sea level, which are not relevant for our analysis, to zero to approximate an ocean surface.

## 2.2 Atmospheric backgrounds

The atmospheric background wind and temperature data used for ray-tracing of MWs are generated from ECMWF ERA5 reanalysis data (Hersbach et al., 2020; , C3S), sampled on a $0.3° \times 0.3°$ grid, or about $33\,km \times 33\,km$ at the equator. For our ray-tracing experiments, we want the background to contain all global and synoptic scale features but no GWs, which are potentially resolved by the numerical weather prediction (NWP) model. Following Strube et al. (2020), a scale separation approach is therefore applied to the data set using a zonal Fourier transform with a cutoff zonal wavenumber of 18, which correspond to wavelengths of about 2200 km at the equator. In the meridional direction, the data is smoothed using a 3rd order Savitzky-Golay filter of 31 subsequent data points ($\sim 9°$ total window width). For the use in GROGRAT, the smoothed background is sampled onto a grid of $2°$ latitude and $2.5°$ longitude, or about 220 km and 280 km at the equator. In the vertical the data is interpolated to equidistant altitudes of 0.5 km spacing. For global ray-tracing experiments, 6 hourly snapshots, and for the specific case study of the Southern Andes in Sec. 4.2, hourly snapshots are used. To ensure smooth transitions in between, GROGRAT uses a 4-dimensional spline interpolation.

In addition to the ERA5 reanalysis data, single snapshots of operational forecast data of the ECMWF integrated forecast system (IFS) is used for validation of MW temperature perturbations in Sec. 4.2. This data set is given on a higher resolution of $0.1°$, or about 10 km, in the horizontal and allows therefore resolution of GWs down to about 40 km (Skamarock, 2004). The scale separation is performed analogously to the ERA5 data described above (the number of points in the meridional filter has been increased to 91 to achieve the same filter width of $\sim 9°$).

## 2.3 HIRDLS satellite data

The horizontal GWMF distributions generated by our MWM for the year 2006 are validated and compared in Sect. 5.1 to satellite data from the HIgh Resolution Dynamics Limb Sounder (HIRDLS, Gille et al. (2003)) instrument. The horizontal sampling of these measurements is about $80 - 100\,km$ along track. Here we use a data set specially prepared for the UTLS (Upper Troposphere/Lower Stratosphere) region, spanning the altitudes from 14 km to 25 km. As discussed by Strube et al. (2020), below 20 km altitude zonal wavenumbers of 10 and higher need to be taken into account to describe the background. These cannot be self-consistently estimated from a single-track low Earth orbit satellite (Salby and Callaghan, 1997). In order to isolate the small-scale gravity wave (GW) contributions, ERA5 background data with an altitude-dependent zonal wavenumber cutoff has been subtracted from the retrieved HIRDLS temperature measurements. Below 10 km altitude a cutoff at zonal wavenumber 20 (about 2000 km at the equator), above 20 km a cutoff at 6 (about 6700 km at the equator) has been used. In between, the cutoff decreases linearly from 20 to 6. In addition, the background removal as described in Ern et al. (2018) that is based on HIRDLS data only has been applied in a second step to account for imperfections of ERA5. The resulting vertical profiles of temperature residuals have been used for the calculation of GWMF as described in Ern et al. (2018). To analyze the lower stratosphere (20 km and below) and simultaneously avoid the influence of the tropopause, the vertical window of the

MEM/HA method (Preusse et al., 2002) was reduced from 10 km, which is usually used for stratospheric altitudes (e.g. Ern et al., 2004), to 5 km. Such a reduced window size is adequate, since in the lower stratosphere the average vertical wavelengths are much lower than in the mid stratosphere and mesosphere (e.g. Chane-Ming et al., 2000; Yan et al., 2010; Ern et al., 2018). In addition, the HIRDLS data set has been high-pass filtered in terms of vertical wavenumbers using a 5th-order Butterworth filter with vertical wavelength cutoff at 12 km, similar to Ehard et al. (2015).

Note that the scale-separation methodology of the ERA5 data used for background removal differs from the generation of the ray-tracing backgrounds. Since we are interested in the GW content of the measurements, we need to carefully remove the larger-scale dynamics, such as Rossby waves, from the background field. In the lower stratosphere, these can reach zonal wavenumbers as high as 6, but considerably higher in the troposphere, which is why the filter is designed with a linear decrease of cutoff wavenumber with altitude. The ray-tracing simulations are not as sensitive to small remnants of smaller-scale dynamics, and, therefore, the scale separation described in Strube et al. (2020) is used there.

For this study, GWMF is binned within rectangular overlapping bins of $15°$ in longitude and $5°$ in latitude sampled every $5°$ in longitude and $2.5°$ in latitude from the original profiles given along the satellite orbits. The use of overlapping bins introduces spatially dependent auto-correlations to some extent, which leads to smearing out of the global distribution. The advantage is, however, an increase in statistics for each bin. The vertical resolution of the data set is 1 km, which corresponds to the vertical resolution of HIRDLS. Note that due to the vertical window of 5 km used in the MEM/HA method the given levels are representative for $\pm 2.5$ km around the corresponding altitude.

## 3   Mountain Wave Model

The Mountain Wave Model (MWM) presented in this paper consists of three independent parts. The first one is the identification of mountain ridges from elevation data. The use of ridges follows the approach presented by Bacmeister (1993) and Bacmeister et al. (1994), but different methods are used for the parameter determination. The second part of the model consists of the translation of the determined ridge parameters into MW launch parameters. In the third part, the so-initialized gravity waves are propagated through the background atmosphere using the ray-tracer GROGRAT (Marks and Eckermann, 1995).

### 3.1   Ridge Identification

The first specific task of our MWM is the estimation of MW parameters, such as horizontal wavelength, amplitude, orientation, and location from a fit of an idealized mountain ridge to the topography. This allows launching a monochromatic GW exerted by the detected mountain ridge later on. An overview of the algorithm and examples for the intermediate steps of the ridge identification as applied to a part of South America is given in Fig. 1.

First, the elevation data is divided into overlapping slices of $10°$ width in latitude spanning the full globe in longitude. These slices are generated every $7.5°$ in latitude, leading to a $2.5°$ overlap. The longitudes of the topography slice are resampled onto a 1' latitude, or about 1.85 km, equivalent grid (at the meridional center), such that the resulting grid is equidistant in the center. The grid distance is, however, still increasing equator-ward (decreasing pole-ward). This resampling mitigates possible errors

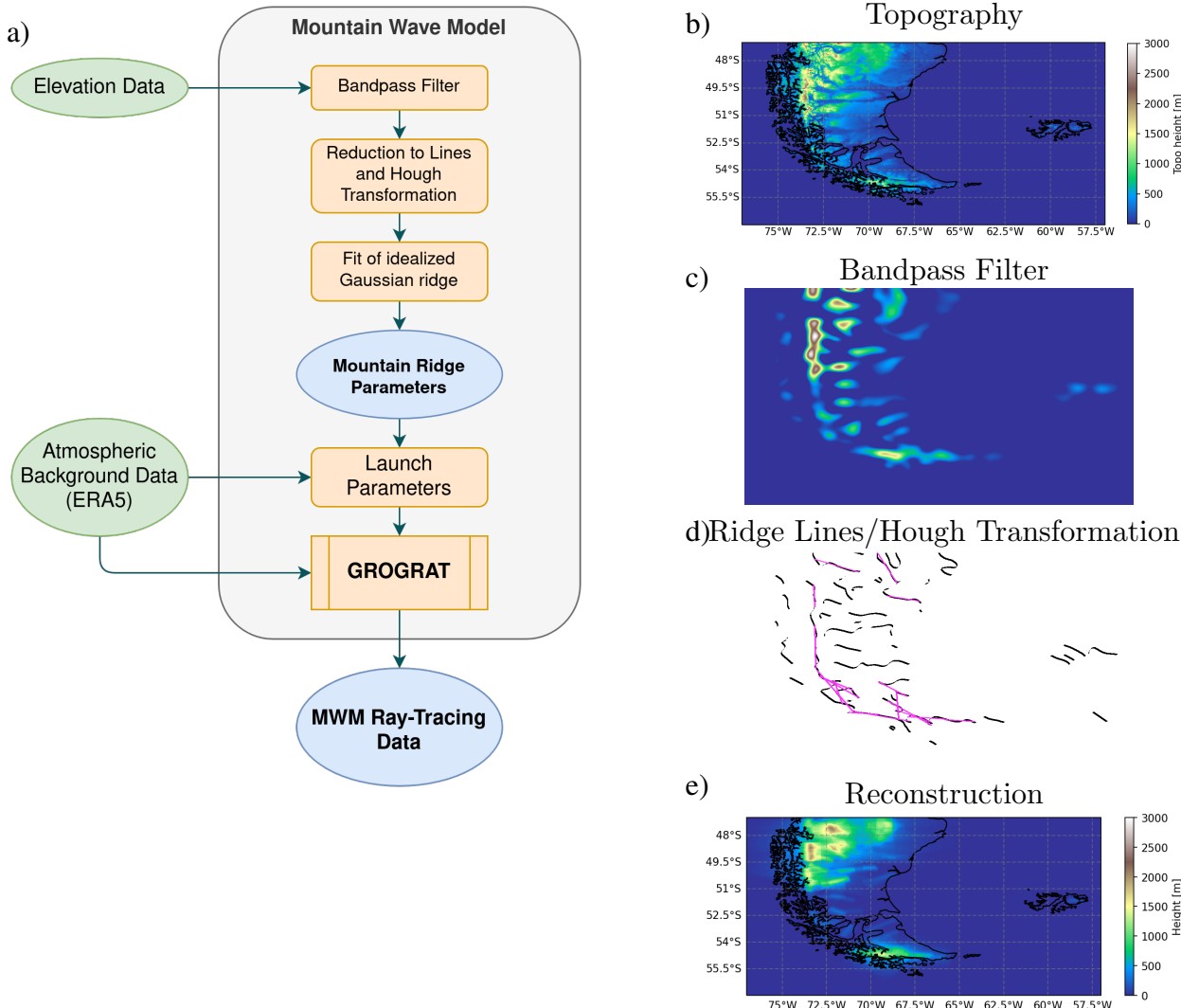

**Figure 1.** Flowchart of the algorithm and the intermediate steps exemplified for South America. Panel a) – Flowchart describing the main steps of the Mountain Wave Model. Input data is shown in green, internal processing steps in yellow and MWM output in blue. Panel b) - e) – example of the intermediate steps and output of the source detection algorithm for the scale interval [80 km, 150 km]. b) – input topography data, c) – bandpass-filtered topography on an equidistant grid, d) – reduction to the ridge lines and Hough transformation for a single direction and scale. Black lines represent the detected ridge lines of the bandpass filtered field, magenta are the found straight line segments from the Hough transformation. The considered direction here is South-West to North-East. And e) – reconstruction of all fitted idealized mountain ridges.

in the scale separation and fitting of ridge widths and lengths due to differently scaled dimensions at high latitudes. Technically the division into slices limits the maximum ridge length to $10°$ in latitude, which in practice, however, never occurred. In order to identify mountain ridges of different scales, a Gaussian band-pass filter is applied as a second step. This is calculated as the

| Interval No. | lower boundary [km] | upper boundary [km] |
|:---:|:---:|:---:|
| 1 | 80 | 150 |
| 2 | 150 | 250 |
| 3 | 250 | 400 |
| 4 | 400 | 600 |
| 5 | 600 | 850 |
| 6 | 850 | 1100 |
| 7 | 1100 | 1500 |
| 8 | 1500 | 2000 |

**Table 1.** Band-pass intervals used for detecting mountain ridges in the topography for this study.

difference between the elevation data convoluted with a 2-dimensional Gaussian function of different scales which are given as the limits of the considered scale interval. The filtering also removes large-scale plateaus, which are not of interest in this iteration of our model. In this study, we are using the band-pass intervals given in Table 1. The resulting GW spectrum consists of both, inertia-gravity waves and non-hydrostatic GWs, with $\hat{\omega} \gg f$. In the following, both are referred to as GWs.

For each band-pass filtered slice of topography, the main ridge lines (or arêtes) are identified by detecting a corresponding sign change in the gradient, which itself is calculated by convolution with an optimized $3 \times 3$ Sobel operator (the Scharr operator, Jähne et al., 1999). To account for ridges of different orientations, this identification (and gradient calculation) is performed for 4 different directions separately (zonal, meridional, and both diagonals). The result of this step is a field of line structures, representing the ridge lines (c.f. Fig. 1 d)). We determine the characteristics of these lines using a probabilistic Hough transformation (e.g. Duda and Hart, 1972; Kang et al., 1991), which yields the locations, lengths, and orientations of lines contributing to the total structure. The advantage of this approach is the sampling of arbitrary lengths and orientations by the same method. For completeness, the Hough transform as well as its probabilistic variants and their sensitivity to parameters are briefly described in App. B.

After the line-like features in the bandpass-filtered topography have been identified, the width and height of each possible mountain ridge at the line's location and orientation is estimated by a fit of an idealized Gaussian-shaped mountain. Note that the location is taken from the lines found by the Hough transformation while the fit itself is performed on the bandpass-filtered topography. The cross-section of the idealized Gaussian ridges is given by:

$$f(x; h, a) = h \exp\left(-\frac{x^2}{2a^2}\right) \tag{1}$$

with $x$ being the distance perpendicular to the ridge line and $h$ and $a$ parameters for the mountain's height and width, respectively. The fit minimizes the mean absolute error and is performed with a 2D ridge, constructed by extending this cross-section for the length of the line, which was determined by the Hough transformation.

As a result of the combined ridge-finding algorithm we obtain a set of ridges with the following parameters: ridge length $L$ and local Cartesian ridge coordinates $X$ and $Y$ (representing ridge location in zonal and meridional directions, respectively), angle between $X$ axis and the ridge $\theta$, best-fit width $a$, and best-fit ridge height $h$.

The previous step results in a collection of mountain ridges. We assume that each of these ridges can excite a MW. In order to propagate the wave with GROGRAT we need wave vector and wave amplitude. The displacement amplitude is calculated from the best-fitting idealized mountain height $h$ as $\zeta = \frac{h}{\sqrt{2\pi}}$. The factor $\frac{1}{\sqrt{2\pi}}$ stems from linear modeling of a two-dimensional ridge with Gaussian shape (e.g. Nappo, 2012). The horizontal wave vector is chosen perpendicular to the ridge orientation and the horizontal wavelength is set to $\lambda_{\mathrm{hor}} = 2\pi a$, where $a$ is the width of the best-fitting idealized Gaussian ridge. This is the wavelength of maximum response for the given mountain shape (c.f. Nappo, 2012), i.e. the strongest mode of all possibly excited GWs, and gives an approximation of the full spectrum with a single monochromatic GW. Lastly, the vertical wavelength can be found from the dispersion relation and background data (see e.g. Fritts and Alexander (2003)), which is taken care of by the ray-tracer. In the specific simulations, a single MW is launched at the center of each mountain ridge at every simulated time step (i.e. every 6 hours in the global and every hour in the Southern Andes case) with launch parameters derived from the corresponding atmospheric conditions. An overview of all parameters estimated for each MW by the MWM is given in Table 2.

Using overlapping latitude slices, we sample the same topography more than once. In addition, it is not guaranteed that a single mountain ridge results in only one straight line in the Hough transformation. Thus, we expect redundancies in the ridge collection at this stage. In order to avoid double counting, we test each ridge against all other ridges by the following criteria:

- horizontal wavelength differs by less than 20%

- orientation differs by less than 22.5°

- distance parallel to the ridge is no more than $0.5L$

- distance perpendicular to the ridge is no more than $0.5\lambda_{\mathrm{hor}}$

If for any ridge pair, all of these criteria are fulfilled, they are assumed to describe the same underlying ridges and only the one with the better fit to the bandpass-filtered topography is used. After this filtering, we end up with the final mountain ridge database.

## 3.2 Ray Tracer

Based on the mountain ridge database established in Sect. 3.1, this section describes the derivation of MW launch parameters for specific atmospheric conditions and details of the propagation calculation within the ray tracer GROGRAT (Marks and Eckermann, 1995). Here, we use a modified version of GROGRAT that accounts for the spherical geometry along the ray paths as derived by Hasha et al. (2008).

In this framework, the lower boundary condition for each individual GW is given by the location (longitude, latitude, and altitude) and launch time, the horizontal wave vector $(k, l)$, the initial amplitude, and the ground-based frequency. Since we

**Table 2.** Parameters derived within the Mountain Wave Model – either directly from topography data or from other parameters

| Parameter | Description | Estimation |
|---|---|---|
| $L$ | Ridge length | from Hough transformation |
| $X, Y$ | Lon, lat position | from Hough transformation |
| $\theta$ | Math. angle of ridges w.r.t. local x-coordinate | from Hough transformation |
| $a$ | Width of idealized ridge | from Gaussian fit |
| $h$ | Height of idealized ridge | from Gaussian fit |
| $\lambda_{\mathrm{hor}}$ | Horizontal wavelength | $2\pi a$ |
| $\zeta$ | Displacement amplitude | $\frac{h}{\sqrt{2\pi}}$ |

are considering mountain waves, the ground-based frequency of all our waves is assumed to be zero at launch, $\omega_{gb} = 0$, which in turn leads to the intrinsic frequency $\omega = -kU - lV$, where the zonal and meridional background winds, $U$ and $V$, and the horizontal components of the wave vector, $k$ and $l$, are determined by atmospheric background data and the MWM respectively. Positive/negative $\omega$ corresponds to waves propagating against/with the wind. Since MWs are only excited propagating against

the wind flow, we will assume positive $\omega$ in the following.

To account for surface friction of the low-level wind and potential blocking at low wind speeds, a reduced, effective displacement amplitude is calculated following the discussion in Barry (2008, pp. 72-82), who states that the conversion factor between kinetic and potential energy due to surface friction effects is about 0.64. In addition, the amplitude of the displacement excited by air forced vertically over the Gaussian mountain is assumed to be about half the air parcels total vertical displacement:

$$\zeta_{\mathrm{eff}} = \min\left( \zeta, \frac{0.32 U_{\mathrm{par}}}{N} \right), \tag{2}$$

where $N$ is the Brunt-Väisälä frequency and $U_{\mathrm{par}}$ is the horizontal wind velocity projected onto the wave vector. This means that the displacement amplitude is reduced in case the wind and stability do not allow for the full amplitude the mountain could excite. Using this effective displacement, the wind amplitude, $U_{\mathrm{amp}}$, can be calculated following linear theory (e.g. Fritts and Alexander, 2003) via

$$U_{\mathrm{amp}} = \frac{N}{\sqrt{1 - \left(\frac{f}{\omega}\right)^2}} \zeta_{\mathrm{eff}}. \tag{3}$$

Here, $f$ is the Coriolis parameter and $\omega$ is the intrinsic frequency of the considered GW.

The ray-tracing of the excited GWs itself is performed by (a modified version of) GROGRAT (Marks and Eckermann, 1995), which implements the ray equations derived in Lighthill (1978) including corrections for spherical geometry as derived by Hasha et al. (2008). Refraction and propagation are calculated using the equations

$$\frac{dx_i}{dt} = \frac{\partial \omega}{\partial k_i} \quad \text{and} \quad \frac{dk_i}{dt} = -\frac{\partial \omega}{\partial x_i}, \tag{4}$$

where $x_i$, $k_i$ are the $i$-th component of the spherical position- and wave vector, respectively. The derivative $\frac{d}{dt} = \frac{\partial}{\partial t} + c_{g,i}\frac{\partial}{\partial x_i}$, where summation over $i$ is implied, is the Lagrangian derivative for an observer following the GW with its group velocity and $\omega$ is the intrinsic frequency given by the dispersion relation

$$\omega^2 = \frac{N^2(k^2 + l^2) + f^2(m^2 + \frac{1}{4H^2})}{k^2 + l^2 + m^2 + \frac{1}{4H^2}}. \tag{5}$$

Here $(k, l, m)$ are the components of the wave vector and $H$ is the atmospheric density scale height ($\rho = \rho_0 \exp\left(-\frac{z}{H}\right)$). GROGRAT takes into account background fields varying in space and time, allowing for 4D ray-tracing of GWs. The background fields (cf. Sect. 2.2) are internally interpolated to the current ray location using precalculated cubic spline coefficients. This allows for efficient calculation and avoids discontinuities in the background variables and their derivatives.

The dispersion relation in Eq. 5 is derived for small-amplitude waves in a slowly varying background flow and neglecting 240 vertical wind. Acoustic waves are neglected within the derivation of the Boussinesq approximation. However, in the final calculations, the wave amplitude grows with decreasing density as usual. For the full theory, see (e.g. Nappo, 2012; Fritts and Alexander, 2003).

For the prediction of GW amplitudes along the ray path, GROGRAT considers the vertical flux of wave action, $F = c_{g,z}A$, where $A$ is the wave action and $c_{g,z}$ the vertical component of the group velocity. The corresponding equation is given by 245 (Marks and Eckermann, 1995, , Eq. 4):

$$\frac{dF}{dt} = -\frac{2}{\tau}F - Fc_{g,z}\nabla \cdot \boldsymbol{j}, \tag{6}$$

where $\boldsymbol{c_g} = c_{g,z}\boldsymbol{j}$ is the wave's group velocity and $\tau$ the parameterized damping time scale. The last term on the right-hand side is neglected since it would need evaluation using a "ray tube" technique, which is not implemented in GROGRAT (c.f. Marks and Eckermann, 1995; Lighthill, 1978). A more precise consideration using conserved wave action along the path requires a 250 much more involved description of the wave packet in full phase space (c.f. Muraschko et al., 2015). For an estimation of the error introduced by neglecting the last term in 6, we approximate it locally from background fields and compare the results to standard GROGRAT in App. D. In our specific investigation, this correction leads to unchanged GW amplitudes (horizontally) close to the sources and enhanced amplitudes for GWs propagating far from the sources. Since horizontal deformation of the wave packet is caused by refraction, turning, and changing backgrounds, laterally far propagating GWs are especially prone to 255 this effect.

Although the results in App. D are reasonable, for consistency with previous studies we are not taking the correction into account here, and all following simulations of this study are performed with the standard GROGRAT amplitude calculation. The currently-used method of ignoring the last term in Eq. 6 has been used in a number of studies with validation to observations (Preusse et al., 2009; Krisch et al., 2017; Krasauskas et al., 2023). Compared to this, the approximated ray-tube method is 260 less well-validated and deserves a dedicated study to be properly introduced and tested. Therefore, we give only first results in App. D for a demonstration of the size of the effect.

The following calculations of this study are done with the standard GROGRAT amplitude calculation.

In addition to the approximated amplitude estimation, GROGRAT might in principle suffer from occurrence of caustics (e.g. Lighthill, 1978), which, however, do not strongly affect the simulated amplitudes as discussed by Hertzog et al. (2002).

Damping due to turbulence of the background is based on approaches presented in Hines (1960) and Pitteway and Hines (1963), depending on the inverse Prandtl number and the background diffusion coefficient (the vertical profile of which is taken from the approximation in Hocking (1991)). Radiative damping terms are taken from Zhu (1993).

In this study, we use the implementation of the saturation scheme described in Fritts and Rastogi (1985). This takes vertical dynamical (Kelvin-Helmholtz) instability, which is especially relevant for low-frequency waves, and convective instability,
where the wave's local temperature perturbations break vertical stability, into account.

For more details on the inner workings of GROGRAT, see Marks and Eckermann (1995) and Eckermann and Marks (1997).

### 3.3 Representation of ray-tracing data

For the analysis and discussion of Sect. 4 and Sect. 5, ray-tracing data generated by the MWM is presented using two approaches: as residual temperature structures, i.e. the temperature perturbations associated with the GWs, and as momentum
flux distributions. The post-processing steps to generate these data sets from model output are described in this section.

#### 3.3.1 Residual temperature reconstruction

For the residual temperature fields, we aim to reconstruct GWs and their temperature perturbation on a specified spatial $(x, y, z)$ grid for a selected time $t$ for all considered rays, i.e. as a superposition of all predicted MWs. In particular, for the selected time $t$ all rays that are still propagating (i.e. launched before but not yet terminated) at this time are considered. Each individual ray-
280 trace is represented by a wave packet centered around the ray-path position of the reconstruction time. The parameters needed for this reconstruction, that is spatial location, wave vector, phase, and amplitude, are linearly interpolated to the selected time. The phase at a given point of the spatial reconstruction grid is calculated via

$$\phi_{\text{tot}} = \phi + k_{\text{hor}} d_{\text{along}} + m d_z + \frac{c_{\text{m}}}{2} d_z^2. \tag{7}$$

Here $k_{\text{hor}}$ and $m$ are the horizontal and vertical wavenumber, $d_{\text{along}}$ is the horizontal distance from the center of the GW in
direction of the wave vector and $d_z$ the vertical distance of the corresponding grid point to the ray's location. $\phi$ is the current phase at the ray-path of the wave given by the ray-tracer. This phase is calculated by GROGRAT via integrating $\phi(t, x_i)$ from launch to the current position along the ray path (Marks and Eckermann, 1995). The last term accounts for a linear approximation of the change in vertical wavelength along the vertical with chirp rate $c_{\text{m}} = \frac{\Delta_{\text{m}}}{\Delta_z}$ and $m(z) \approx m(z_0) + c_{\text{m}} d_z + \mathcal{O}(d_z^2)$. The chirp rate is calculated as the finite difference derivative of $m$ for the closest time steps around target altitude $z$.
The linear approximation of the dependence of the vertical wavelength on altitude increases the reconstruction performance significantly where it changes rapidly, e.g. below critical layers (e.g. Nappo, 2012). In testing, we found that considering only the leading order, i.e., $m(z) \approx m(z_0)$, leads to inconsistent phase transitions between wave packets excited by the same mountain ridge at different times.

The spatial extent of the wave packet perpendicular to the wave vector is estimated using the length $l$ of the ridge exciting the
MW. In this direction, the amplitude is scaled with an additional symmetric 6th-order Butterworth function, $\left(1 + (x/S)^{12}\right)^{-1/2}$,
with $S = l/2$ as scale for a smoother transition to zero at the edges. In the direction of the horizontal wave vector, a Gaussian-
like envelope with $\sigma = \frac{\lambda_{\mathrm{hor}}}{2}$ is applied. Likewise in the vertical direction a Gaussian with $\sigma = \frac{\lambda_z}{2}$ is used. The total contribution
of a single wave packet is therefore given by

$$T = -T_{\mathrm{amp}} \cos(\phi_{\mathrm{tot}}) \frac{1}{\sqrt{1 + \left(\frac{2d_{\mathrm{perp}}}{l}\right)^{12}}} \exp\left(-\left(\frac{2d_{\mathrm{along}}}{\lambda_{\mathrm{hor}}}\right)^2 - \left(\frac{2d_z}{\lambda_z}\right)^2\right), \tag{8}$$

where $T_{\mathrm{amp}}$ is the temperature amplitude given by the ray-tracer, $d_{\mathrm{along}}$, $d_{\mathrm{perp}}$, and $d_z$ are the horizontal distances along and
perpendicular to the wave vector and vertical to the location given by the ray-tracer. $\lambda_{\mathrm{hor}}$ and $\lambda_z$ are the interpolated horizontal
and vertical wavelengths, respectively. Note that the waves start with a cold phase directly above the mountain where the
integrated phase along the ray is $\phi_{\mathrm{tot}} = 0$.

There are a few uncertainties introduced by the simplifications that have been made in this reconstruction of temperature
perturbations. For one, in Eq. 7, the horizontal change of wavenumbers $k$, $l$, and $m$ is neglected, since only the vertical
change can be calculated reliably from a single ray path. In addition, the vertical change of horizontal wavenumbers has been
neglected, because the vertical change in vertical wavenumber dominates here (especially when approaching critical levels (e.g.
Nappo, 2012)). In addition, the change of the shape of wave packets is not considered here as their footprint is assumed to be
rectangular at all times. Due to the horizontal extent of some wave packets and the horizontal shear of group velocities, this is
only correct in a first approximation, but since the exciting mountain ridges, in general, are of limited length this approximation
is reasonable.

The total temperature perturbation, i.e. the total distribution of residual temperatures, is taken as the superposition of all
these individual fields for all ray-traces.

### 3.3.2 Momentum flux

For comparisons with satellite and other model data, GWMF of all rays has to be expressed as a superposition on a regular
spatial grid. Similar to the residual temperature, the wave parameters of the rays are interpolated to the considered time.

The spatial distribution of the pseudo momentum flux (further also referred to as GWMF) is performed across the specified
data grid using the same wave packet assumption as for temperature perturbations in Eq. 8. The maximum pseudo momentum
flux of the wave, $F_{\mathrm{max}}$, is given by the raytracer (Marks and Eckermann, 1995), but can also be calculated from the temperature
amplitude following the relation given by Ern et al. (2004). They state GWMF $\sim T_{\mathrm{amp}}^2$ and therefore the GWMF of a wave
packet, $F$, has to decay faster than the temperature perturbation (by a factor of 2). The oscillating term, $\cos\phi_{\mathrm{tot}}$, has to be
dropped because $F$ depends only on the amplitude and not the phase. As in Sec. 3.3.1, the edges of the wave packet are
smoothed with the same 6th-order Butterworth function. The resulting equation in analogy to Eq. 8 is then given by:

$$F = F_{\max} \frac{1}{\sqrt{1 + \left(\frac{2d_{\mathrm{perp}}}{l}\right)^{12}}} \exp\left(-2\left(\frac{2d_{\mathrm{along}}}{\lambda_{\mathrm{hor}}}\right)^2 - 2\left(\frac{2d_z}{\lambda_z}\right)^2\right). \tag{9}$$

The size of a wave packet is finite compared to the 1D trajectory given by the ray-tracer and a single ray may contribute to several grid cells of the regular target grid. On the other hand, if a grid cell is larger than the wave packet, only a fraction of the grid cell is covered and this has to be taken into account by normalization. In order to account for this effect, we are supersampling the GWMF of each wave on a finer grid (here on a $3 \times 3$ subgrid resolution for each grid point, which suffices to sample even the smallest scale waves of this study) and averaging over the finer grid points within each original grid box.

This gives us a numerical approximation of the grid cell integrated GWMF. We estimated the error of this approximation to be below $\sim 5\%$ for randomly oriented GWs of $100\,\mathrm{km}$ horizontal wavelength and decreasing for longer GWs. Since we are interested in the GWMF density, we divide the total contribution by the grid cell's area:

$$F_{\mathrm{tot}} = \frac{F_{\mathrm{grid}}}{A_{\mathrm{grid}}} = \frac{\int F dA}{A_{\mathrm{grid}}}, \tag{10}$$

where $F_{\mathrm{tot}}$ is the wave's contribution to the GWMF density distribution, $F_{\mathrm{grid}}$ is the GWMF in the corresponding grid cell,

the integration is across the (horizontal) area of the grid cell and $A_{grid}$ is the total horizontal surface area of the corresponding grid cell in the data grid.

    The total GWMF density distribution is again calculated by summing up the single contributions of every ray-traced wave packet for each time step.

## 4   Model Validation

In this section, we validate the MWM presented in Sect. 3 in two different ways. First, we investigate the performance of describing the underlying topography and its structures by the idealized ridges and second, we look at the resulting temperature residuals of MWM predicted mountain waves in comparison to ECMWF IFS data.

### 4.1   Detected structures and scales in the MWM

    To investigate the capability of the MWM to adequately represent the topographic structures, we show a comparison of the

underlying elevation to small ($\leq 150\,\mathrm{km}$) and large scales ($>150\,\mathrm{km}$) of detected ridges in Fig. 2. Although a perfect reconstruction of the underlying topography is not what we wanted to achieve with the MWM, the ridge-like mountain chains should be represented by the model. Our algorithm of fitting idealized ridges of various scales and sampling in four directions implies that a single topographic feature could be captured multiple times. This is especially the case for very large-scale and plateau-like features. Therefore, the elevation height of the reconstruction might be higher than the topography itself.

Higher values in absolute (superposed) terrain height do not inevitably lead to an overestimation in GW activity. For example, if there are two ridges lying on top of each other at a right angle, depending on the wind, only one of both ridges will excite a GW. Therefore, the correlation between terrain height and GW temperature amplitude is not as straightforward.

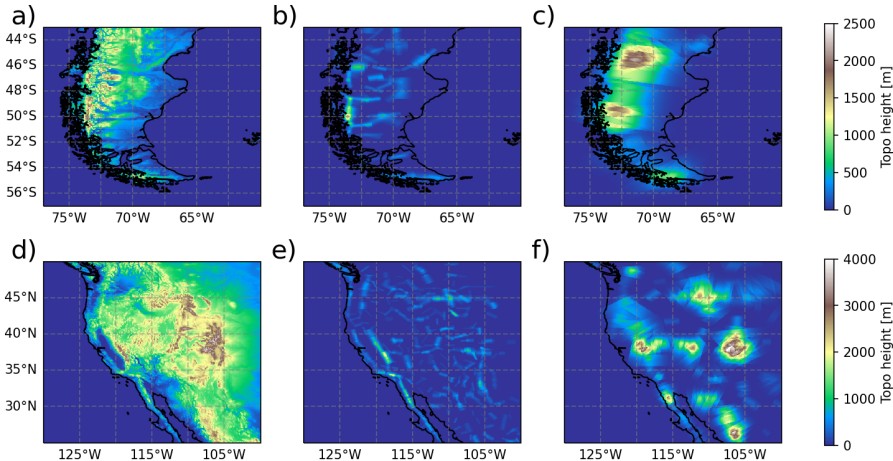

**Figure 2.** Comparison of the underlying topographic data (a and d) to the reconstruction from all idealized Gaussian mountain ridges identified by the MWM (b, c, e and f). Panel a, b and c show the Southern Andes region, panel d, e and f the west of North America. The reconstruction is separated into small scales ($\leq 150\,\mathrm{km}$, panels b and e) and large scales ($>150\,\mathrm{km}$, panels c and f).

Figure 2a shows the true elevation of the terrain of South America, while panels b and c show the small and large scales detected by the MWM, respectively. The reconstructions are generated by superposing all found ridges linearly. The small-scale ridges are located mainly in the main north-south oriented mountain ridge where we would expect them, but additionally, there are also quite a few in the east and at the tip of Tierra del Fuego with an east-west orientation. These eastern ridges can also be found in the topography itself. The general structure represented by the small scales agrees well with what we see from the topography. The large scales form a broad ridge-back along the Andes with a maximum above the highest elevation of the southern Andes around $50°$S and another one to the north, where the topography shows a large-scale plateau-like elevation, that is sampled along multiple directions and therefore more pronounced in the reconstruction.

In Fig. 2d, the elevation profile of the east coast of the USA including the Rocky Mountains is shown. Again, panels e and f show the detected small and large scales, respectively, as detected by the MWM. This case poses a more difficult problem for the MWM due to the topography being even more complex. While South America exhibited ridge-like features reaching down to sea level from the start, the Rocky Mountains are already located on top of a very large-scale elevation feature. Nevertheless, the small-scale ridges represent elongated mountain chains, like for example the Sierra Nevada and Baja California, well and the structures can be matched to the topography. A more scattered structure is seen in the large-scale features. Increased elevation is found above high mountain ranges, like the Rocky Mountains, Sierra Nevada, and Sierra Madre Occidental. Again the topography is well represented by the identified idealized mountains.

More detail on the reconstruction of the southern Andes topography for different scales and a similar analysis for the Himalaya and South Africa region, considered later on in this study, is given in App. C.

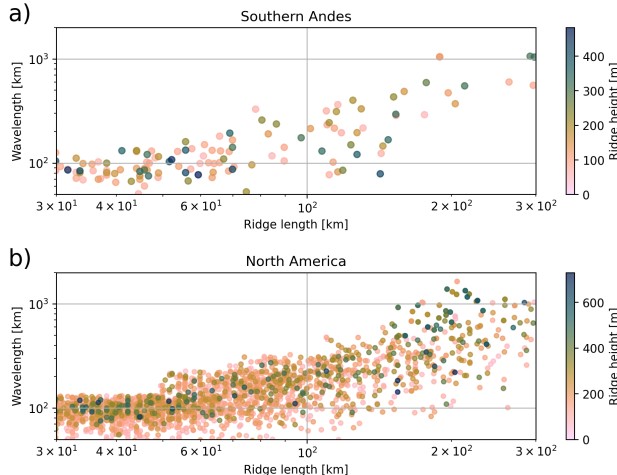

**Figure 3.** Scales of detected ridges that contribute to the approximation of the Southern Andes region (Fig. 2b and c) and western North America (Fig. 2d and e). The scatter shows the length of the ridge versus the detected wavelength in km (cf. Sect. 3.1) and the corresponding height in color shading. Note the logarithmic scales on both axes.

The different scales of the detected mountain ridges and their distributions are shown in Fig. 3 for South America and the Rocky Mountains. Both cases are very comparable in their general distribution of ridge height, length, and horizontal wavelength. The MWM detects mostly ridges with small lengths between ∼40 km and 200 km. Although there are a few outliers with lengths of up to ∼400 km, a natural topography is not easily described by straight ridges due to curving or bends in the mountain structure. Thus the longer ridges are usually only detected for the largest scale, low amplitude features (high amplitudes are only expected for plateau-like topography). Ridge heights are almost evenly distributed in a range of ∼100 m to 600 m and extend up to 800 m. The distribution is minimally shifted towards higher ridges in the Rocky Mountain case (Fig. 3b) accounting for the higher elevation in this region. Considering the horizontal wavelengths of the detected ridges, they are evenly distributed between 70 km and ∼300 km with sparser distribution for longer scales up to 1250 km. This can be attributed to the small scales in general being of shorter length as well and thus, they do cover not as much area and multiple ridges are needed for the description of the topography. The larger-scale ridges on the other hand cover a large area and fewer of them are needed for a similar region.

In conclusion, the MWM detects and represents orographic features of the elevation of various scales under consideration that the representation of elevation data by two-dimensional ridges has some intrinsic problems (especially with isotropic and plateau-like features). Although this is no indication of whether we cover all relevant scales, it provides confidence in the underlying ridge detection algorithms as a tool to extract the main MW sources and the corresponding parameters.

## 4.2 Residual temperature as compared to ECMWF operational analysis data

Figure 4 compares the residual temperature due to gravity waves taken from ECMWF IFS operational analysis data on 21.09.19 at 06:00 UTC to a reconstruction estimated from MWM data. As described in Sect. 2.2, the IFS data set has been scale separated in the same manner as the ERA5 data used for ray-tracing backgrounds, only that now we are interested in the remaining residual fields representing small-scale gravity wave perturbations instead of the background. In the considered region, the resolution of $0.1°$ corresponds to a horizontal resolution of about $10\,\mathrm{km}$ meridionally and $6\,\mathrm{km}$ zonally. We estimate the smallest scale GWs resolved in the data set to be about $80\,\mathrm{km}$ horizontal wavelength by the scale, where the energy spectrum deviates strongly from a slope of $k^{-3}$ (c.f. Skamarock, 2004). The reconstruction of residual temperature from the MWM is described in Sect. 3.3.1.

First, we consider the residual temperatures below the tropopause at an altitude of $8\,\mathrm{km}$ (Fig. 4 a and b). At this altitude, we do not yet expect strong horizontal propagation of smaller-scale MWs, which typically starts to become relevant above the tropopause, thus we expect the MWM to exhibit GW patterns close to the main mountain structures. Nevertheless, the MWM data exhibits a large-scale pattern to the east of the continent above the Pacific Ocean, indicating that MWs of comparatively large scales (and thus high horizontal group velocity) are strongly propagating below the tropopause. And indeed, we see similar structures in the IFS model data. Note, however, that the IFS models GWs of all sources, and therefore, the seen features could be (partly) due to convection, jet fronts, and geostrophic (or spontaneous) adjustment (where an out-of-balance jet radiates excessive energy as inertia-gravity waves (e.g. Fritts and Alexander, 2003; Williams et al., 2003; de la Camara and Lott, 2015)) or even other tropospheric processes, that are not filtered out well in our scale separation. Above the Andes, we see the typical north-south oriented wave fronts and an enhanced region of small-scale perturbations at around $50°\mathrm{S}$, $74°\mathrm{W}$ in both data sets. The warm temperature phase follows the coastlines in both data sets. The MWM also shows trailing waves at the tip of Tierra del Fuego, which are oftentimes seen in temperature observations of this region (e.g. Preusse et al., 2002; Jiang et al., 2012; Kruse et al., 2022). Both data sets agree in terms of the magnitude of temperature amplitudes.

At $20\,\mathrm{km}$ altitude (Fig. 4d and e), we expect to observe at least some oblique propagation of MWs due to stronger winds and wind gradients. Still, both data sets show the warm phase following the coastline, with trailing waves in the south. The highest amplitude residuals are found in the same location of $\sim42.5°\mathrm{S}$, which is in the lee of the highest elevation topography. In terms of structure, both show a large-scale pattern interfering with GWs of significantly shorter wavelengths than the main structure, about $100$–$150\,\mathrm{km}$ and smaller. Features of very short scales (i.e. below about $60\,\mathrm{km}$) are not as well represented by the MWM temperature reconstruction as by the IFS, which is partly due to the lower scale limit set to $80\,\mathrm{km}$. Above the Falkland Islands, both show GWs that propagated towards the east, the GWs of the MWM are of relatively low amplitude, while the IFS shows higher amplitude and larger-scale waves.

At $30\,\mathrm{km}$ altitude, the polar vortex turns from mainly eastward to a more north-eastward direction above the Andes. This leads to a change in the horizontal wind gradients, and in turn to GWs refracting and turning as well (Fig. 4g and h). Both data sets show gravity waves mainly facing to the south-west instead of westward, as a consequence of horizontal refraction towards the stronger winds in the southeast (Krasauskas et al., 2023). The MWM predicts significant MW activity above the Atlantic

Ocean, which is (mostly) stemming from the main Andes mountain ridge and has propagated to the east. These patterns are confirmed by the IFS data, which shows a large connected band of GWs above this region. The MWM does not show such high continuity due to the (short) mountain ridge approximation and reconstruction from single wave packets. Nevertheless, the phases of different wave packets fit well together and a coherent structure is seen. Propagation to the west is only faintly predicted by the MWM, while the IFS data shows strong GWs far above the ocean.

We analyzed the spectral characteristics of the waves around $45°$-$50°$S, $60°$-$66°$W in Fig. 4g at $30\,\mathrm{km}$ altitude using the S3D technique (3D wave fitting in a subdomain, previously used by e.g. Preusse et al. (2014); Geldenhuys et al. (2021); Krasauskas et al. (2023). S3D returns estimates of the 3D wavenumber $(k,l,m)$ which are then used as input for a reverse ray-tracing calculation using GROGRAT. This calculation suggests that these waves originate in a region with large-scale, low elevation topography leeward of the main Andes ridge ($\sim 49°$ - $51°$S, $70°$W in Fig. 4). Since this topography is "plateau-like" it is missed by our ridge finding algorithm (not shown). It is also possible that these waves were excited by orographically-linked convection, which was observed by Worthington (e.g. 2002, 2015). These processes are currently not accounted for in the MWM. Smaller-scale features in the northern part are represented correctly in the MWM both in terms of orientation and amplitude. Also, both models show small-scale fluctuations at around $\sim 50°$S, $73°$W.

Figure 4c, f and i shows the wind situation at this time. We see a turning wind with altitude: while the wind is passing the Andes mostly eastward below $20\,\mathrm{km}$, the wind turns north-north-east above. Thereby GWs are refracted and the phase fronts change alignment to north-west to southeast. The turning of GWs in the Southern Andes region (same as in Fig. 4) within the MWM can be seen in Fig. 5, which accounts for all GWs launched between 20.09.19 00:00 and 21.09.19 23:00 with each mountain ridge launching a single GW every hour. Although there are MWs launched with various directions between southward and westward, at around $25\,\mathrm{km}$ altitude most of the MWs turn south-westward, leading to this being the dominant wave orientation. This also implies that the change in wave field characteristic is not happening due to filtering, i.e. breaking (e.g. at a critical level) and thereby reduced visibility, of westward facing GWs, but instead due to refracting GWs in the turning wind profile.

In comparison to the high-resolution IFS simulations, the MWM does perform quite well, if we consider the simplicity of the approach. Of course, we do not expect the MWM to represent the exact structure of the IFS, since nature is more complex than this superposition of linear 2D-like MWs can account for. The aim of the MWM is to predict the horizontal propagation in a comprehensive way allowing for a straightforward investigation of propagation paths and momentum transport patterns. We have seen that the model is capable of doing so and therefore is a useful tool to estimate MW residual temperature structures. Especially the GW field characteristics, like turning and propagation of momentum, are captured quite well, and therefore the MWM can be used to investigate GW parameters of observations and propagation patterns in the following section.

## 5 Prediction of global GWMF distributions

In the following, we use the MWM and it's GWMF predictions to explain some observational features of monthly mean HIRDLS satellite data via investigation of the GW parameters and wind profiles, i.e. GW filtering due to wind conditions that

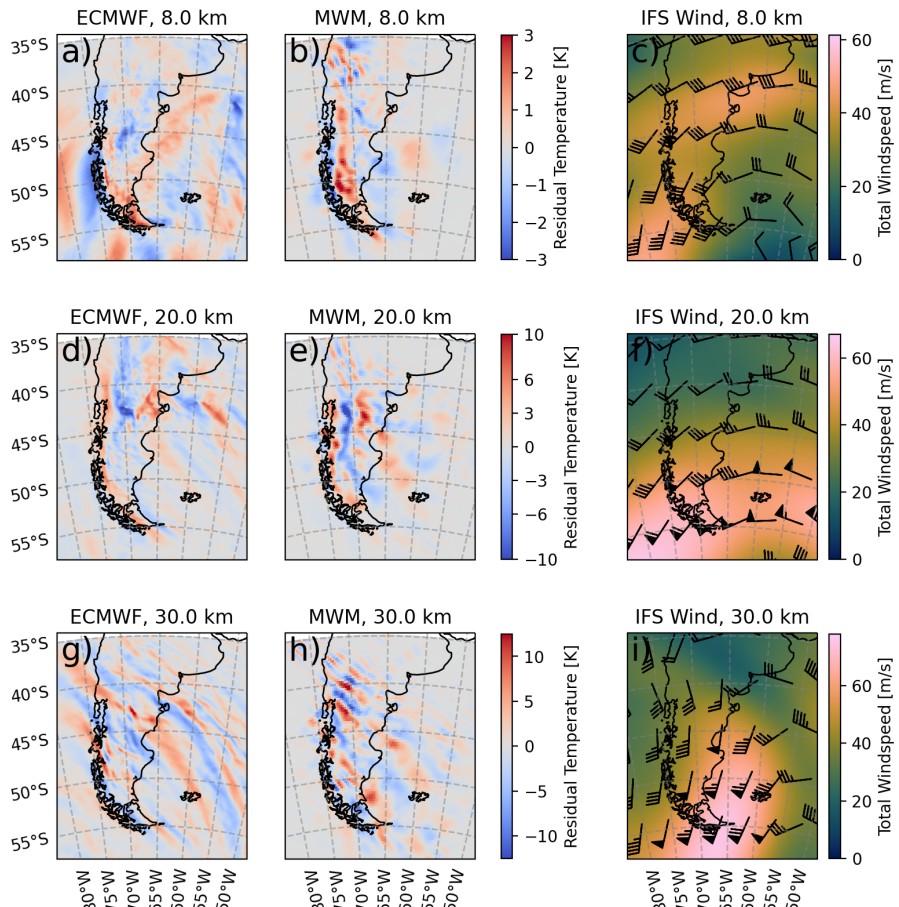

**Figure 4.** Temperature residuals from ECMWF IFS operational analysis data for 21.09.19 06:00 UTC (left column) and the corresponding reconstruction from MWM data at the same time (middle column). Horizontal cuts are shown at altitudes of 8 km, 20 km and 30 km in the top, middle and bottom row respectively. Note the differing color scales between different altitude levels. Synoptic wind fields as predicted by the IFS are shown on the right column, where colorshading gives the total wind speed and wind barbs show the wind direction.

lead to a critical level for MWs. In particular, we try to answer the question of why there is not as much GW activity seen
by satellites as expected above the Himalaya and the Rocky Mountains. Afterward, we look into predicted orographic GWMF throughout the year, which agrees with previous studies of MW propagation. Here, the MWM predicts strong year-round horizontal MW propagation.

## 5.1 Global distributions of momentum flux

HIRDLS in general does not show strong GW activity above the Himalaya and Rocky Mountain region in January (Ern et al.,
2018), contrary to the general understanding, that mountain waves are able to propagate far into the stratosphere in the winter

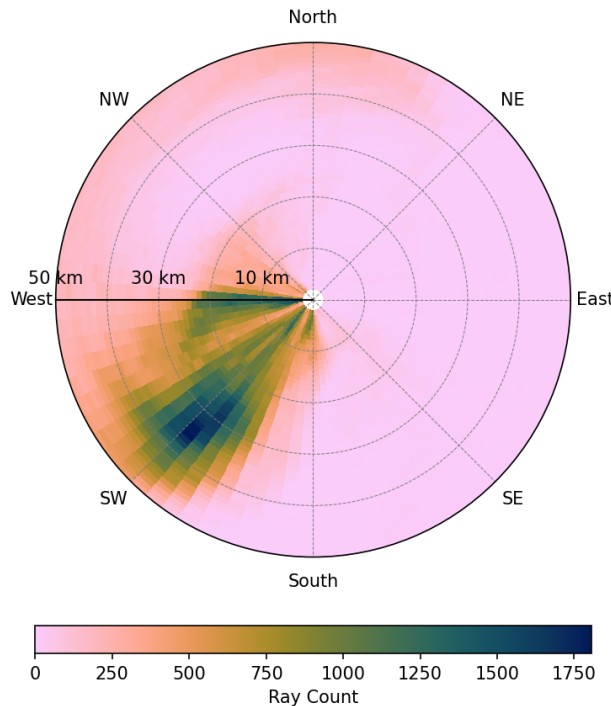

**Figure 5.** Direction vs. altitude distribution of MWs as modeled by the MWM in the Southern Andes region (same as in Fig. 4). This considers all GWs launched between 20.09.19 00:00 and 21.09.19 23:00 with each individual mountain ridge launching a single GW every hour. The radius gives the altitude of a given MW, angle gives orientation of the wave vector. The color-shading gives the total count of rays for a given altitude-orientation bin. There is a clear trend for GWs to turn to south-west at around 25-30 km. Although the sum over all orientations for a given altitude is monotonically decreasing with altitude, there is a maximum at around 35 km with south-west oriented GWs.

months. We will investigate possible reasons behind this in the following by comparing global horizontal distributions of GWMF retrieved from HIRDLS to predictions of the MWM at altitudes of 16 km, 20 km, and 25 km. We will also compare both data sets in terms of propagation patterns found in winter above the Drake Passage and the Southern Ocean.

The distributions retrieved from HIRDLS show gaps in the lower altitudes due to the tropopause and clouds. The former limits observations in the tropics and parts of the subtropics at 16 km and 20 km, while the latter limits observations in regions of subtropical convection from the data at 16 km. The satellite observations pick up GW signatures of all sources. In winter these are mainly orography as well as frontal systems and jet imbalances (Wu and Eckermann, 2008; Alexander et al., 2016), in summer the main source is convection. In contrast, the MWM only shows orographic GWs, which has to be kept in mind when comparing both. In particular, this leads to possibly higher observed GWMF throughout the observations, which might not be

mirrored by the MWM data. This could either suggest a different GW source, if there is no similar structure in the MWM data, or a superposition of orographic and other sources if a feature is structurally seen in the MWM.

For a direct comparison of global GWMF distributions, we apply an observational filter that accounts for HIRDLS observation geometry (Trinh et al., 2015) to every ray of the MWM before calculating the GWMF distribution following Sect. 3.3.2 on a $1.5°$ resolution and bin the resulting GWMF in the same way as the HIRDLS profiles in order to derive comparable global distributions (see Sect. 2.3). Two case studies for January and July 2006 are presented in this section.

### 5.1.1 January 2006

Figure 6 shows monthly mean total GWMF distributions for January 2006 as retrieved from HIRDLS (left column) and predicted by the MWM (right column) at altitudes of 16 km, 20 km and 25 km. The strongest pattern in the HIRDLS observations is found above the Himalaya and Altai Mountains (Mongolia), where we see two local maxima. The maximum above the Himalaya dominates at 16 km, but, with increasing altitude, weakens stronger than the one above the Altai mountains. Therefore, at 25 km only the pattern above Mongolia remains. This pattern is also rather consistent throughout the considered altitudes. The same structural pattern is seen in the MWM, although with lower amplitudes. Features in both regions show comparable local maxima at lower levels and, at higher altitudes, the one above the Himalaya is reduced analogously to the observations.

To understand this northward shift, we can investigate the properties of GWs in the different regions within the MWM. Fig. 7 shows histograms of horizontal and vertical wavelengths at 16 km, and 25 km for both regions as calculated from the MWM. We see that the horizontal wavelengths are almost the same at both altitudes and regions. Conversely, the vertical wavelengths differ strongly. The MWs of the southern region (above the Himalaya) exhibit longer vertical wavelengths, which are also strongly suppressed by the observational filter (with a cutoff at $\lambda_z = 12$ km) at 16 km altitude. Propagating upwards, they refract strongly towards short vertical wavelength due to a negative vertical gradient of zonal wind (see Fig. 9). There are at least two possible reasons for these GW features missing in the satellite observations at higher altitudes that are considered here: For one, the vertical resolution of HIRDLS is about 1 km (Wright et al., 2009), which, in principle, allows for the detection of GWs with a vertical wavelength as low as 2-4 km. Some waves are refracting even below this vertical wavelength and can therefore not be picked up by the instrument. Another reason might be strong amplitude GWs, which are often found in this region, that could completely break instead of propagating further with an amplitude reduced below the saturation limit in a strong wind vertical shear (Kaifler et al., 2015). Such a complete breakdown of GWs is currently not captured by GROGRAT simulations. The MWM could be a suitable tool to test this hypothesis in other, more specific case studies, by implementing different breaking schemes.

A very strong GW signature in the MWM is seen above the Rocky Mountains. The amplitudes are overestimated in comparison to the satellite observations, nevertheless a similar structure is visible. In higher altitudes, this signature is strongly reduced until it almost vanishes at 25 km. There is also a minor southward shift of GWMF towards California visible which is also hinted at by the satellite data. This feature sits, however, right at the edge of the observation and is therefore not completely seen. The strong reduction in amplitude can be explained by similar arguments as for the Himalaya region: there is a strong negative vertical wind shear above the Rocky Mountains reducing the allowed amplitudes of GWs strongly. Compared to the Himalaya region, in the MWM the MWs launch with much stronger amplitudes (about a factor of two, not shown), making them more likely to encounter saturation or complete breakdown (there is plenty of evidence of strong MWs and their breaking

in this region, e.g. Guarino et al. (2018)). The latter process might be a reason for the overestimation at 16 km (the negative wind shear starts already at roughly 10 km, where waves could already break). Therefore, the strong signature seen in the predictions could be an indication of a process that is not yet modeled within the MWM.

Another strong feature predicted by the MWM is local maxima in the Southern Hemisphere above New Zealand and the southern Andes, which are matched partially by the observations. These maxima strongly decrease in higher altitudes and vanish completely at 25 km as expected, since MWs are filtered by the wind reversal at around 20 km in the summer hemisphere. The MWM prediction is stronger than the observations, which could, again, be related to the above-mentioned processes. The prediction shows strong eastward propagation above New Zealand, while the satellite data seems to show signs of strong oblique propagation towards the east of both sources. The presence of this pattern at 25 km altitude in the observations is, however, not consistent with them being MWs as well. Instead, these are most likely of other origins.

In the North Atlantic region, the MWM predicts GW sources in Newfoundland, southern Greenland, Iceland, and Scandinavia. Strong eastward propagation of MWs is seen especially above Iceland, where the pattern of GWMF merges with the feature above Scandinavia. In addition, the MWM predicts eastward propagation from Newfoundland towards southern Greenland at 25 km, even though the GWMF values predicted by MWM are, generally, suppressed at this altitude. The HIRDLS observations show similar, although more complex features in this region. The aforementioned MW sources are clearly visible but merge into a band of strong GW activity at 20 km and above. This band follows the path of the polar vortex and might therefore be related to local GW sources such as jet imbalances and fronts (Geldenhuys et al., 2021). The occurrence of other sources than orography in the observations is strengthened by the (slight) GWMF increase between 20 km and 25 km.

To further investigate the reasons for the differences between the MWM and the satellite observations, we use blocking diagrams similar to the ones introduced in Taylor et al. (1993) as well as vertical profiles of horizontal wind. The regions of interest are shown in Fig. 8. We investigate differences in propagation conditions for non-orographic Gws above the Himalaya compared to Mongolia, above the Rocky Mountains, and above southern Africa, where a strong GWMF pattern arises in the HIRDLS data at 25 km altitude (see Fig. 6). For illustration of the general wind conditions, Fig. 8 shows the monthly mean zonal wind at 20 km.

In the blocking diagrams shown in Fig. 9a – d, the criterion of waves encountering a critical level whenever the intrinsic frequency of a GW goes to zero, $\omega_{intr} \to 0$, is used, with:

$$\omega_{intr} = \omega_{gb} - \boldsymbol{k_{hor}} \cdot \boldsymbol{U} \tag{11}$$

$$\Leftrightarrow \quad \omega_{intr} = \omega_{gb} \left( 1 - \frac{U_{par}}{v_{ph}} \right). \tag{12}$$

Here, $\boldsymbol{k_{hor}}$ and $\boldsymbol{U}$ are the horizontal wave- and wind vector, $U_{par}$ is the wind speed projected onto the horizontal wave vector, and $v_{ph}$ the horizontal phase speed of the GW. Note that the derivation of this equation requires $\omega_{gb} \neq 0$, which is not true for the considered MWs close to the launch site or in a static background. In addition, the Coriolis parameter is neglected within this consideration, which would restrict the intrinsic frequency even more ($|f| < \omega_{intr}$) and hence leads to a stronger restriction of phase speeds.

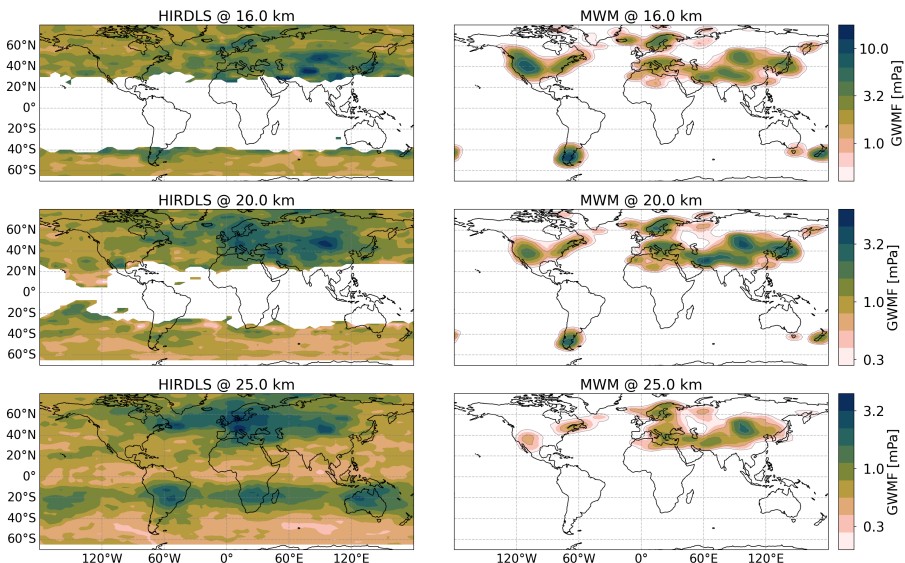

**Figure 6.** Monthly mean of global GWMF distribution for January 2006. Left column shows HIRDLS satellite data (see Sect. 2.3), right column shows distributions produced by the MWM. The different rows present data for 16 km, 20 km and 25 km altitude from top to bottom respectively. Note the logarithmic color scales.

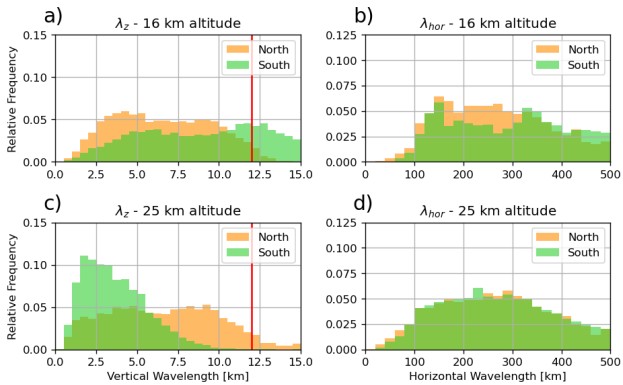

**Figure 7.** Distribution of vertical (left) and horizontal (right) wavelengths as found by the MWM at altitudes of 16 km (top) and 25 km (bottom). The northern region corresponds to the Altai Mountains at $42.5°$-$55.0°$N $75°$-$105°$E, the southern region to the Himalaya at $30.0°$-$42.5°$N $65°$-$95°$E. The vertical red line on the left panels marks the cutoff wavelength of $\lambda_z = 12$ km for the present HIRDLS data.

The curve of $\omega_{\mathrm{intr}} = 0$ is a circle in phase speed diagrams with center at $(\frac{U}{2}, \frac{V}{2})$ and radius $R = \frac{1}{2}\sqrt{U^2 + V^2}$ for zonal and meridional background winds $U$ and $V$ and covers the restricted, i.e. blocked or filtered, part of the phase speed spectrum. These curves can be superposed on a phase speed diagram for all altitudes from the surface up to 25 km altitude to give a measure of how strong or widespread across altitudes the critical levels are for GWs with source at the ground. This is done in Fig. 9a

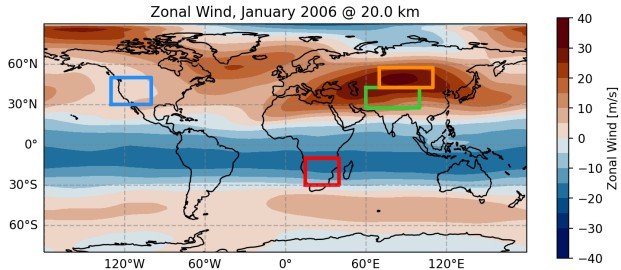

**Figure 8.** Regions of interest for critical layer filtering considered in this study shown on top of the monthly mean zonal wind at 20 km altitude: Himalaya (green), Mongolian Plateau (orange), Rocky Mountains (blue) and southern Africa (red). The same colors as in Fig. 7 have been used for the Mongolia and Himalaya regions.

– d, where the color shading gives the percentage of altitude levels, that exhibit critical levels for GWs of the corresponding (ground-based) phase speed. In other words, the color shading gives an estimate of the probability for GWs with given phase speed to be filtered by a critical level below 25 km. Note, however, that these diagrams can only hint at critical levels for MWs near their launch location and in an approximately constant wind profile, where $\omega_{\mathrm{gb}} \approx 0$. In these conditions, critical levels are encountered wherever the horizontal wind projected onto the horizontal wave vector becomes zero. As an additional metric, the monthly mean vertical profiles of horizontal wind for the four regions are shown in Fig. 9e and f.

Comparing the critical layer filtering patterns for the Himalaya and Mongolian plateau in monthly mean winds in Fig. 9a and c, we see that a wider range of phase speeds is restricted by critical layer filtering in the southern region, above the Himalaya. This potentially leads to a stronger suppression of non-orographic GWs, which might be part of why the GWMF in HIRDLS declines so fast with altitude in Fig. 6. If we look at the vertical profiles of monthly mean winds, which are shown in Fig. 9e, neither of the regions show a wind reversal and thus MWs should in general propagate similarly well in both regions. However, the zonal wind of the southern region exhibits a strong vertical gradient. The wind speed peaks at around 12-13 km with a pronounced maximum of about $45\,\mathrm{ms}^{-1}$, which is roughly 5 km below the level of maximum wind speed of the northern, Mongolian, region. This high wind speed allows for MWs of higher amplitudes, which afterward encounter a strong negative wind shear leading to refraction towards small vertical wavelengths and wave breaking due to reaching the saturation limit. Kaifler et al. (2015) suggests that high amplitude GWs reaching saturation might break completely instead of propagating further with reduced amplitude. The lack of this effect in the GROGRAT simulations could be one reason for the enhanced GW activity above the Himalaya predicted by the MWM (Fig. 6). Above the Altai mountains, the propagation conditions are more favorable due to a more consistently strong wind, that is slowing down only above 25 km. Therefore the observations and the model do not see as strong a reduction in activity as above the Himalaya region.

Since the here considered blocking diagrams are not directly applicable to MWs close to their sources, we additionally show alternative blocking diagrams for MWs with $\omega_{\mathrm{gb}} \approx 0$ in App. E. Figure E1a and c show that due to the wind profile, the (horizontal) phase space in the Himalaya region is less restricted and, therefore, might exhibit more (diverse) MW activity in the stratosphere. The Mongolian Plateau, on the other hand, shows a much more restricted initial phase space. This underlines

that the strong northward shift of the maximum in the HIRDLS observations compared to the MWM could stem partly from non-orographic GWs measured above the northern part as well as MWs refracting to vertical wavelengths that the observation data does not pick up.

Next, we want to investigate the Rocky Mountains. The blocking diagram for this region is shown in Fig. 9b and corresponding wind profiles are given in Fig. 9 f (blue lines). The blocking diagram exhibits high values at and around the origin indicating low wind speeds at various heights. The dumbbell-like shape with structures on either side of the origin is a consequence of a wind reversal. This is confirmed in the wind profiles, where we see the reversal of horizontal winds at about 22 km. This reversal prevents any MW activity to propagate further upward. A possible reason for the wind reversal despite being in the winter hemisphere is the location of the polar jet. In Fig. 8 we see that the polar jet in the monthly mean is not located above the Rocky Mountains but much further north (about $75°N$), which strongly affects the propagation criteria for MWs. In terms of shape of the wind profile, the situation is similar to the Himalaya region with a peak wind speed of about $25\,ms^{-1}$ at $\sim11$ km altitude followed by a strong negative wind shear. In addition, the low-level wind speeds are around $6\,ms^{-1}$, which is four times as strong as in the Himalaya region. The MWs of this region, therefore, have the potential to launch with much higher amplitudes, making them more likely to reach saturation and complete breaking due to high amplitudes as described by Kaifler et al. (2015). Since this is not represented in GROGRAT, the MWM may predict much stronger GWMF in this region, than there actually is in reality.

Lastly, we want to briefly consider southern Africa to exclude MWs as a source for the pattern seen in HIRDLS at 25 km. The corresponding blocking and wind profiles are shown in Fig. 9d and f (red lines). The blocking diagram has a pronounced dumbbell shape from a wind reversal, which is confirmed by the wind profiles at low altitudes (about 4 km). This makes propagation of MWs impossible. Conversely, only small parts of the phase speed spectrum are blocked. These are ideal conditions for GWs of sources that generate a wide range of phase speeds like convection (e.g. Salby and Garcia, 1987; Alexander and Dunkerton, 1999; Preusse et al., 2001; Choi and Chun, 2011; Trinh et al., 2016). Therefore we can conclude that the observed patterns appears not due to orography but other sources (most likely convection).

To summarize, the predictions of the MWM and the calculated wave parameters can explain the shift of focus of GW activity from the Himalaya to the Altai Mountains and therefore solve the question, of why there is not as much GW activity as would be expected from the topography itself. The MWM shows, that parts of the GW spectrum refract to very short vertical wavelengths, which makes them hard to be detected by the satellite. The refraction to smaller vertical wavelengths is distinguished from GW breaking by the GW amplitudes as calculated within GROGRAT, which, in general, do not reach saturation although the vertical wavelengths shorten significantly. The feature above the Himalaya is more pronounced without the instrument-specific observational filter. On the other hand, the MWM is showing a strong signature above the Rocky Mountains, as would be expected from the topography. Since this is not present to such a degree in the satellite data, it might be a hint at a GW process, that is not captured as of now (e.g. total breakdown of GWs reaching a saturation level (Kaifler et al., 2015)). In total, the MWM has proven to be a useful tool to investigate the orographic part of GWMF observations.

## 5.1.2 July 2006

In the following, we are considering predictions of GWMF for July 2006 and using calculated MW parameters (mainly the wave vector) to explain features found in HIRDLS observations. Fig. 10 shows global horizontal distributions of GWMF as retrieved from HIRDLS (left column) and predicted from the MWM after the application of the observational filter (right column) at altitudes of 16, 20 and 25 km. The main feature in both data sets is the maximum above the southern Andes. At 16 km HIRDLS shows a strong maximum at around $\sim 52°$S accompanied by a weaker separate maximum directly north at around $\sim 42°$S. In higher altitudes, the southern maximum vanishes, while the northern persists. In total, this leads to a northward shift of the global maximum.

The HIRDLS data set shows a strong eastward spread around these maxima up to about 30°W over the Atlantic. A comparison with the MWM, which shows a very similar pattern with a larger maximum above the southern Andes, shows that the observed spread of GWMF can be explained by eastward propagating MWs. In a textbook case of oblique propagation from a single source region, the extent should increase with altitude. This is true in the MWM predictions, where MWs reach as far as 30°W at 25 km. The satellite observations, however, show a decrease at 20 km followed by an increase in spread at 25 km altitude. Although the maximum of the MWM prediction is not as precisely localized as in the observations, it shows the same northward shift with higher altitude.

The MWM allows for an in-depth look at what is causing the northward shift seen in observations by investigating individual GW parameters in these regions. Fig. 11 shows histograms of vertical and horizontal wavelengths of all GWs between 37°-47°S and 47°-57°S and longitudes between $40°$-90°W at 16 km and 25 km. This indicates a strong increase in vertical wavelength in the southern part, while the northern part remains almost unchanged. Horizontal wavelengths remain mostly unchanged in both regions. Therefore in the southern part, the GWs refract towards vertical wavelengths larger than the cutoff wavelength of 12 km that is used in the generation of the HIRDLS data set and are thus filtered out at higher altitudes. This finding is confirmed by the unfiltered observation data (without the cutoff at $\lambda_z = 12$ km), which shows a broad maximum at all altitudes (not shown). In addition, the MWM shows more GW activity in the south without the observational filter, which could mean that HIRDLS picks up more of the horizontal spectrum than we assume in the observational filter (c.f. Fig. 13, Trinh et al. (2015)). This strong filtering at higher latitudes is also a likely reason for the relatively weak GWMF predictions of the MWM around the Antarctic Peninsula.

Another major predicted feature is strong GW activity around the Antarctic Peninsula and eastward trailing GWMF, especially at higher altitudes. These predictions agree with observations in terms of extent of the horizontal propagation. The HIRDLS data, however, shows another peculiar pattern: GWMF is increasing again above 20 km altitude. Since the data set is limited to about 63°S, it is not clear where this is coming from. One source could be orographic waves propagating from further south. This is not seen in the MWM though. Even without the observational filter, there is some northward propagation above the Antarctic Peninsula, but it is far too little to compensate the reduction in orographic GWMF with altitude. This feature in the observations might be related to MWs due to katabatic flow (Watanabe et al., 2006) or other, non-orographic processes like imbalances of the polar jet or frontal systems. Partly, this lack of GWMF might also be related to the strong filtering in high

latitudes by the observational filter. Note that although GWs excited by katabatic flow are also categorized as MWs, they are not considered by the MWM since they are excited at drops in elevation, which are not detected by the presented algorithm. The eastward propagation seen at 25 km in both data sets is in agreement with previous studies by e.g. Sato et al. (2012).

Enhanced GW activity is seen in the MWM prediction above the Southern Alps and the Great Dividing Range/Tasmania, which has been shown by Eckermann and Wu (2012) to be a strong MW source. The MWs from Australia and Tasmania show a relatively localized pattern with increasing altitude, while the MWs from the Southern Alps show strong south-eastward propagation especially at higher altitudes. These features are hard to separate from the background in the HIRDLS data and can therefore not be entirely validated. The observation, however, shows some enhanced GWMF around the Southern Alps, which stretches to the southeast and merges with the background at around 170°W. A look into unfiltered MWM data (see Fig. 13) shows that there is strong north-eastward propagation from eastern Antarctica, which can partly explain enhanced fluxes in this region.

Another predicted pattern found in both data sets is enhanced GW activity above the Southern Alps and the Great Dividing Range/Tasmania (a study of this as a MW source has been done by Eckermann and Wu (2012)). The location and strength of local maxima fit nicely between the two. In the observations, these features are of comparable strength as the background and thus difficult to disentangle. At higher altitudes, MW activity from Australia/Tasmania is reduced significantly and rather localized. MWs excited by the Southern Alps on the other hand show strong eastward oblique propagation already at 16 km altitude. In the observations, it is not clear whether the long south-eastward trail from New Zealand is caused by MW propagation or a different GW source, however looking into unfiltered MWM data shows a strong north-eastward propagation of MWs from East Antarctica, which can partly explain enhanced fluxes.

Another predicted feature is MW activity of similar strength to New Zealand above South Africa with propagation towards the southeast. This region is usually not regarded as a hot spot for GWs, however, we will show that this feature is fairly consistent throughout SH winter in Sect. 5.3. In the satellite data, this feature is obscured by the belt of high GWMF in the Southern Ocean but could be interpreted as the bend towards the southeast of Africa, which is seen in this belt.

## 5.2 Zonal mean momentum flux distributions

Figure 12 shows monthly and zonal mean of total GWMF as observed by HIRDLS (a, b) and predicted by the MWM (c – f) for January (left column) and July (right column) 2006. Panels a – d show the height range limited by the HIRDLS data set (14 - 25 km), while panels e and f extend further from 10 - 30 km. The MWM considers only GWMF of orographic origin, while it was shown in Sect. 5.1 that HIRDLS measured high background fluxes all around the globe including all possible GW sources (see Fig. 6 and Fig. 10). In the zonal mean, this leads to rather high values compared to the MWM prediction, which shows gaps above regions, where no MWs are present (e.g. oceans, tropics at high altitude). This leads to the MWM zonal means being of reduced values, which is why the comparison between the data sets is focused on the structural features. The MWM gives information about which patterns in the observations are caused by MWs and which are of other origins. Mainly in the tropics, the zonal cross-sections show gaps in the HIRDLS data due to clouds and the tropopause seen in Sect. 5.1. While clouds restrict the line of sight of the instrument, data below the tropopause has not been used in the diagnosis of GWMF due

to the discontinuity in stability it represents, which could lead to wrongly estimated GW parameters. The black contour lines in Fig. 12 show the monthly mean zonal wind as taken from ERA5 data.

For January 2006, shown in Fig. 12a and c, the same features as in the discussion of Sect. 5.1.1 can be recognized in the zonal mean. A local maximum at around $35°N$ at low altitudes, which moves northward to about $50°N$ at $20\,km$ corresponds mainly to the strong GWMF above the Himalaya and Altai mountains and the associated northward shift with increased altitude. The Southern Hemisphere shows most dominantly the southern Andes at around$45°S$. Another low altitude maximum at $30°S$ in the observations is probably not representative of the zonal mean, as Fig. 6 shows, that it stems from a very small region with few data points west of Australia. The predicted GWMF is mostly restricted to below the wind reversal, although there is a part of the Andean MWs, that reach up to $25\,km$ at around $40°S$. This can be attributed to their horizontal propagation and refraction, to circumvent the critical level filtering (also complete filtering would require zero meridional wind as well).

Corresponding data for July 2006 is shown in Fig. 12b and d for HIRDLS observations and MWM predictions, respectively. As in Sect. 5.1.2, the dominant feature in both data sets is the GWMF above the southern Andes and the Southern Ocean around $40°$-$50°S$. The prediction from the MWM shows enhanced GWMF around the neck region at roughly $16\,km$, $30°S$. A quick check against the MWM without the observational filter applied (Fig. 12f) shows that this is most likely not due to (northward) propagation, but due to the shift of MW parameters towards values that are better observable by the instrument, i.e. towards longer vertical wavelengths, and therefore less suppressed after filtering. The observations do not show southward propagation towards $60°S$ explicitly due to the strong band of GWMF obscuring individual features in the zonal mean. The predicted zonal mean from the MWM, on the other hand, does show neither strong southward propagation from the Southern Andes nor northward propagation of MWs from Antarctica, which could lead to the enhanced fluxes in the southernmost observations at highest altitudes (about $22$–$25\,km$). The model predicts, however, strong northward propagation of meridional momentum flux from the Antarctic Peninsula towards $60°S$ with decreasing intensity (not shown). As mentioned in Sect. 5.1.2, this enhancement seen in the observations is, therefore, most likely of non-orographic origin and generated by sources higher up in the atmosphere like jet fronts and spontaneous adjustment. In the Northern Hemisphere, the strongest MW activity is seen above $65°N$ in the MWM predictions can be attributed to Greenland as a source, followed by another local maximum above the Himalaya around $40°N$. Both features are confirmed by the observations. The MW activity is well confined by the wind reversal in the summer hemisphere in the simulation, as would be expected for MWs.

In order to discuss the effect of the observational filter, raw MWM data without any filtering applied is shown in Fig. 12e and f for January and July respectively. In general, a reduction of absolute values by a factor of roughly 3 - 10 due to the observational filter can be seen, which depends on the wavelengths and the wave's orientation to the satellite track. Nevertheless, there are also structural changes. For January, the maximum above the Himalaya gets reduced and therefore a net northward shift can be seen. Without the observational filter, there is also an additional feature in the tropics extending quite high. At high latitudes, the Maximum above the Antarctic Peninsula and another smaller one in the Arctic are strongly suppressed by the observational filter. This stronger effect at high latitudes is due to the hard cutoff in vertical wavelength at $12\,km$ in the observational filter. In high latitudes, waves of longer vertical wavelength are more dominant since the intrinsic frequency is usually higher due to being confined to $N^2 > \omega^2 > f^2$. The vertical wavenumber in mid-frequency approximation is given as $|m| = N\frac{|k_{\mathrm{hor}}|}{|\omega|}$, where

$N$ is the Brunt-Väisälä frequency and $k_{\mathrm{hor}}$ the horizontal wavenumber. Thus at higher latitudes, where $|f|$ is larger, GWs have in general larger minimal frequencies leading to higher minimal vertical wavelengths.

July shows a very similar picture: the Antarctic Peninsula is strongly suppressed and the maximum between $30°$-$50°$S is reduced in width. Also, the gap at $60°$S is increasingly closing at higher altitudes, while the opposite can be seen after application of the observational filter. This gap is closing even more at higher altitudes than shown. Most of the GWMF propagating into this region originates at the Antarctic Peninsula, but also smaller islands like the Falkland Islands, South Georgia and the Kerguelen Islands as well as the southern Andes contribute.

In both cases, the MWM predicts a comparable strong feature just north of the equator, reaching up as high as $20\,\mathrm{km}$ in January. Since the observations are limited due to clouds, this is only seen in the model data. The origin of the MWs is around Thailand/Malaysia and the Philippines. Another feature, that is worth further investigation are MWs crossing the critical level filtering (e.g. around $40°$S in January). The MWM can be used in an in-depth study, to find the pathways and conditions of these MWs. This is beyond the scope of this model overview study though.

## 5.3   Time evolution of GWMF distributions

Now we apply the MWM to quantify oblique propagation in more detail and get a look at GWMF transport patterns. As presented, the model allows to investigate the evolution of MW activity throughout the year and related seasonal patterns. Fig. 13 shows horizontal monthly mean GWMF distributions for January to December 2006 at $25\,\mathrm{km}$ altitude. Note however that the observational filter used in Sect. 5.1 was not applied to this data, a comparison to Fig. 6 and Fig. 10 can therefore give an indication of the effect of the observational filter.

At $25\,\mathrm{km}$ we expect already strong oblique propagation as in our ray-tracing experiments, the strongest horizontal propagation takes place in a relatively small layer above the tropopause. The summer wind reversal is also below this level. Most MWs in the summer hemisphere are therefore filtered out.

The commonly expected features are evident in the time series: during summer time, MW activity is strongly reduced due to wind filtering at the stratospheric wind reversal. This corresponds to April to September in the northern and December to February in the Southern Hemisphere. In the Northern Hemisphere, however, we see an interesting feature: the MW activity above Asia is suppressed for a longer time period (April to September) than above the Rocky Mountains and in high latitudes (May to approximately August). The reasons behind this could be the position of the wind reversal and might only be a feature special to 2006, which is considered here.

The data shows a band of high GWMF above the northern Atlantic connecting New England to Greenland, Iceland and Europe, which is only interrupted from May to September. The same pattern is visible in HIRDLS observations for January (Fig. 6 a) but was reduced after applying the observational filter to the MWM data.

The total maximum of GWMF throughout the year is located in the Southern Andes and Antarctic Peninsula region, as the current understanding would suggest. There is strong lateral propagation from this region eastward up to the zero meridian, which is in agreement with the findings of Sato et al. (2012). This zonal propagation and advection was not seen to the same extent in the previous filtered MWM data at $25\,\mathrm{km}$ altitude (Fig. 10) due to the rather small strength and specific wave charac-

teristics. The eastward transport of GWMF is an all-year-round phenomenon with similar shape in all months though absolute values vary with season and are strongest in austral winter. An interesting finding is additional propagation westward of the Andes, which occurs throughout the year with maximum extent in October and November. This seemingly leeward propagation has been observed by the 2019 SouthTRAC campaign (Rapp et al., 2021) and extensively investigated by Krasauskas et al. (2023).

A feature not seen in the filtered data for austral winter (Fig. 10) is a stretched band of GWMF starting at South Africa reaching to the Kerguelen Islands and beyond towards Antarctica. This season in general shows the contribution of smaller oceanic islands, like the Kerguelen, South Georgia, and the Falkland Islands, for generating GW activity at around $60°$S and thus slowing down the southern polar vortex in GCMs. These enhanced fluxes above the Kerguelen Islands are also seen in the HIRDLS observations for (austral) autumn months (not shown), while it is overshadowed by the large-scale globe-wrapping GWMF band for July shown in Fig. 10. The distributions agree with the findings of Perrett et al. (2021), which state that GWs above smaller oceanic islands tend to propagate shorter horizontal distances compared to the ones above the Southern Andes and Antarctic Peninsula. The strong MW activity and propagation seen above South Africa is a new finding, that has not been considered before. Due to it's persistence for about 4 months, it has a significant impact on the atmosphere.

In general, the MWM predicts strong horizontal propagation all around the globe, especially in the Southern Andes - Drake Passage - Antarctic Peninsula region and New Zealand. But also other parts, that are not as prominently known for it, like the smaller oceanic islands, northern Atlantic and South Africa, should not be neglected. Another point to be made is the similarity and consistency of propagation patterns. For example, if there is strong activity in the Southern Hemisphere, it will be advected towards the west. This might be a general pattern, that approximates a large part of MW propagation throughout the year. Such a simple, flow independent transport pattern could be used for improving MW parameterizations in GCMs.

## 6 Conclusions

In this study, we present a straightforward approach and model for the localization and quantification of orographic gravity wave sources. Using a similar approach as Bacmeister (1993), we use a fit of idealized Gaussian mountain ridges to topographic elevation data for an approximation of the main orographic features able to excite mountain waves. These Gaussian ridges allow for an estimation of the main MW parameters of horizontal wavelength, orientation, and amplitude. We show that our model is able to represent the general ridge structures found in topographic data. Plateaus and largest scale features (which could also lead to MWs by katabatic flow) are not considered in the current state.

Using a modified version of the ray-tracer GROGRAT (Marks and Eckermann, 1995), we quantify vertical and oblique propagation as well as refraction of excited MWs within the model in time-dependent background atmospheres. Our results show that the MWM is capable of reproducing residual temperature structures comparable to the high-resolution ECMWF IFS operational analysis. Though the agreement is only qualitative, the ones with expected orographic origin are well reproduced regarding orientation, scale, and amplitude. In particular, smaller-scale features agree in location and amplitude between both data sets. Comparisons of global MWM gravity wave momentum flux (GWMF) distributions to corresponding HIRDLS ob-

servations between altitudes of 16 - 25 km provide good agreement both in terms of patterns and amplitude (after application of an observational filter to the MWM data, accounting for the measurement geometry, see Trinh et al. (2015)). This applies in particular to regions directly above mountainous areas but also to downstream regions, where MWs have propagated via oblique propagation. The degree of horizontal propagation is compatible with the results of previous studies of Sato et al. (2012). The agreement is therefore adequate for our subsequent MWM-based analyses.

Investigations of the evolution of MW parameters show that some of the waves refract to very long (Southern Andes) or very short (Himalaya) vertical wavelengths and thus move out of the visibility range of the HIRDLS data set presented here. Refraction leads to a northward shift of the maximum above the Southern Andes and a reduced signal above the Himalaya. In addition, a study of critical level filtering due to the wind profiles shows that GWs, in general, find better vertical propagation conditions above Mongolia compared to Himalaya. This is associated with a stronger suppression of GW (and MW) activity above the Himalaya.

The wind and propagation conditions above the Rocky Mountains show a complex situation for January 2006. The low-level winds are very strong (a factor of 4 stronger than in the Himalaya region) and allow therefore for excitation of strong amplitude MWs. In the lower stratosphere, the excited waves encounter a strong positive vertical wind shear, allowing the amplitude of the waves amplitudes to grow before they reach a strong negative wind shear level above leading up to a wind reversal. In this strong negative shear, high amplitude MWs reach saturation in our model and propagate further at saturation amplitude. However, it would also be possible that they break completely and instead deposit all momentum locally (Kaifler et al., 2015), a process that is currently not captured by the MWM. If this process is physical, the consequence would be that we find an overestimation of GW activity once saturation of these strong MWs sets in. One implication would be that the resulting GW drag will be predicted at the wrong altitude by the ray-tracer, although it is not certain how much energy of such a breaking wave would be redistributed to other processes, e.g. secondary wave generation.

The interpretation of comparisons of monthly and zonal mean GWMF in the height range of 14-25 km to corresponding HIRDLS data proved to be difficult due to systematic differences between the data sets, like the overlapping signals of non-orographic gravity waves in the observations. Nevertheless, we obtain a good agreement in terms of wave structures for regions of high topographic wave activity. Another finding is that it could be worthwhile to implement katabatic MWs in order to obtain improved GWMF predictions northward of Antarctica and southward of Greenland. MWM predictions for July 2006 without observational filtering show, that the gap at $60°S$ is closing mainly by MWs propagating from the Antarctic Peninsula towards the Drake Passage, but also from contributions of smaller islands (Falkland Islands, South Georgia, and Kerguelen Islands).

Global monthly mean GWMF distributions throughout the year show that the oblique propagation of MWs is not only a seasonal phenomenon but important during the whole year. Especially MWs excited in the Southern Andes and Antarctic Peninsula propagate strongly for most of the year, both east- and westward. But also in other regions, e.g. in the Northern Atlantic, horizontal relocation of momentum by MWs is important for a large part of the year (about seven months in 2006). This part of our study also underlines the importance of smaller islands around $60°S$ for the GWMF budget and the need for a better representation of GWs in GCMs.

One of the main goals of this study is the quantification of oblique mountain wave propagation. The results show, that the presented approach is well-suited to shed light on the behavior of MWs and their appearance in observation data (especially since the MWM provides access to the parameters of each launched GW). The model might also help in disentangling the influence of primary mountain waves (which are the only waves our model is considering) and waves of other sources (also secondary waves). Another sign for reasonable representation of oblique propagation is the gap at $60°$S closing at higher altitudes, which is partially seen in the zonal mean GWMF comparison of Sect. 5.1 and Sect. 5.2.

The results of this study suggest that MWs generated by katabatic flow as well as isotropic mountains (e.g. smaller islands) are worth investigation and inclusion in the MWM for it to capture all possible orographic sources. However, the MWM's performance is good and comparisons of the model's residual temperature to observation campaigns are worthwhile as another tool for explaining the measured GWs path and origin. Thereby the model can be used to separate MWs from non-orographic GWs in observations. Our results support that horizontal propagation of MWs is a strong and global effect that stays important throughout the whole year and is currently not considered in lower-resolution GCMs.

Since there is a good agreement of the wave field characteristics of the MWM to observations and more sophisticated models, it can be used as a predictor for the momentum transport of MWs. In particular, we have shown that there is a more or less general pattern of GWMF redistribution throughout the year, which can be used as a first-order approximation of the horizontal momentum relocation in GCMs. Due to the implementation using a ray-tracer, all necessary information, such as location, momentum, and scales, are covered by the MWM, which are needed for the estimation of such a propagation pattern. Implementation of this into a GCM could improve predictability, especially in the polar vortex region. Bölöni et al. (2021) and Kim et al. (2021) have already shown that ray-tracing can be used to improve the representation of subgrid-scale GWs in atmospheric models and that this is a path worth investigating.

*Data availability.* HIRDLS Level-2 data are available via the NASA Goddard Earth Sciences Data and Information Services Center (GES DISC) at https://acdisc.gesdisc.eosdis.nasa.gov/data/Aura_HIRDLS_Level2/ (NASA GES DISC, 2022)

*Code and data availability.* Access to the code and data is possible upon request to the authors.

## Appendix A: Mountain Wave Model Details

### A1  Bandpass filter

The bandpass filter in use in the MWM is based on convolutions with a Gaussian function kernel. Given the small- and large-scale bounds $\lambda_{\mathrm{small}}$ and $\lambda_{\mathrm{large}}$, the corresponding $\sigma$, i.e. the width (or standard deviation) of the Gaussian, of the kernels is

calculated depending on the grid spacing of the topography $(d_x)$ via:

$$\sigma_i = \frac{\lambda_i}{6 d_x}. \tag{A1}$$

In this way, both scales are converted to a pixel scale, which is then used for the smoothing. For the final bandpass filtered topography $H_{\mathrm{bandpass}}$, we subtract the large scales from the small-scale smoothed field:

$\quad H_{\mathrm{bandpass}} = H * G(\sigma_{\mathrm{small}}) - H * G(\sigma_{\mathrm{large}}). \tag{A2}$

## A2    Mountain ridge fit

To find the best-fitting idealized mountain ridge to each ridge candidate detected by the Hough transformation and the corresponding parameters, a least squares fit is performed on a rectangular cutout of the bandpass filtered topography, $H_{\mathrm{clip}}$, around the identified ridge candidate. This cutout is generated by cropping the topography data in a way such that the ridge candidate

is oriented horizontally in the center. The length of this cutout is given by the length of the line segment as given by the Hough transformation, or in other words the length of the ridge candidate, and the width is set to $\lambda_{\mathrm{large}}$, which is the upper cutoff of the bandpass filter.

Since the idealized ridges are of Gaussian shape, i.e. strictly positive, the lowest elevation of the cutout is shifted to zero via

$$\tilde{H}_{\mathrm{clip}} = H_{\mathrm{clip}} - \min\left(H_{\mathrm{clip}}\right). \tag{A3}$$

Following this preprocessing, the fit with the idealized ridge, $f(h,a)$, with the parameters height $h$ and width $a$, is performed by minimizing the least squares difference between the idealized ridge and the cutout of the topography:

$$\operatorname*{minimize}_{h,\sigma} F_{\mathrm{cost}} = \sqrt{\sum_{i,j} \left( f_{ij}(h,a) - \tilde{H}_{\mathrm{clip},ij} \right)^2}. \tag{A4}$$

Here $i$ and $j$ are the (zonal and meridional) grid indices. This cost function is minimized in terms of the ridge parameters $h$ and $a$ using standard methods, which results in the best-fitting idealized Gaussian mountain ridge for the given topography

cutout. The starting values for the optimization are chosen as $h_0 = 100\,\mathrm{m}$ and $a_0$ corresponding to the center of the considered scale interval.

In principle this ridge could take any shape, here we are using a Gaussian shape as given in Eq. 1.

In addition, if even the optimized parameters result in a bad fit, i.e. a high value of the cost function in Eq. A4, the ridge is disregarded and not considered for the final ridge collection.

**Appendix B: The (Probabilistic) Hough Transformation**

The Hough transformation used in this study and its probabilistic variant are described in the following. In addition, the sensitivity on parameters and their choice for this study is given.

The line detection assumes equidistant grid points and is therefore performed on a local Cartesian grid.

## B1 Algorithm and Parameter Choice

The Hough transformation can be interpreted as a discretized version of the Radon transformation, which allows for de- and reconstruction of multidimensional functions to/from the function values integrated along straight lines (e.g. Herman, 2009). This transformation is the basis of computer tomography. While the Radon transformation is formulated on continuous functions, the Hough transformation acts on discretized fields and is thus better suited for digital image and data processing (Duda and Hart, 1972; Kang et al., 1991).

The Hough transformation aims at detecting straight-line structures in 2D images (or a general 2D data set) and explicitly relies on the fact that all lines passing through a point $X = (x, y)$ can be described by the parameters $(R, \theta)$, where $R = x \cos(\theta) + y \sin(\theta)$, for $\theta \in [0, \pi)$. Then $R$ is the line's distance to the origin and $\theta$ the inclination of the line w.r.t. the horizontal. The space spanned by $R$ and $\theta$ is also called the Hough space.

The line detection is performed in a "voting" procedure, where every non-zero entry in the data casts votes to all possible lines passing through this data point and thereby increasing the likelihood of these lines being general features in the data set. Technically this is realized by initializing a zero-valued matrix $Hf(R, \theta)$, also called (Hough space) accumulator because it counts the votes, with discrete dimensions $R \in [-r_{\max}, r_{\max}]$ and $\theta \in [0, \pi)$. For each non-zero entry of the data set with coordinates $X_j = (x_j, y_j)$, a value of $R_{j,i}$ is calculated for every discrete value of $\theta_i$ using the equation above. The corresponding entries in the accumulator matrix, $Hf(R_{j,i}, \theta_i)$, are then incremented by one, "voting" for these lines being a feature in the data set. After repeating this process for all non-zero entries in the initial data set, the accumulator matrix exhibits local maxima exactly at the locations with line parameters $(R, \theta)$ that correspond to straight line structures in the data set.

The probabilistic Hough transformation improves upon this algorithm in a few ways. In the probabilistic version, the non-zero entries are processed in a random order but still "voting" in the same way with votes counted in a Hough space accumulator. However, if at any point a threshold of votes for any line is reached, the data set is tested for the corresponding line feature starting from the last processed data point. This is done by traversing the data set along the corresponding line in both directions in search of further connected points belonging to the same line segment. If a gap of at least length $l_{\text{gap}}$ is encountered, where the data set has zero-valued entries, the line traversing in this direction ends. This results in start and end points of the feature in the initial data set and thereby localizes it fully. If this detected feature is longer than a minimum line length $l_{\min}$, it is accepted as a line feature in the data for further processing.

The main advantage of the probabilistic algorithm for our use case is the fact, that it results in line segments with defined start and end points which directly give the length, orientation, and position of the line feature.

## B2 Sensitivity of the probabilistic Hough Transform

As stated in the main text, the probabilistic Hough transforms capability to detect mountain ridges is highly affected by the minimal line length, $l_{\min}$, and the maximal gap along each line, $l_{\text{gap}}$. This is especially true in the consideration of topography data, since natural ridges typically do not form perfect straight lines, but form arcs and/or other more complex structures. This section investigates the sensitivity of the Hough transform on these parameters in the case of the Southern Andes.

To this end, we tested different combinations of $l_{\min}$ and $l_{\mathrm{gap}}$. The corresponding results are displayed in Fig. B1, where the maximum line gap, $l_{\mathrm{gap}}$, varies with column from 10 km to 50 km and the minimum line length, $l_{\min}$, varies with row from 30 km to 120 km. For small $l_{\min}$ and $l_{\mathrm{gap}}$ (Fig. B1 a), most structures are well covered by detected line segments. However, these are in general very short and thus would not necessarily correspond to 2D-like mountain ridges. For higher $l_{\min}$ in combination with a small $l_{\mathrm{gap}}$ (e.g. Fig. B1 g, k), only very straight structures are detected, which gives good candidates for ideal long-stretched mountain ridges but neglects possible smaller-scale ones. The allowed gap in these line segments is too short to account for any bends in the underlying structures. A small $l_{\mathrm{gap}}$ combined with a very high $l_{\min}$ (Fig. B1 j) is highly restrictive and does detect barely any line segment in the complex topography skeleton. A high $l_{\mathrm{gap}}$ (Fig. B1 c, f) on the other hand leads to line segments spanning in between parallel line structures, where there is no underlying support in either the skeleton or the topography.

Since there is a need to also detect natural curved and non-perfect straight mountain ridges, one needs to balance both parameters. There is certainly a trade-off, if one wants to detect mountain ridges of arbitrary length with a single set of $l_{\min}$ and $l_{\mathrm{gap}}$.

We found that $l_{\min} = 60$ km and $l_{\mathrm{gap}} = 30$ km (shown in Fig. B1 e) results in a reasonable trade-off for both parameters. With a minimal line length of 60 km most of the features are detectable within the scale ranges that we are interested in. This choice of parameters is used for all results in the present study.

## Appendix C: Topography approximation

This section gives more detail on the performance of reconstructing the topography from the fitted ridges. First additional information on the Southern Andes case is provided to give a better overview of how each single spectral band is detected from the topography. Following this is the approximation by ridges of the Himalaya and southern Africa region, which are mentioned explicitly in the text.

### C1  Southern Andes ridge finding

The ridge-finding algorithm operates on scale intervals and detects the ridges in the bandpass filtered topography data. This is illustrated in Fig. C1, where the topography after application of the bandpass filter is shown in the left column and the corresponding reconstruction of detected ridges in the right column. Note that each spectral band was reconstructed by taking the maximum of all ridges that cover the same spot. The four largest scale intervals given in Table 1 yielded no ridges in this region and are therefore not shown.

Figure C1 shows that each individual contribution to the full spectrum of elevation features in the original elevation data is detected and reconstructed in a good way. Features agree in orientation height and length.

## C2 Himalaya and southern Africa

The same topography reconstruction separated in small- and large-scale contributions as in Fig. 2 for the Mongolia region and southern Africa are shown in Fig. C2. Similar comments as for the Southern Andes and Rocky Mountain region are applicable here. The very large-scale plateaus, especially the Tibetan Plateau as a whole, are not described due to the limitation in terms of horizontal scales. small-scale features, on the other hand, are approximated well in terms of orientation and location. The total height for the larger, i.e. 200 km and above, scales are, as for the Rocky Mountains region, over-represented due to multiple ridges contributing to the same feature in different directions.

## Appendix D: Amplitude correction due to horizontal dispersion

The amplitude calculation in GROGRAT neglects contributions from the last term in Eq. 6. This might, however, lead to deviations for dispersing wave packets. Therefore, we estimated the contribution of this term locally according to

$$c_{g,z} \nabla \cdot \boldsymbol{j} = c_{g,z} \left( \partial_x \frac{c_{g,x}}{c_{g,z}} + \partial_y \frac{c_{g,y}}{c_{g,z}} \right), \tag{D1}$$

where the analytical expressions for $c_{g,i}$ were taken from Marks and Eckermann (1995). This gives a local approximation of the change in the horizontal extent of the wave packet along the ray path.

The GWMF experiments for June and July 2006 in comparison to the unmodified GROGRAT amplitudes are shown in Fig. D1 and Fig. D2, respectively. To both, the observational filter of HIRDLS has been applied. In direct comparison, we see that the general amplitude above strong orography is mostly unchanged by the modification. On the other hand, regions, where GWs propagate to are enhanced by this modification. This is as expected, since the term in Eq. D1 is related to the horizontal dispersion of the GW, which is stronger for refracting and turning waves. GWs that propagate far from their sources are more likely to encounter differing wind conditions and thereby refract or turn. The local approximation of this effect is, however, not a replacement for ray tube techniques describing the GW extent in phase space (e.g. Muraschko et al., 2015).

## Appendix E: Mountain Wave blocking diagrams

Since the blocking diagrams described in Taylor et al. (1993) are not directly applicable to orographic GWs close to their source where $\omega_{\mathrm{gb}} \approx 0$, we show a different type of blocking diagram in this section. The considerations are based on the relation of wave vector to intrinsic frequency and the lower limit for the frequency of GWs:

$$\omega_{\mathrm{intr}} = -kU - lV, \tag{E1}$$

$$\omega_{\mathrm{intr}} \overset{!}{\geq} |f|. \tag{E2}$$

Here $f$ is the Coriolis parameter, $k$ and $l$ are the zonal and meridional wavenumbers, and $U$ and $V$ are the zonal and meridional background wind speeds. Combining both equations allows an estimation where the intrinsic frequency drops below the Coriolis frequency for a given wind speed and direction, which directly corresponds to a critical level for MWs and therefore restricts vertical propagation.

Figure E1 shows the restricted (horizontal) phase space for the same regions and wind profiles as Fig. 9a – d. Panels a and c show the Himalaya and Mongolian Plateau region, respectively, where basically the opposite of the more general blocking diagrams can be seen. While Fig. 9 shows that the Himalaya region has the more restricted phase speed space for non-orographic GWs, for GWs of orographic origin, the Himalaya region shows a more favorable phase space. Only half the phase space is strongly restricted here, while about two-thirds are restricted above the Mongolian Plateau. This strengthens our finding that the northward shift seen in the HIRDLS observations is due to stronger non-orographic GWs in the northern region and MWs refracting to large vertical wavelengths in the southern region.

The Rocky Mountain region in Fig. E1b is highly restricted for mountain waves, as is expected due to the wind reversal, and therefore almost no MWs will reach up to 25 km altitude. Although there is a hint at a wind reversal in the vertical wind profile of the southern Africa region, this is not confirmed by Fig. E1d, where a large part of the GW spectrum is blocked, but by far not the full phase space as in Fig. E1b. The weak surface level winds (c.f Fig. 9f) are, however, very unfavorable conditions to launch and propagate MWs to the stratosphere in the first place.

*Author contributions.* SR and PP conceptualized the study, ME performed HIRDLS processing and analysis, SR conceptualized and developed the MWM and performed the simulations. PP and MR supervised the study and JU acquired the funding. SR wrote the manuscript. All authors provided scientific input and reviewed the manuscript.

*Competing interests.* We, the authors, declare that there are no competing interests present.

*Acknowledgements.* The work of SR was funded by the German Research Foundation (Deutsche Forschungsgemeinschaft, DFG) project UN 311/4-1. In addition, SR and his visit to JB was partly funded by DFG project PR 919/5-1. The work of LK was partly funded by the Federal German Ministry for Education and Research (Bundesministerium für Bildung und Forschung, BMBF) under grant 01LG1907 (project WASCLIM) in the frame of the Role of the Middle Atmosphere in Climate (ROMIC)-program. The work of ME was supported by the German Research Foundation (Deutsche Forschungsgemeinschaft, DFG) project ER 474/4-2 (MS–GWaves/SV), which is part of the DFG research unit FOR 1898 (MS–GWaves). The work by ME was also supported by the Federal German Ministry for Education and Research (Bundesministerium für Bildung und Forschung, BMBF) project QUBICC (grant no. 01LG1905C), which is part of the Role of the Middle Atmosphere in Climate II (ROMIC-II) program of BMBF.

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

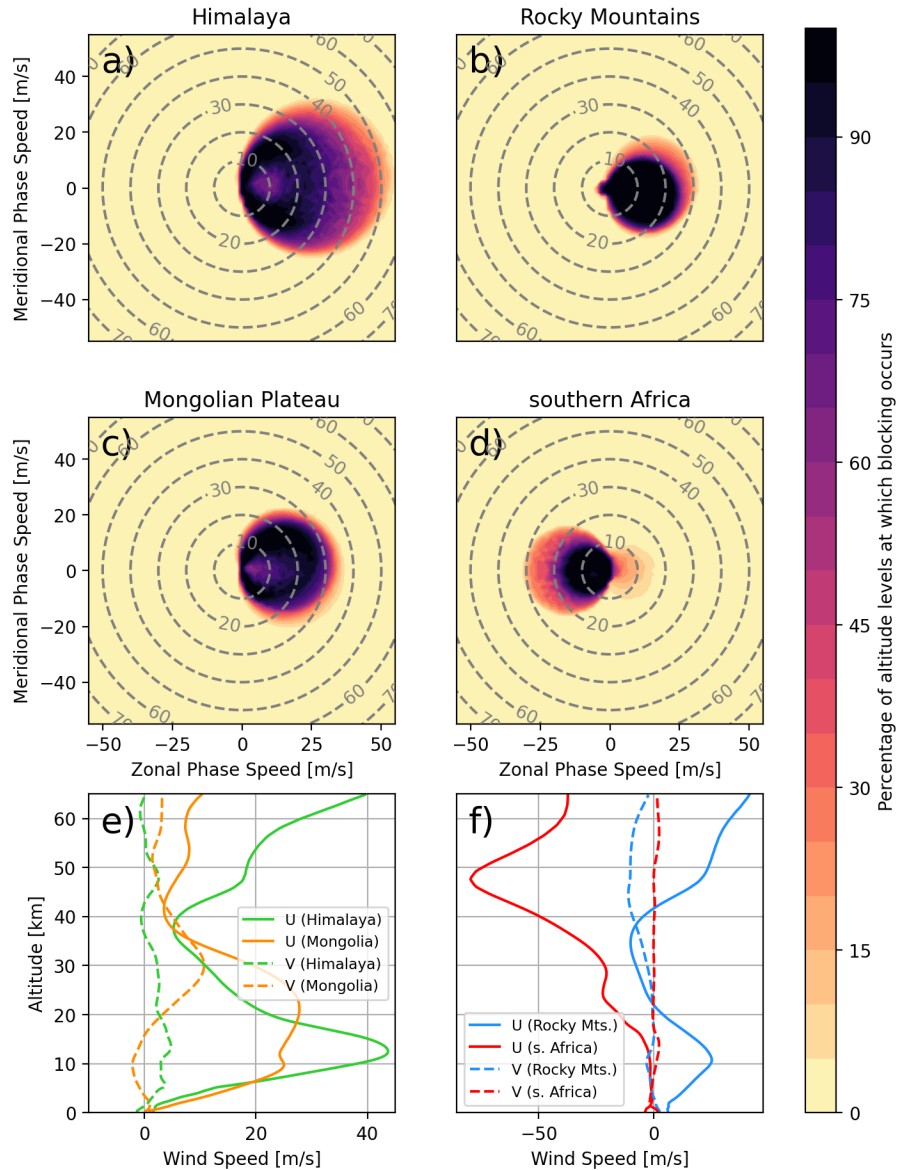

**Figure 9.** Blocking diagrams as introduced in Taylor et al. (1993) for the four regions shown in Fig. 8 for the time period of January 2006: a) Himalaya, b) Brazil, c) Mongolian plateau and d) southern Africa. Color shading gives the fraction of altitude levels between the surface and 25 km that exhibit a critical level for the corresponding part of the GW phase speed spectrum. Alternatively this can be interpreted as an estimate on the probability that a GW of given (ground-based) phase speed passes beyond 25 km without being filtered by a critical level. The mean wind profile of the considered region has been taken for the calculation of blocking occurring at each individual level. Panels e) and f) show the monthly mean vertical profiles of zonal (solid) and meridional (dashed) wind for the Himalaya and Mongolian Plateau regions and Brazil and southern Africa region, respectively (colors correspond to the regions in Fig. 8).

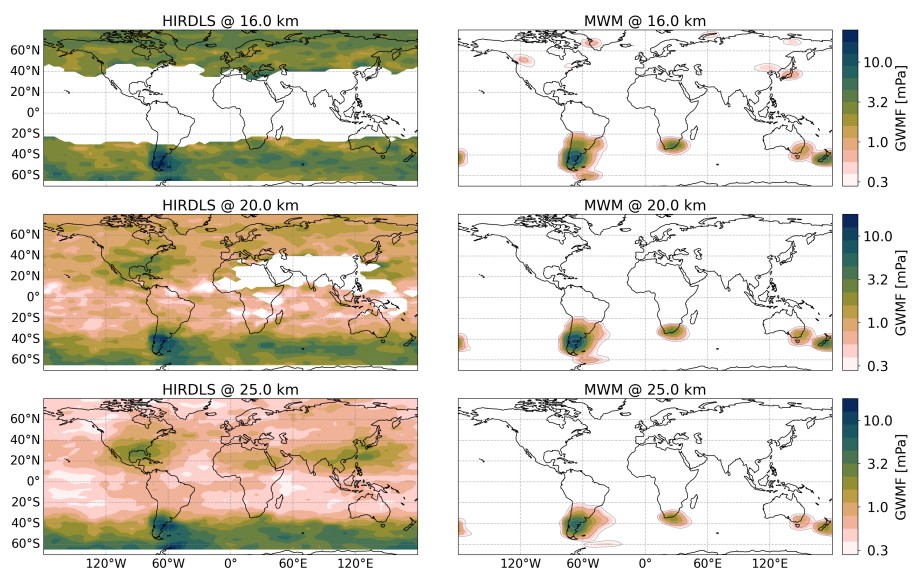

**Figure 10.** Same as Fig. 6 but for July 2006. Note the logarithmic color scales.

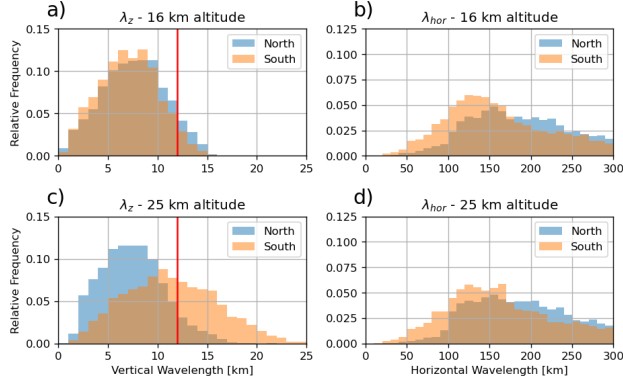

**Figure 11.** Distribution of vertical (left) and horizontal (right) wavelengths as found by the MWM at altitudes of $16\,\mathrm{km}$ (top) and $25\,\mathrm{km}$ (bottom). The northern region corresponds to $37°$-$47°\mathrm{S}$ and the southern region to $47°$-$57°\mathrm{S}$, both between $40°$-$90°\mathrm{W}$. The vertical red line in panels a) and c) marks the cutoff wavelength of $\lambda_z = 12\,\mathrm{km}$ for the present HIRDLS data.

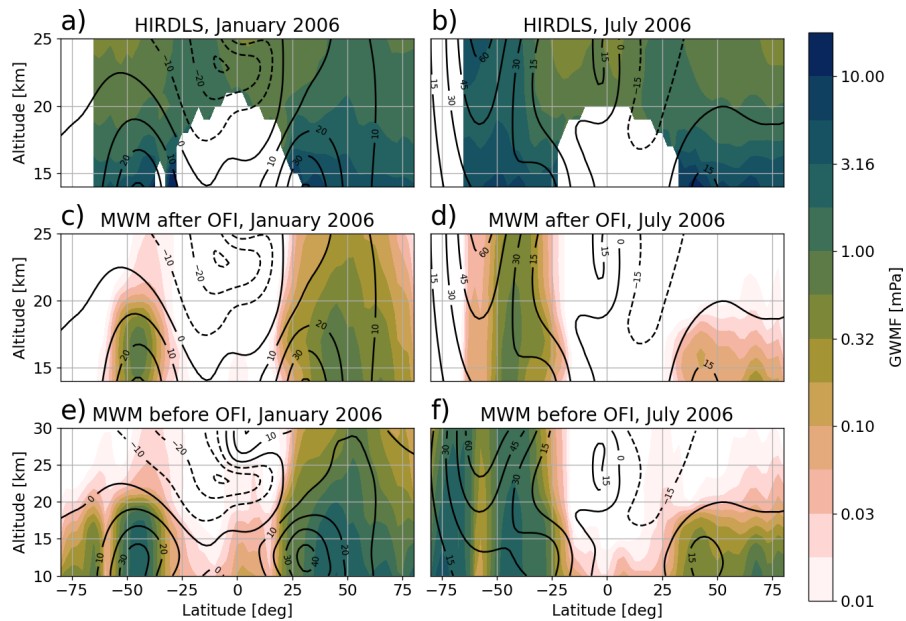

**Figure 12.** Monthly and zonal mean GWMF for January (left column) and July (right column) 2006. Panels a) and b) show HIRDLS data, panels c) and d) MWM data after application of the observational filter and panels e) and f) MWM data without any filtering. Contour lines show the monthly and zonal mean zonal wind for the corresponding month. Note that panels e and f show a wider altitude range than a – d.

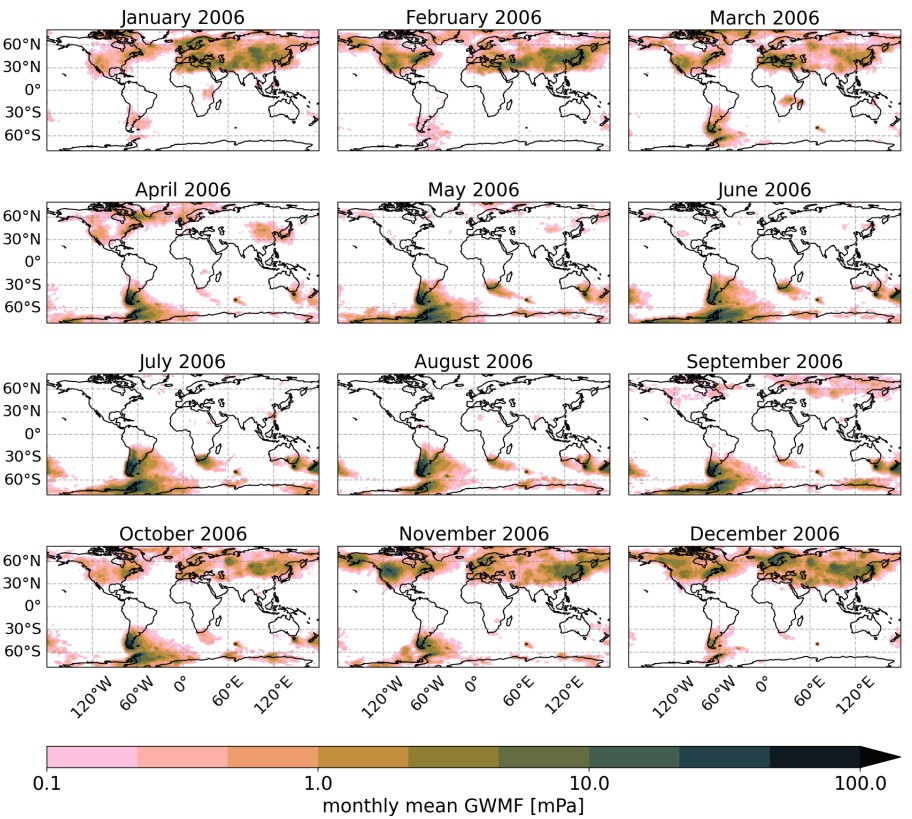

**Figure 13.** Horizontal monthly mean GWMF distribution for each month of 2006 at 25 km altitude. Shown is MWM data without the observational filter applied.

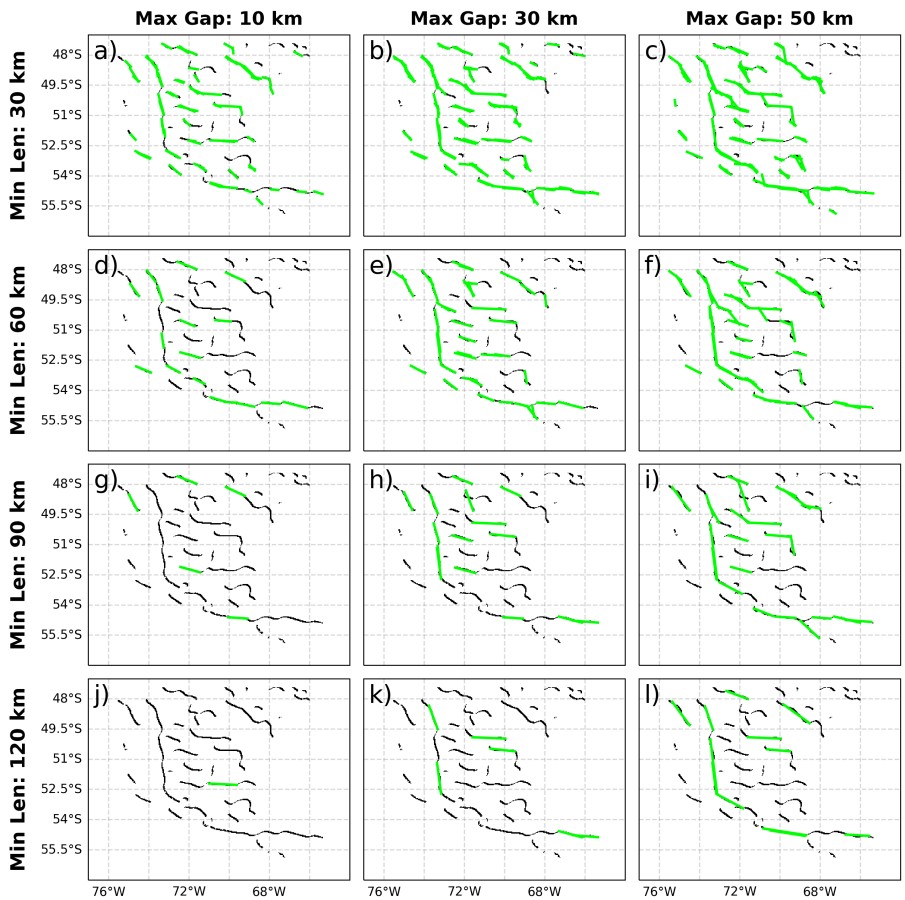

**Figure B1.** Line structure skeleton of the Southern Andes region after applying a bandpass of scales 100-200 km. The detected line segments are shown in green on top of the skeleton. Maximum line gap varies from left to right with values of 10 km, 30 km and 50 km in the given column. Minimum line length varies with row with values of 30 km, 60 km, 90 km and 120 km respectively. Panel e shows the parameters, that have been used throughout this study.

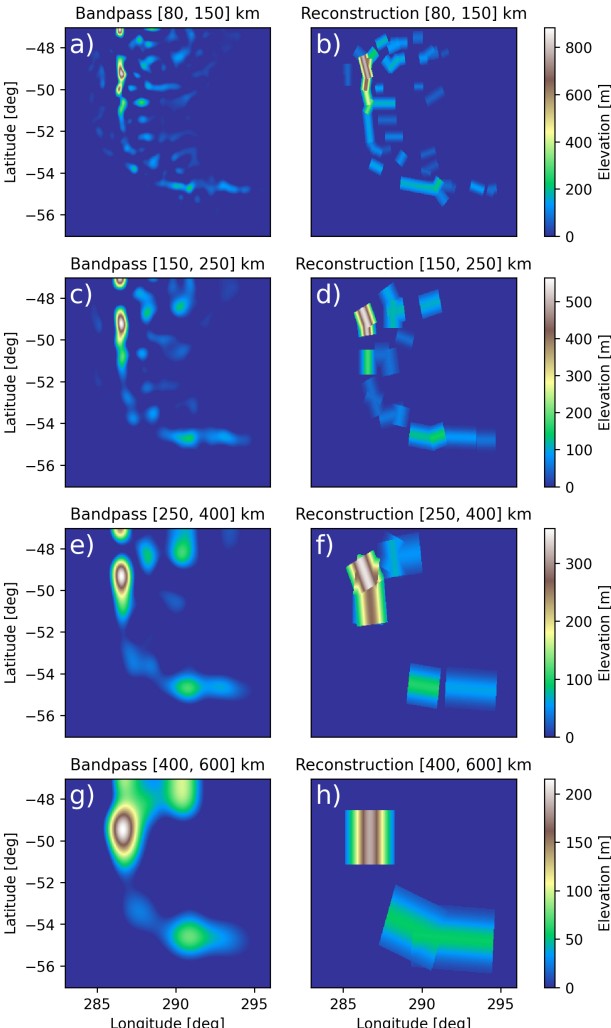

**Figure C1.** Reconstruction from the different detected idealized two-dimensional ridges in the given bandpass filtered elevation data. The left column show the result of the bandpass filter, the right column the corresponding detected ridges. The spectral band varies with row from smallest to largest scales. For the reconstruction, if multiple ridges cover the same spot, the maximum height was taken. Note that only the four smallest scale intervals yielded ridges and are therefore shown here.

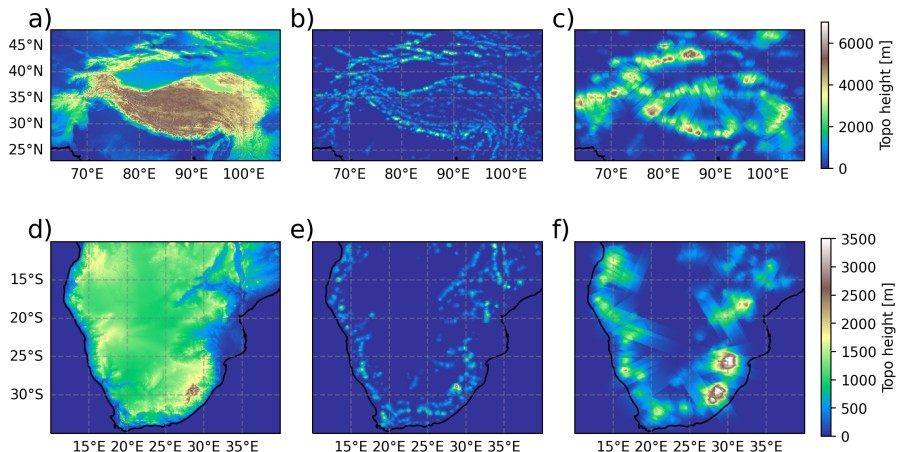

**Figure C2.** Same as Fig. 2 but for the Mongolia region (a – c) and southern Africa (d – f). The underlying topographic data is shown in panels a and d, respectively, while the reconstruction from the identified idealized Gaussian mountain ridges are shown in panels b, c, e and f. The reconstruction is separated into small scales ($\leq$ 150 km, panels b and e) and large scales (>150 km, panels c and f).

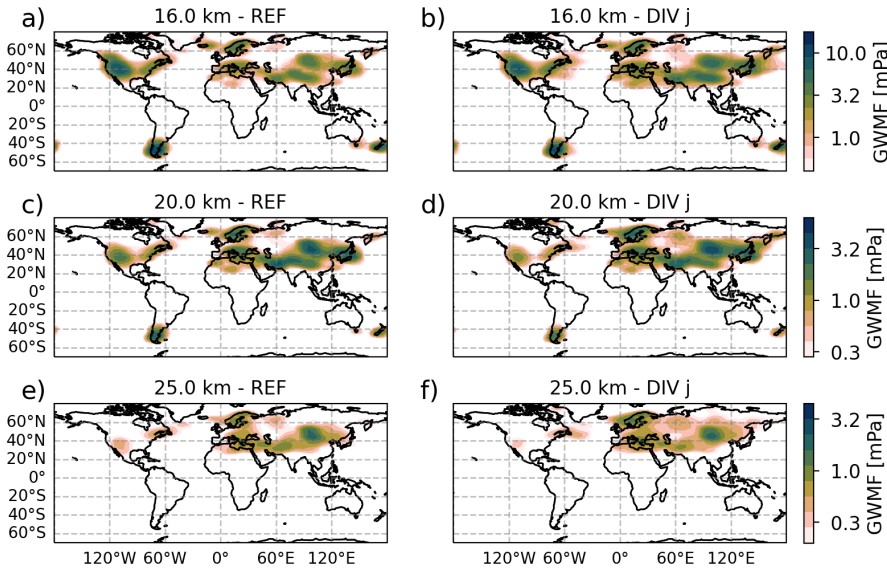

**Figure D1.** Similar to Fig. 6, the monthly mean GWMF prediction for January 2006 from the MWM is shown at different altitudes. The left column uses the standard GROGRAT amplitude correction, the right column a modified version, where the last term in Eq. 6 is approximated locally according to Eq. D1. The observational filter of HIRDLS has been applied to both data sets. Note the logarithmic color scale.

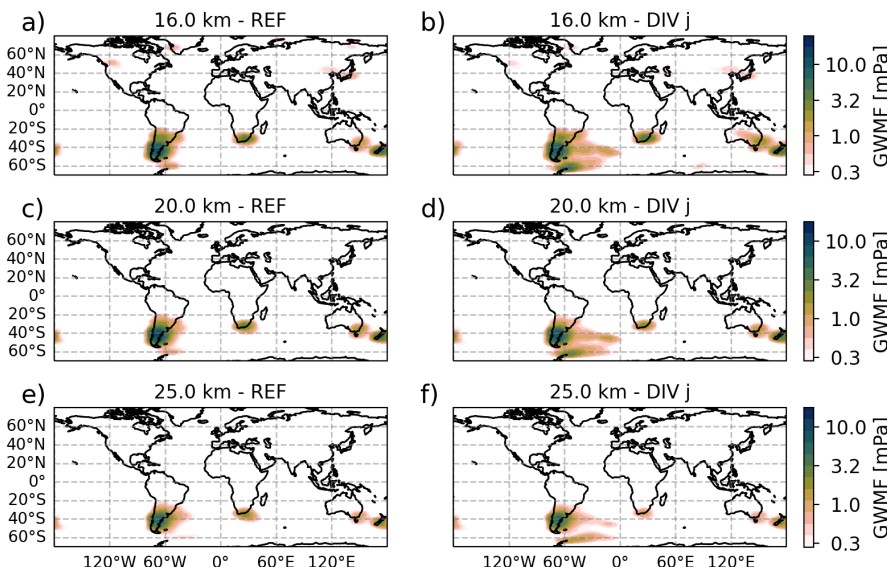

**Figure D2.** Similar to Fig. 10, the monthly mean GWMF prediction for July 2006 from the MWM is shown at different altitudes. The left column uses the standard GROGRAT amplitude correction, the right column a modified version, where the last term in Eq. 6 is approximated locally according to Eq. D1. The observational filter of HIRDLS has been applied to both data sets. Note the logarithmic color scale.

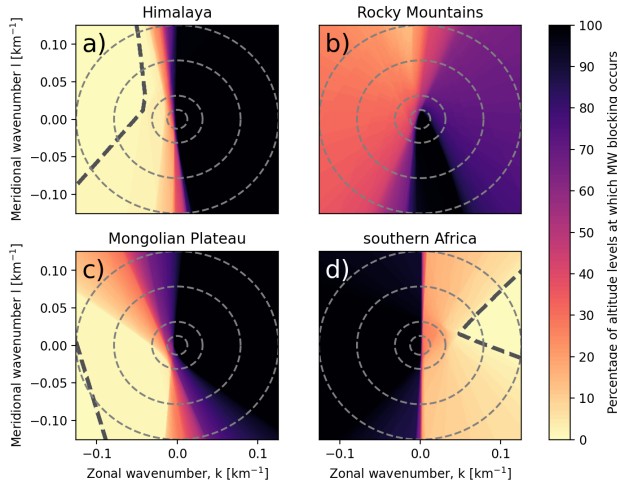

**Figure E1.** Mountain wave blocking diagrams similar to the ones given in Fig. 9 but more applicable for MWs with $\omega_{\mathrm{gb}} \approx 0$. In this case, the critical level filtering is based on the wind profile and Eq. E2. The percentage of altitudes from the surface to 25 km at which an MW of given horizontal wavenumbers is blocked is given in color shading. The dashed line separates the region of (horizontal) phase space that does encounter no critical level at any level (radially outwards w.r.t. the dashed line). Circular gird lines show horizontal wavelengths of 500 km, 200 km, 80 km, and 50 km (from the center outwards). Note that this diagram does not consider refraction, which could lead to MWs maneuvering around critical levels in phase space on their propagation path.