# Peer review of "A mountain ridge model for quantifying oblique mountain wave propagation and distribution"

_EGUsphere, 2022_

## Author Response (AR1)

**1 General Reply**

We, the authors, thank both reviewers again for their thorough and helpful reviews. The comments and general remarks have been used to make the article more clear and accessible as well as more detailed on specifics of the presented model. Responses to all specific comments are given below.

In particular, the main deficiencies as stated by the editor have been addressed:

- A discussion of the limitations of the original GROGRAT including the modifications that have been implemented in our version has been added to the manuscript.

- The general methodology, including boundary conditions and simulation setup is now described in more detail. Specific references have been added for further reading details on used methods.

- The text has been revised for a clearer and better presentation in general. Missing references to figures, typos and grammar mistakes have been fixed.

**2 Response Referee No. 1**

We thank the referee for the thorough review of our article. The comments and suggestions were very helpful in improving the presentation of the model to be both more comprehensive but also more precise in the terminology.

The responses to specific comments are given below, where the original reviewer comments are given in italic and any text given in blue has been added to the text in response to the comment.

**2.1 General comment**

*This paper presents and tests a mountain wave (MW) model that aims to represent the characteristics of orographic gravity waves generated by the Earth's mountain ranges, with a particular emphasis on those that have horizontal propagation, and thus that may exert drag remotely from the regions where they are generated. This aims to address a deficiency in current MW parametrizations which use a single-column approach. The model represents the MW field as the superposition of 2D waves generated by elongated ridges adjusted to optimally fit the main mountain ranges, and a ray tracer algorithm GROGRAT to compute their propagation, and is tested using HIRDLS satellite observations and ECMWF IFS model data. While this is an important and interesting topic, and the science presented in the manuscript appears to be sound, it is difficult to be certain about this, as the presentation is at times unclear and confusing, omitting important details, lacking references, and phrasing explanations in an unnecessarily complicated way. I believe these presentation issues are sufficiently profound to require major revisions.*

All of these points are mentioned in the specific comments below and are therefore addressed in detail at the corresponding location. In general, the clarity in presentation of the methodology has been improved and more detail on the model and analysis is given in the revised version.

**2.2 Specific comments**

1. *Line 9: "This study presents the MWM [mountain wave model] itself". This is not done in sufficient detail. For example, the mathematical expression for the GWMF (gravity wave momentum flux) is not presented anywhere (unlike the expression for the residual temperature, Eq. (8)), and this is an unacceptable omission, given that a large portion of the results are fields of the GWMF.*

   In the revised version the presentation of the MWM details are more fleshed out and presented in a more comprehensive and clear way. This includes the mathematical formulation of the GWMF expression and the assumptions therein as well as the descriptions of the ridge fit, and the Hough transformation in the Appendix.

2. *Lines 20-21: "Various studies also argue for a significant role of gravity waves in the occurrence of Sudden Stratospheric Warming (SSW) events [...] and even their shape". It is not clear what is meant here by "their shape".*

   Their shape refers to whether the SSW event is a split vortex or a displaced vortex event. The text was changed in a way to make this more clear: "..and even their shape, i.e. whether the polar vortex splits or displaces"

3. *Lines 26-27: "small scale GWs caused by the sub-grid-scale orography and convection are approximated by a parametrization scheme". I have strong reservations whether the waves under focus in this study (with horizontal wavelengths above 80 km) can be considered "small scale" and if they need to be parametrized, except in climate models. This should be more clearly emphasized.*

   Following Skamarock [2004], the minimum resolved horizontal scales are about 4 times the grid resolution. Therefore for example the ERA5 [Hersbach et al., 2018] with a resolution of about 0.3° can resolve GWs with scales larger than about 120 km, which is about the spectrum that we consider in our study. However, in GCMs with horizontal resolutions of about $1° - 2°$, the minimum resolved wavelengths are $500 - 1000$ km. GWs that are not resolved in these models are considered 'small' in this study, as they are small scale in comparison to the resolved processes (also planetary waves).

   The text was changed to describe the focus on small scale in terms of GCMs in a better way.

4. *Lines 38: "(Polichtchouk et al., 2018)". The left bracket should be immediately before 2018 instead.*

   The text has been changed accordingly.

5. *Line 47: "In the middle atmosphere they [MWs] can be measured from satellites". In this paper, the terminology "gravity waves" is used indiscriminately for waves affected by rotation, which I would classify as "inertia-gravity waves instead". Only waves with horizontal wavelengths of at most a few 10s of km are purely gravity waves. But in Table 1, the lower boundary of the shortest band-pass interval is 80 km. Waves with this wavelength will typically always be affected by rotation of the Earth, which is reflected in their dispersion relation, Eq. (5).*

   Although the spatial scales of the GWs we are considering here are in the regime of inertia-gravity waves, the spectrum also includes non-hydrostatic gravity waves with $\hat{\omega} \gg f$. So in general, it would be wrong to classify all GWs in this study as inertia-gravity waves.

   A comment on this has been added in Section 3.1.

6. *Lines 91-92: "This data set models the earth's surface, including ocean bathymetry, on an 1 arc-minute resolution". For the reader to get a more intuitive view, express this in km as well.*

   The resolution corresponds to about 1.85 km at the equator. This has been added to the data description.

7. *Line 96: "sampled on a 0.3º x 0.3 º grid. Again, express this in km as well, for greater clarity.*

   The resolution corresponds to about 33 km at the equator. This has been added to the data description.

8. *Line 99: "cutoff zonal wave-number of 18". How many km does this correspond to? Please specify.*

   The cutoff wavelength corresponds to about 2200 km at the equator. The description has been added to the text.

9. *Line 101: "the smoothed background is sampled onto a grid of 2º latitude and 2.5º longitude". Again, express this in km as well, to aid the reader.*

This corresponds to about 220 km and 280 km at the equator, respectively. This has been added to the description.

10. *Line 110: "HIRDLS temperature measurements". This would be a good point to specify the horizontal resolution of these measurements.*

The along-track sampling of HIRDLS is about 80–100 km. This resolution is somewhat reduced by the method of GW estimation by combining adjacent profiles [Ern et al., 2004]. If every pair of profiles is considered, the along-track sampling remains the same - just shifted by half an along-track sampling step. However, for momentum fluxes we cannot use every pair (only about 60% of them) because the GWs in the two profiles do not always match. This means, there will be regions of unchanged along-track sampling, and coarser sampling in other regions because we have to leave gaps.

A note on this has been added to the text: "The horizontal sampling of these measurements is about $80 - 100$ km along track."

11. *Lines 119-120: "For this paper, GWMF is binned within rectangular overlapping bins of $15^o$ in longitude and $5^o$ in latitude sampled every $5^o$ in longitude and $2.5^o$ in latitude". This would be a good point to specify the horizontal resolution of these data, which I believe is higher than $5^o$ or $2.5^o$.*

The original resolution was about 90 km along track, where all orbits of the corresponding months have been used. This information was added in regards to the previous comment. The text does note that the GWMF is binned from the vertical profiles along the orbits: "For this study, GWMF is binned within rectangular overlapping bins of $15°$ in longitude and $5°$ in latitude sampled every $5°$ in longitude and $2.5°$ in latitude from the original profiles given along the satellite orbits."

12. *Lines 130-131: "horizontal wavelength, amplitude, orientation and location". This description suggests that each of these waves (generated by each ridge) are represented as monochromatic waves (as seems to be confirmed later on). If so, this would be a good point to mention it.*

This is a good point! The text now includes the comment, that monochromatic GWs are launched at every ridge.

13. *Line 133: "overlapping slices of $10^o$ in latitude and every $7.5^o$ spanning the full globe in longitude. Does this correspond to the maximum length of each ridge?*

Technically this limits the maximum ridge width in latitude to $10°$, or about 1100 km. However in the topography data considered in this study, no such ridges have been found. Also it is reasonable to describe such long ridges with multiple smaller segments to account for differing wind conditions and therefore propagation patterns along the mountain ridge.

A note on this has been added: "Technically this limits the maximum ridge length to $10°$ in latitude, which in practice, however, never occurred."

14. *Line 149: "The cross section of the idealized Gaussian ridges is given by:". It should be noted that, in reality, a Gaussian ridge would produce waves that, although 2D, are not monochromatic, unlike what seems to be assumed in the MWM.*

This is also a good point. In this study, the GW response of the Gaussian ridge is only approximated by the strongest mode excited by the ridge. A note on this has been added at the translation from Gaussian ridge width to monochromatic wavelength (where $\lambda_{\mathrm{hor}} = 2\pi a$ is assumed).

15. *Lines 157-158: "The amplitude is taken as half the height $h$". Given that a correction is introduced for the effect of low-level flow blocking by Eq. (2), is there a justification for taking $h/2$ as the amplitude instead of $h$? This should be commented on in the text.*

The separation of these correction factors is made because of their different meaning. Taking $h/2$ as the maximum possible amplitude stems from a sinusoidal mountain with valley-to-peak height of $h = 2a$ resulting in a GW with amplitude $a$. This analogy has been transferred

to the Gaussian mountain and in other words limits an air parcel flowing over the mountain from crossing the surface.

The flow dependent correction limits the amplitude even further in the case that the wind conditions are not sufficient to excite the maximum amplitude.

The factor of $\frac{1}{2}$ was an approximation of the amplitude estimation from linear modeling which yields the exact factor of $\frac{1}{\sqrt{2\pi}}$, which is now also stated correspondingly in the text:

"The displacement amplitude is calculated from the best-fitting idealized mountain height $h$ as $\zeta = \frac{h}{\sqrt{2\pi}}$. The factor $\frac{1}{\sqrt{2\pi}}$ stems from linear modeling of a two-dimensional ridge with Gaussian shape [e.g. Nappo, 2012]."

16. *Line 159: "the horizontal wavelength is set to ...". This presupposes monochromatic waves (for each ridge source). This approximation should be mentioned explicitly.*

   See above, this point is now noted in the text

17. *Line 190, Eq. (3): In this equation U_amp appears to be the horizontal velocity perturbation associated with the wave, and should be identified as such in the text. But this is not currently done.*

   The description of $U_{\mathrm{amp}}$ is now part of the section right before the equation.

18. *Lines 195-196: "Lagrangian derivative". Is this defined following the mean flow, or following the total flow (including the wave velocity perturbations? Please mention this.*

   Actually the Lagrangian aspect of the derivative relates to an observer that is following the GW along with its group velocity. To make this more clear, the related sentence was changed to "The derivative $\frac{d}{dt} = \frac{\partial}{\partial t} + c_{g,i}\frac{\partial}{\partial x_i}$, where summation over i is implied, is the Lagrangian derivative for an observer following the GW with its group velocity.."

19. *Line 198: "H the scale height". It is not obvious to the reader that scale height this is. Please briefly specify what it means.*

   This is the atmospheric density scale height, i.e. $H = \frac{RT}{g}$, with temperature $T$, specific gas constant $R$ and the acceleration due to gravity $g$. This has been made clearer in the revised version.

20. *Line 218: "residual temperature structures". It is not at all clear at this point what "residual temperature" means. Later, it becomes clearer that it is the temperature perturbation associated with the waves. But it needs to be explained at this point what it refers to.*

   Good point, this should be made more accessible. The referenced sentence was changed to "..as residual temperature structures, i.e. the temperature perturbations associated with the GWs, and..".

21. *Line 228-229: "phi is the current phase at the ray-path of the wave given by the ray-tracer". How is this determined? It is not clear from Eqs. (4)-(6).*

   The phase is internally integrated along the ray path within the ray tracer. As in any wave, it can be expressed as $\phi = \omega t - k_i x_i$. The integration is necessary to account for variable frequency and wave number along the path.

   An additional sentence was added to clarify: "$\phi$ is the current phase at the ray-path of the wave given by the ray-tracer. This phase is calculated by GROGRAT via integrating $\phi(t, x_i)$ from launch to the current position along the ray path [Marks and Eckermann, 1995]."

22. *Line 229-230: The last term accounts for linear frequency modulation in the vertical with chirp rate ...". What is the physical basis for this? A relevant reference should be cited.*

   The linear chirp is a linear approximation of a vertical wavelength, that changes in height. Since the vertical wavelength could change quite rapidly on encountering e.g. a critical layer, the linear approximation is much better suited for a reconstruction in altitude than stretches with fixed vertical wavelength. The basis of this is found in textbooks [e.g. Nappo, 2012] and articles [e.g. Fritts and Alexander, 2003] alike.

The section was changed to: The last term accounts for a linear approximation of the change in vertical wavelength along the vertical with chirp rate $c_\mathrm{m} = \frac{\Delta_\mathrm{m}}{\Delta_z}$, which is calculated as the finite difference derivative of $m$ for the closest time steps around target altitude $z$. The linear approximation of the dependence of the vertical wavelength on altitude increases the reconstruction performance significantly where it changes rapidly, e.g. below critical layers [e.g. Nappo, 2012].

23. *Line 233: "symmetric Butterworth function". How is this defined? Is it the function involving the 12th power in denominator in Eq. (8)? If so, this should be made explicit.*

    This has been made more explicit in the text: In this direction, the amplitude is scaled with an additional symmetric 6th-order Butterworth function, $\left(1 + (x/S)^{12}\right)^{-1/2}$, with $S = l/2$ as scale for a smoother transition to zero at the edges.

24. *Line 247: "the momentum flux of each wave packet is distributed across the specified data grid using Eq. (8) analogously for GWMF". It is not clear how this analogy works. As mentioned earlier, it is necessary that an expression for the momentum flux is presented, and it is explained where it comes from.*

    This section was rephrased with more information which assumptions have been carried over from the temperature perturbation consideration and which terms have to be modified and why. Although the equation for GWMF is very similar to the one for temperature perturbation, is has been included in the revised version nevertheless. We added an explanation as follows:

    The spatial distribution of GWMF is performed across the specified data grid using the same wave packet assumption as for temperature perturbations in Eq. 8. The maximum GWMF of the wave, $F_\mathrm{max}$, is given by the raytracer [Marks and Eckermann, 1995], but can also be calculated from the temperature amplitude following the relation given by Ern et al. [2004]. They state GWMF $\sim T_\mathrm{amp}^2$ and therefore the GWMF of a wave packet, $F$, has to decay faster than the temperature perturbation (by a factor of 2). The oscillating term, $\cos \phi_\mathrm{tot}$, has to be dropped because $F$ depends only on the amplitude and not the phase. As in Sec. 3.3.1, the edges of the wave packet are smoothed with the same 6th-order Butterworth function. The resulting equation in analogy to Eq. 8 is then given by:

    $$F = F_\mathrm{max} \frac{1}{\sqrt{1 + \left(\frac{2d_\mathrm{perp}}{l}\right)^{12}}} \exp\left(-2\left(\frac{2d_\mathrm{along}}{\lambda_\mathrm{hor}}\right)^2 - 2\left(\frac{2d_z}{\lambda_z}\right)^2\right). \tag{1}$$

25. *Line 248: "since GWMF  T^2". Where is this shown? A backing reference is necessary.*

    This stems mainly from the relation in Ern et al. [2004], where it was shown that GWMF $\propto T_\mathrm{amp}^2$. The relevant reference has been added at this point.

26. *Lines 251-252: "we are supersampling the GWMF of each wave on a finer grid (3x3 subgrid resolution for each grid point)". Is there any particular reason why it is 3 x 3? Please explain.*

    The choice of 3x3 is fine enough for the data we are processing here. This leads to a sampling distance of about 50 km, which is fine enough for the smallest considered wavelengths. A note on this has been added to the text.

27. *Line 257: "the footprint of the grid cells of the horizontal distribution". It is not clear what this means. Please explain in the text.*

    This was referring to the surface area of bottom side of the grid cube, i.e. the horizontal area covered by the corresponding grid cell.

    The referenced sentence has been changed to clarify: "..and $A_{grid}$ is the total horizontal surface area of the corresponding grid cell in the data grid."

28. *Lines 266-267: "reconstruciton" should be "reconstruction" instead.*

    This typo is fixed in the revised version.

29. *Lines 308-309: "The operational analysis data set is provided on a 0.1° resolution and capable of resolving mesoscale gravity waves". In km this is around 10km, I think, and this should be mentioned in the text.*

    This has been clarified by including the sentence: "In the considered region, this corresponds to a horizontal resolution of about 10 km meridionally and 6 km zonally."

30. *Line 313: "smaller scale MWs". Smaller than what? Please specify.*

    Here, smaller was referring to much smaller than the dominating large scale pattern. This would correspond to wavelengths of about 100–150 km and smaller.

    This has been clarified in the text: "..interfering with GWs of significantly shorter wavelengths than the main structure, about 100–150 km and smaller."

31. *Line 316: "spontaneous adjustment". This expression is thrown in here, as if it was obvious what it refers to, when in fact this phenomenon may not be known to a sizeable part of the readership of this paper. Please briefly introduce the concept, possibly backed with a reference.*

    This has been changed to the more common name "geostrophic adjustment" including a brief description and some main references, that explain the process in more detail [Fritts and Alexander, 2003, Williams et al., 2003, de la Camara and Lott, 2015] since further explanation would leave the scope of this research article.

32. *Lines 326-327: "The IFS, however, seemingly does a better job of resolving very small scales". Better than what? Please be more explicit.*

    Here, the better should refer to "better than the described MWM". To make this more explicit, this sentence was changed to: "Features of very short scales (i.e. below about 60 km) are not as well represented by the MWM temperature reconstruction as by the IFS, which is partly due to the lower scale limit set to 80 km."

33. *Line 327: "towards towards". Eliminate this repetition included by mistake.*

    Fixed this mistake in the revised version.

34. *Lines 349-350: "the change in wave field characteristic is not happening due to filtering of westward facing GWs". Filtering by what process? Please be more specific.*

    The filtering at this point refers to the reduction in amplitude and thereby visibility by breaking due to e.g. a critical level due to the change in wind direction. This has been made more specific in the revised manuscript: "is not happening due to filtering, i.e. breaking (e.g. at a critical level) and thereby reduced visibility, of westward facing GWs"

35. *Line 360: "GW filtering due to wind conditions". What is the mechanism that causes this filtering. Please specify it in the text.*

    This wind filtering happens at a critical level, which is dependent on the background wind profile. As this is explained in the section 5.1.1 in more detail, the revised article specifies the process just briefly via: "..GW filtering due to wind conditions that lead to a critical level for MWs."

36. *Figure 5: It should be mentioned in the caption whether the results shown in this figure are for the whole globe, or in some localized region, e.g. the Andes.*

    This is only the Southern Andes region, and the caption in the revised version now reflects that as well.

37. *Line 373: "spontaneous imbalance". Again, add a relevant reference to this topic.*

    There are two main references hinting at jet imbalances as main sources: Wu and Eckermann [2008], Alexander et al. [2016]. Both are now part of the text and the specific term "spontaneous imbalance" was changed to the more general "jet imbalances".

38. *Line 386: "decreases in strength stronger". This does not seem to be correctly phrase. Please rephrase this passage.*

The sentence has been rephrased to "The maximum above the Himalaya dominates at $16\,\mathrm{km}$, but, with increasing altitude, weakens stronger than the one above the Altai mountains."

39. *Line 418: "oblique propagation". In what direction? This is not specified, but it should.*

This has been specified to: "oblique propagation towards the east of both sources"

40. *Line 433: "the criterion of waves blocking". Does this refer to wave absorption by critical levels? If so, it would perhaps be better to include standard terminology. Usually, blocking is used in the context of low-level blocking, as described by Eq. (2). The process I think the authors are referring to is attenuation or suppression of gravity waves by critical levels, and perhaps it should be named so.*

This section was completely rephrased as to use standard terminology. While the term "blocking diagram" is commonly used to show the excluded part of the phase speed spectrum, all other references to this kind of blocking have been changed to refer to critical level filtering instead. The text was changed to:

"The curve of $\omega_{\mathrm{intr}} = 0$ is a circle in phase speed diagrams with center at $(\frac{U}{2}, \frac{V}{2})$ and radius $R = \frac{1}{2}\sqrt{U^2 + V^2}$ for zonal and meridional background winds $U$ and $V$ and covers the restricted, i.e. blocked or filtered, part of the phase speed spectrum. These curves can be superposed on a phase speed diagram for all altitudes from the surface up to $25\,\mathrm{km}$ altitude to give a measure of how strong or widespread across altitudes the critical levels are for GWs launched at the ground. This is done in Fig. 9a – d, where the color shading gives the percentage of altitude levels, that exhibit critical levels for GWs of the corresponding (ground based) phase speed. In other words, the color shading gives an estimate of the probability for GWs with given phase speed to be filtered by a critical level below $25\,\mathrm{km}$. Note, however, that these diagrams are only an indication of critical levels for MWs with $\omega_{\mathrm{gb}} \approx 0$, which are encountered wherever the horizontal wind projected onto the horizontal wave-vector becomes zero. As an additional metric, the monthly mean vertical profiles of horizontal wind for the four regions are shown in Fig. 9e and f."

41. *Lines 440-441: "These contours have been superposed from the surface to 25km altitude and divided by the number of contours". This is phrased in a rather confusing way. It is unclear what contours the authors are talking about. Please clarify.*

The section has been reworked for a better explanation of the blocking diagram and of what is shown in Fig. 9. By this rework, also the next comment is addressed. The rewritten part is:

"The curve of $\omega_{\mathrm{intr}} = 0$ is a circle in phase speed diagrams with center at $(\frac{U}{2}, \frac{V}{2})$ and radius $R = \frac{1}{2}\sqrt{U^2 + V^2}$ for zonal and meridional background winds $U$ and $V$ and cover the restricted, i.e. blocked, part of the phase speed spectrum. These curves can be superposed on a phase speed diagram for all altitudes from the surface up to $25\,\mathrm{km}$ altitude to give a measure of how strong or widespread across the levels the blocking layer is for a GW launched at the ground. This is done in Fig. 9a – d, where the color shading gives the percentage of altitude levels, that are blocking GWs of the corresponding (ground based) phase speed. In other words, the color shading gives an estimate of the probability GWs with given phase speed to be blocked below $25\,\mathrm{km}$."

42. *Line 442: "the percentage of altitude". Again, this concept is unclear, and the authors need to be more explicit in explaining it.*

See comment above.

43. *Lines 482-483: "The MWM shows, that parts of the GW spectrum refract to very short vertical wavelengths, which makes them hard to be detect by the satellite". How can this process be distinguished from wave breaking, it that is possible?*

Within GROGRAT, this is distinguishable by the saturated amplitude of each ray in combination with the vertical wavelength. We have seen a reduction in vertical wavelength, but no significant reduction in saturated amplitude. If these GWs were breaking, their amplitude would be strongly suppressed. Of course a smaller vertical wavelengths leads to a tendency to saturate earlier and thereby (partial) wave breaking. This is however not seen in the data we have considered here.

A note on this has been added to the section: "The refraction to smaller vertical wavelengths is distinguished from GW breaking by the GW amplitudes as calculated within GROGRAT, which, in general, do not reach saturation although the vertical wavelengths shorten significantly."

44. *Line 486: "total breakdown of GWs reaching saturation". In what sense is this process different from what the authors call "blocking" (around Eqs. (10)-(11))?*

While blocking, or a critical level, will lead to an ever shrinking allowed saturated amplitude of the GW approaching it (which is therefore breaking), the other direction is not true. Due to the density decrease with height, the amplitude of any GW is increasing $\propto \exp \frac{z}{2H}$, with $H$ being the atmospheric density height scale, and therefore any GW will reach saturation at some point in the atmosphere. This, however, does not have to be at a critical level. GROGRAT handles this in a way that the GW propagates further with a reduced amplitude, while it might also be possible, that hitting such a saturation level could lead to the GW breaking down (without the critical level).

Such a possibility was discussed by Kaifler et al. [2015] before. This reference has been added to the corresponding section. The difference might be clearer due to the revision of the blocking section.

45. *Line 490: "obsrvations" should be instead "observations".*

This mistake has been fixed in the revised version.

46. *Caption of Fig. 9, line 2: "percentage of altitude levels". It is unclear what this means.*

The caption has been changed to a more descriptive text that should be more clear on what is shown: "Color shading gives the fraction of altitude levels between the surface and 25 km that block the corresponding part of the GW phase speed spectrum. Alternatively this can be interpreted as an estimate on the probability that a GW of given (ground based) phase speed passes beyond 25 km without encountering a critical level. The mean wind profile of the considered region has been taken for the calculation of blocking occurring at each individual level."

47. *Line 517: "far to little" should be instead "far too little".*

This mistake has been fixed in the revised version.

48. *Line 518: "we can attribute this feature in the observations to MWs due to katabatic flow". Can the authors explain in more detail why these MWs are not represented in the MW model?*

Although GWs excited by katabatic flow are also categorized as MWs, they are not considered by the MWM since they are excited at drops in elevation, which are not detected by the presented algorithm. The algorithm is capable of detecting idealized ridges, which do not exert GWs due to katabatic flow. A note on this has been added to the referred section.

49. *Line 519: "like spontaneous imbalances of the polar jet or ". The end of this sentence is incomplete.*

"..or frontal systems." has been added to end the sentence correctly.

50. *Lines 534-535: "Weak GW activity above the northern Rocky Mountains, Greenland and the Japanese Sea can be assigned to structurally agreeing counterparts in HIRDLS". It is not clear what "structurally agreeing counterparts" means. Can this be expressed more simply?*

Due to the comment of another reviewer, this section has been removed, since it does not provide any valuable comparison.

51. Line 547: "due to clouds and the tropopause". It is unclear in what way the tropopause will produce these gaps. Can the authors explain this in more detail?

This was not clearly enough in the original text. Data below the tropopause was not used in the analysis of GWMF estimation since it represents a discontinuity in stability and could therefore lead to wrong association estimation of GWs with the used methods. The text reflects this by the additional sentence: "While clouds restrict the line of sight of the instrument, data below the tropopause has not been used in the diagnosis of GWMF due to the discontinuity in stability it represents, which could lead to wrongly estimated GW parameters."

52. *Line 560: "due to shift of MW parameters towards better observable values". it is not clear what this means. Does it refer to longer vertical wavelengths? In any case, please be more explicit.*

This section was rephrased to "due to propagation, but due to the shift of MW parameters towards values that are better observable by the instrument, i.e. towards longer vertical wavelengths, and therefore less suppressed after filtering." to be more specific.

53. *Lines 569-570: "due to the observational". A word seems to be missing after the word "observational". Please check and correct.*

This was referring to the observational filter. The word "filter" has been added accordingly.

54. *Lines 591-592: "As presented, model allows" should be instead "As presented, the model allows".*

This has been changed in the revised version.

55. *Line 620: "The distributions agree to the findings" should be instead "The distributions agree with the findings".*

This has been changed in the revised version.

56. *Line 700: "the corresponding sigma of the kernels". Not clear at all what this is. What does sigma represent. Add this information.*

$\sigma$ is the width of the Gaussian. This information has been added as follows: "the corresponding $\sigma$, i.e. the width (or standard deviation) of the Gaussian,"

57. *Line 707: "rectangular cutout". Not clear what "cutout" means in this context. The authors need to provide more details.*

This section has been complemented by a few sentences to make clear what the cutout is:

"The least squares fit is performed on a rectangular cutout of the bandpass filtered topography, $H_{clip}$. This cutout is a cropped part of the topography data centered around the identified ridge candidate. The crop is performed in a way such that the ridge candidate is oriented horizontally in the center. The length of this cutout is given by the line length of the Hough transformation, or in other words the length of the ridge candidate, the width.."

58. *Section A2 of Appendix A is, as a whole, quite difficult to follow. Some additional information would help.*

This section A2 has received a major overhaul in the new revision and is hopefully both, easier to follow and more precise in the presentation.

59. *Line 718: "it's probabilistic variant" should be instead "its probabilistic variant".*

This has been changed accordingly in the revised version.

60. *Line 721: "Radon transformation". It is not obvious to the reader what this is. Please add a reference that explains it.*

The text now refers to one of many possible reference regarding the Radon transformation [Herman, 2009]. Also the text has been a bit more fleshed out to give more immediate context.

61. *Line 725: "empty accumulator". Again, it is not clear at all what this refers to. What is an accumulator? Why is it empty?*

The accumulator is a two-dimensional counting matrix, that is initialized with all zero entries spanning the full Hough space, i.e. the space of all possible lines in a given data set. Each non-zero entry in the data, where the lines shall be detected adds 1 to the accumulator entries, that correspond to all possible lines through that entry. The matrix therefore accumulates "votes" of the initial data set.

This has been made more specific and clear in the revised version.

62. *Lines 725-726: "this matrix". It is not clear what "matrix" refers to? The accumulator? If so, please mention before that the accumulator is a matrix.*

This was indeed misleading and is now changed in a way that the accumulator is introduced as a matrix with all zero entries.

63. *Lines 728-729: "(that pixel in the input image basically gives a 'vote' for all straight lines passing through it)". This explanation is quite unclear. It is unclear what "pixel" the authors are talking about, and what this "vote" signifies. Please clarify.*

Pixel was referring to the data in the initial two-dimensional data set, which oftentimes is an image. Since this switch of context is quite abrupt, the revised section sticks to "data set entries". The voting concept is quite intuitive, if it is presented properly. The revised version should do better at introducing how each non-zero entry "votes" for all lines passing through it as valid lines.

64. *Section B1 of Appendix B as a whole is also quite hard to understand. Please not only improve the description, but also add relevant references that may aid the reader.*

This section was completely reworked in order to improve the depiction of the algorithm involved in the Hough transformation. The overhauled section gives more context and a less confusing description of involved parameters and matrices. There are plenty of resources on the Hough transformation, however the earliest work by Duda and Hart [1972] is referenced in the text for more details.

65. *Lines 766-767: "All authors provided scientific input and reviewed the manuscript". However, it is not mentioned who wrote the manuscript (presumably the lead author). Please add that information.*

Information on the writer has been added: "..funding. SR wrote the manuscript. All authors.."

**3 Response Referee No. 2**

We thank the referee for the detailed review of our article. The comments and suggestions have been used to improve both, the presentation and the detail, in which the model is described.

The responses to specific comments are given below, where the original reviewer comments are given in italic and any text given in blue has been added to the text in response to the comment.

**3.1 General comments**

*The present manuscript "A mountain ridge model for quantifying oblique mountain wave propagation and distribution " submitted by Sebastian Rhode and coauthors presents good research on the constraining observation of internal gravity waves through phase space modelling of said waves. I think it will be a valuable contribution to the community but needs a major revision before it can be recommended for publication.*

*In general I have the following criticism:*

*(1) The underlying model GROGRAT has known errors which can have significant impact but can be easily mended. If the coauthors use - as stated in the text - the original version from 1997*

*(Eckermann and Marks) they would need to revise several structural dficiencies before applying the ray-tracing algorithm.*

*(2) A lot of details are missing in the description of the methodologies which hinders the understanding fo some crucial parts of the work. As an example it is unclear in how far the time dependency of the mean flow is taken into account and how it propagates into the ray paths. That makes the interpretation of the presented yearly cycle difficult.*

*(3) Many statements in the text need sources to be backed up. For instance the ridge detection model utilizes the probabilistic Hough transform which is very briefly described but lacks referencing. Without any source and only a brief explanation - as in the scope of this manuscript - the work is not reprodicible to the reader.*

*(4) The text needs a general revision. Several references to figures are wrong, there are many typos and slips of the pen.*

*In the following I will list my line-by-line comments. Note that I omit many of the small typos and gramar mistakes as they are numerous and would make the review longer than necessary.*

**3.2  Specific Comments**

1. *Abstract*

   *- general - The abstract lacks the general purpose of the work. It seems that would be the analysis of wave propagation / transiaent non-linear waves to better understand gravity wave observations. The comparisons are extensively done in the manuscript but are not mentioned here. In return, the deiciencies of the GCMs that arementioned but not adressed may be removed (e.g. cold pole bias, polar vortex breakdown / final warming date).*

   This is a good point. Although the present study is not aimed at improving any deficiencies in GCMs, one of the motivations is to use the model in future studies on these. The main purpose of this work is the presentation of the MWM as a tool for the analysis of MW activity and the capability to support the understanding of observations. The investigation of non-linear waves, however, is not the purpose of this work, due to the limitations of the ray-tracing setup.

   The abstract has been modified to express the main purposes of this specific work in a better way:

   Following the current understanding of gravity waves (GWs) and especially mountain waves (MWs), they have a high potential for horizontal propagation from their source. This horizontal propagation and therefore the transport of energy is usually not well represented in MW parameterizations of numerical weather prediction and general circulation models. In this study, we present a mountain wave model (MWM) for the quantification of horizontal propagation of orographic gravity waves. This model determines MW source location from topography data and estimates MW parameters from a fit of idealized Gaussian-shaped mountains to the elevation. Propagation and refraction of these MWs in the atmosphere are modeled using the ray tracer GROGRAT. Ray-tracing each MW individually allows for an estimation of momentum transport due to both vertical and horizontal propagation. The MWM is a capable tool for the analysis of MW propagation and global MW activity and can support the understanding of observations and improvement of MW parameterizations in GCMs. This study presents the model itself and gives validations of MW induced temperature perturbations to ECMWF IFS numerical weather prediction data and estimations of GW momentum flux (GWMF) compared to HIRDLS satellite observations. The MWM is capable of reproducing the general features and amplitudes of both of these data sets and, in addition, is used to explain some observational features by investigating MW parameters along their trajectories.

2. *- L6f - "This model [...]" i do not understand this sentence. What exactly is associated to what and why?*

   This comment has been addressed along the previous comment. The phrasing has improved to clarify what was meant here.

3. *Section 1 - Introduction*

   *- general - While the first paragraph mentions the roe of internal gravity waves in the atmosphere in gerneral, the second and thir paragraphs focus on the deficiencies of current GCMs. I the following two paragraphs it is explained what the role of orographic waves is and what the presented model is capable of, but not why. Thus the impression is that the presetn study tries to mend deficientcies oin GCMs which is, however neither the scope of the manuscripot nor it is easily transferred to being an SSO parametrization. It would be a lot better to focus less on GCMs but rather focus on the interpretability of observations with respect to detailed wave parameters and sources.*

   The Introduction has received a rework, that sets the focus more on the observation and modeling part and less extensive on the GCM description. Although the final aim was to improve upon MW parameterizations in GCMs, this is subject to a subsequent study and should therefore be touched there.

4. *- L15gf - "ehy are [...] clouds." These statements need referencing.*

   The connection of GWs to large scale dynamic processes is found in Holton [e.g. 1983], Andrews et al. [e.g. 1987] and the connection to clouds for example in Thayer et al. [2003], Saha et al. [2020]. These references have been added to the text.

5. *- L16ff - "Since they propagate [...] and lower thermosphere" This sentence suggests that the waves are propagating dominantly in the vertical. However, the opposite is shown in the manuscipt.*

   The text has been changed to "Since they propagate through the atmosphere , both vertically and horizontally, they transport momentum.."

6. *- L26 - "are approximated" -¿ "are typically approximated"*

   The text has been changed accordingly.

7. *- L42ff - "Both processes could [...] models." This sentence is redundant to the previous one and should therefore be removed.*

   This sentence has been removed from the text.

8. *- L51f - "In order to understand this low stratóspheric GW activity [...]" It is unclear what is meant with this sentence as no wave activity in the Stratopshere was discussed before. At the same point the combination of observations with model studies seems to be the main point of the work. So this should be expanded / explained in more detail.*

   This section has been expanded in the revised version. The revised section reads as follows:

   "In order to understand this comparatively low stratospheric GW activity, observations might be aided by model studies focusing on specific types of GWs. A model describing orographic GW propagation from source to breaking, for example, could shed light into the orographic part of the measured GW spectrum and help disentangle it from other sources. The combination of model and observation data enriches the analysis by providing more data to base the conclusions on but does also provide an opportunity to probe the underlying theory in a real-world application."

9. *- L64 - "wind blocking" The term blocking is not used consistent throughout the manuscript. In the context of orographic (Lee) waves blocking is typically referring to blocked flow over the mountains. However, here it is used as "wind blocking" referring to both the flow blocking at the surface and critical layer filtering of waves due to wind shear. I suggest to either consistently distinguish between flow bnd wave blocking or, even better, use critical layer filtering instead of "wave blocking" as used by Taylor et al. (1993).*

   This has been made more consistent in the revised version. The references to "wind blocking" when actually critical level filtering was referred to, are changed. Only "blocking diagram" as used in Taylor et al. [1993] is still used when talking about the specific figures.

10. *- L79ff - "Predicted GW parameters [...]" I do not understand this sentence.*

The sentence was referring to the consideration of changing wave parameters with altitude as calculated by the ray tracer and the effect on visibility of the instrument. The corresponding sentence has been changed to: "Calculated GW parameters along the ray paths and their change with altitude and critical level filtering are considered as possible causes of some observational features."

11. *Section 2 - Data*

*- L84ff - The formulation of this sentence is unclear. I am guessing that the authors want to describe at which points they require data for their consideration? Also there is at least some detail missing on what is meant by data for "atmospheric winds and temperature(s)". From the latter subsections one may later find out that the authors are referring to the ERA 5 reanalysis.*

The sentence has been rephrased in a more clear way and detail about which specific data sets are used has been added:

"This study uses the ETOPO1 topography data [Amante and Eakins, 2009] for the ridge finding as well as atmospheric background winds and temperature from ERA5 reanalysis [Hersbach et al., 2020] for MW propagation modeling via ray tracing."

12. *Section 2.2 - Atmospheric Background*

*- The authors describe how they use the the ERA5 reanalysis data as background field for the ray tracing algorithm. Hower, several information is missing:*

*(1) What is the temporal resolution? How does the spatially filtered data change over time? Are there temporal filters or interpolation?*

For global scale experiments, 6-hourly data was used, while for the smaller scale Southern Andes case studies hourly snapshots were used. Within GROGRAT all background fields are interpolated using a 4-dimensional spline [Marks and Eckermann, 1995]. The text reflects this now as follows:

"For global ray tracing experiments, 6 hourly snapshots, and for the specific case study of the Southern Andes in Sec. 4.2, hourly snapshots have been used. To ensure smooth transitions in between, GROGRAT uses a 4-dimensional spline interpolation."

13. *(2) The ERA5 dataset is also used for the detrending of the HIRDLS dataset (as explained below). However the spatial filters are different as compared for the ray tracing. There are two questions arising: In how far are perturbation solutions of the MWM and the perturbations from HIRDLES independent of each other when the underlying data is a differently filtered version of the EAR5 data? What is the effect of the different filers? Why are they not the same?*

We apply different filtering of the ERA5 data for obtaining an atmospheric background because the requirements for GROGRAT ray tracing and HIRDLS background removal are very different.

(a) background for ray tracing: Different from previous studies based on high resolution model data, we do not extract the GW signal from ERA5 fields. For the MWM ERA5 fields are just used as an atmospheric background. Potential small remnants of GWs will not have significant effect on the ray traces performed.

(b) HIRDLS background removal: ERA5 is also used as a background for HIRDLS, because the satellite measurement geometry can only resolve zonal wavenumbers up to about 7. At low altitudes, however, Rossby waves in the UTLS zonal wind jets can have relatively high zonal wavenumbers. These are hard to remove from the HIRDLS observations alone and would bias any GW signal extracted from HIRDLS observations. For this reason, we apply an altitude dependent zonal wavenumber cutoff to ERA5 in order to obtain a temperature background containing higher zonal wavenumbers that can be removed from the HIRDLS altitude profiles. Maximum zonal wavenumber at low altitudes is 20 and is reduced to 6 at 20km. This means that the ERA5 background used for HIRDLS will not contain strong

contributions due to GWs. Furthermore, GW amplitudes are strongly underestimated by ERA5, and GWs seen by HIRDLS will usually have much larger amplitudes. Therefore we do not expect the HIRDLS GW signal to be "contaminated" by GWs resolved in ERA5. ERA5, however, is not "perfect". Therefore the standard HIRDLS background removal in, e.g., Ern et al. [2018], which is applied in addition to the ERA5-based background removal, should perform better at 20km and above where atmospheric global-scale waves rarely have zonal wavenumbers exceeding 6. This is why the zonal wavenumber cutoff for ERA5 is gradually reduced to 6 at 20km altitude.

14. *(3) The authors are using IFS data from the operational forcast using a similar kind of filter as for the ERA5 data used for the ray tracing. This dataset is not mentioned here and the impact of the different resolutions in combination with the filtering remains unclear. It would be preferred to explain the differences to clarify the later comparisons and give them a better context / understanding.*

A brief description of the IFS data is now given in this section. The higher resolution of $0.1°$ allows for GWs of small scales down to about $40\,km$ to be resolved within the model [Skamarock, 2004]. The filtering does not change and therefore the large scale background exhibits the same scales as the ERA5 data (in particular, the meridional Savitzky-Golay filter width has been kept at about $9°$, which corresponds to a window of 91 points). The additional section reads:

"In addition to the ERA5 reanalysis data, single snapshots of operational forecast data of the ECMWF integrated forecast system (IFS) is used for validation of MW temperature perturbations in Sec. 4.2. This data set is given on a higher resolution of $0.1°$, or about $10\,km$, in the horizontal and allows therefore resolution of GWs down to about $40\,km$ [Skamarock, 2004]. The scale separation is performed analogously to the ERA5 data described above (the number of points in the meridional filter has been increased to 91 to achieve the same filter width of $\sim 9°$)."

15. *Section 2.3 HIRDLS sat data*

   *- L119ff - The authorts explain that the data is horizontally filtered through overlapping but fixed size bins in latitude and longitude. These bins are therefore not equal in physical size and thus a spatially dependend but strong autocorrelation is introduced into the dataset. What is the impact of the latter?*

   Indeed, spatially dependent autocorrelations will be introduced by the binning in lat-lon. These will result in a smearing-out of the global distribution. For our study, however, this plays only a minor role because for the comparison between MWM and HIRDLS observations the same spatial binning is used.

16. *Section 3.1 RIDGE Identification*

   *- L133f - I do not understand the projection of the orographic data. Are these slices overlapping as well? Or is there an approximation at the boundary between the slices?*

   These slices are overlapping by $2.5°$ in latitude. To clarify this in the text, the corresponding sentence has been changed to:

   "First, the elevation data is divided into overlapping slices of $10°$ width in latitude spanning the full globe in longitude. These slices are generated every $7.5°$ in latitude, leading to a $2.5°$ overlap, and interpolated onto an equidistant grid in terms of physical distance."

17. *- L135f - It remains unclear how the data is filtered. is the Gaussian filter 2-dimensional? Or is the filter applied successively in several directions? How is dioes the filter depend on the projection?*

   The following sentence has been added to the text to clarify: "This is calculated as the difference between the elevation data convoluted with a 2-dimensional Gaussian function of different scales which are given as the limits of the considered scale interval."

18. *- L143 - The Hough transform urgently needs a reference to a source or a more detailed explanation in the appendix as it is not sufficiently explained there. Otherwise the transform remains somewhat a blackbox to the reader.*

    The explanation in the appendix has been expanded for more self-consistency. In addition the first application of the Hough transformation by Duda and Hart [1972] and an improvement made by Kang et al. [1991] have been added as a reference.

19. *- Fig. 1d - This image seems to be very different from the data shown in the appendix (compare Fig. B1), why is that? After studying the appendix it beacomes clear that the used parameters for the Hough transform are not shown in the appendix but also the point data does not seem to coincide in structure. Maybe this is a matter of visual representation?*

    The difference between the skeletons has multiple reasons. For one, the scale intervals of both are chosen differently, in Fig. 1 it is 80–150 km and in Fig. B1 it is 100–250 km. The other reason is the orientation for which the skeleton has been generated. As discussed in the text, for each part of th elevation, four different skeletons are generated. The former is given for direction South-West to North-East and the latter for West to East. A note on the considered scale and direction has been added in the caption of Fig. 1 to clarify.

20. *- L154ff - The meaning of the position and lengths (X,Y,L) as well as the angle (Θ) are unclear because the projection is not clearly described before. Does the algorithm work with a local Cartesian space? If so what si the effect of interpolations?*

    The algorithm is applied to the elevation interpolated to an equidistant grid. The position in longitude and latitude are exact and the angle $\theta$ is estimated in the local Cartesian coordinate system (which is also used within GROGRAT). The interpolation could, in principle, lead to length (and widths) being under/over-estimated in the equator/pole-ward region of each latitude slice. In testing, however, this did not lead to drastic changes or uncertainties and is thus reasonable for the pursued approach of this study.

    The corresponding section was expanded to: "As a result of the combined ridge-finding algorithm we obtain a set of ridges with the following parameters: ridge length $L$ and local Cartesian ridge coordinates $X$ and $Y$ (representing ridge location in zonal and meridional directions, respectively), between $X$ axis and the ridge $\theta$, best-fit width $a$, and best-fit ridge height $h$."

21. *- L158 - The displacement being half the mountain height seems arbitrary here, I am guessig it is derived using linear theory? Please clarify.*

    The factor of $\frac{1}{2}$ was an approximation of the displacement amplitude as excited by a Gaussian mountain as found from linear theory. The actual value is $\frac{1}{\sqrt{2\pi}}$, which is now described in the text. The following sentence has been added to the text:

    "The displacement amplitude is calculated from the best-fitting idealized mountain height $h$ as $\zeta = \frac{h}{\sqrt{2\pi}}$. The factor $\frac{1}{\sqrt{2\pi}}$ stems from linear modeling of a two-dimensional ridge with Gaussian shape [e.g. Nappo, 2012]."

22. *Section 3.3 Ray tracer*

    *- general - Do the authors use the GROGRAT ray tracer as published in Marks and Eckermann (1995) and Eckermann and Marks (1997)? It is well known that this model has several severe shortcomings. Explicitely these are*

    This is a good point. The used version of GROGRAT is not the original version, but has been changed to accommodate the spherical geometry as derived by Hasha et al. [2008]. Therefore the following sentence has been added to the text:

    "Here, we use a modified version of GROGRAT that accounts for the spherical geometry along the ray paths as derived by Hasha et al. [2008]."

23. *(1) The model does not contain the metric terms necessary to compute ray traces on a sphere. It is therefore only capable of solving traces for local Cartesian applications which are, however, bound to have errors for larger horizontal paths as in the present work.*

See above, the version of GROGRAT was modified to account for the spherical geometry [as in Hasha et al., 2008].

24. *(2) GROGRAT suffers from the occurence of caustics as the wave action equation is not expanded in phase space as for instance in Muraschko et al. [2015]. Overlapping wave action densities are therefore not meaningful.*

    Although it is true that GROGRAT might suffer from caustics, in our study, we are following the argument of Hertzog et al. [2002] that the introduced error of the amplitude amplification due to a caustic is negligible except below critical level. Approaching a critical level, however, the individual ray calculations are stopped well below and therefore we do not see this being a problem in practice. Therefore caustics should not be a dominant error source in our simulations.

    A sentence on this was added to the text: "In addition, GROGRAT might in principle suffer from occurrence of caustics [e.g. Lighthill, 1978], which, however, do not strongly affect the simulated amplitudes as discussed by Hertzog et al. [2002]."

25. *(3) The wave action equation is solved in terms of wave action fluxes. However, some arising term on the right-hand side is simply neglected and thus the wave amplitude is not energy preserving.*

    Standard GROGRAT amplitude calculations neglect the mentioned term completely. We have estimated the effect of this by locally approximating the term from background conditions. The impact is not visible for parts directly above mountainous regions, while regions with stronger horizontal propagation show enhanced fluxes after this modification. For an overview of this effect, we added an additional section to the Appendix. The calculations in the rest of his study are done with unmodified GROGRAT to assert comparability to previous studies, where GROGRAT has been successfully used.

    The corresponding section reads:

    ### "Amplitude correction due to horizontal dispersion

    The amplitude calculation in GROGRAT neglects contributions from the last term in Eq. 6. This might, however, lead to deviations for dispersing wave packets. Therefore, we estimated the contribution of this term locally according to

    $$c_{g,z} \nabla \cdot \vec{j} = c_{g,z} \left( \partial_x \frac{c_{g,x}}{c_{g,z}} + \partial_y \frac{c_{g,y}}{c_{g,z}} \right), \tag{R1}$$

    where the analytical expressions for $c_{g,i}$ were taken from Marks and Eckermann [1995]. This gives a local approximation of the change in the horizontal extent of the wave packet along the ray path.

    The GWMF experiments for June and July 2006 in comparison to the unmodified GROGRAT amplitudes are shown in Fig. R1 and Fig. R2, respectively. To both, the observational filter of HIRDLS has been applied. In direct comparison, we see that the general amplitude above strong orography is mostly unchanged by the modification. On the other hand, regions, where GWs propagate to are enhanced by this modification. This is as expected, since the term in Eq. R1 is related to the horizontal dispersion of the GW, which is stronger for refracting and turning waves. GWs that propagate far from their sources are more likely to encounter differing wind conditions and thereby refract or turn. The local approximation of this effect is, however, not a replacement for ray tube techniques describing the GW extent in phase space [e.g. Muraschko et al., 2015].

26. *(4) The ray tracing supposedly uses the Boussinesq approximation (c.f. L203). That would, however, neglect the anelastic amplification which is cruical for predicting the growth of gravity amplitudes with altitude.*

    This is badly phrased. The Boussinesq approximation is only considered for the derivation of the ray equations, not afterwards for the amplitude evolution. The exponential growth of

[Figure]

Figure R1: Similar to Fig. 6, the monthly mean GWMF prediction for January 2006 from the MWM is shown at different altitudes. The left column uses the standard GROGRAT amplitude correction, the right column a modified version, where the last term in Eq. 6 is approximated locally according to Eq. R1. The observational filter of HIRDLS has been applied to both data sets. Note the logarithmic color scale.

[Figure]

Figure R2: Similar to Fig. 10, the monthly mean GWMF prediction for July 2006 from the MWM is shown at different altitudes. The left column uses the standard GROGRAT amplitude correction, the right column a modified version, where the last term in Eq. 6 is approximated locally according to Eq. R1. The observational filter of HIRDLS has been applied to both data sets. Note the logarithmic color scale.

amplitude with altitude is considered in the used version of GROGRAT. The sentence was changed to:

"Acoustic waves are neglected within the derivation of the Boussinesq approximation. However, in the final calculations, the wave amplitude grows with decreasing density as usual."

27. *On the one hand a quantification of the errors remains unpublished, on the other hand there is a range of publications suggesting rather simple fixes for the deficiencies (e.g. Muraschko et al. [2015]; Hasha et al. [2008]). Working with an unrevised GROGRAT does therefore not follow state of the art practice and makes the presented results unreliable with respect to the predicted temperatures and momentum fluxes.*

The deficiency with regard to the geometry have been remedied according to Hasha et al. [2008] in the version that is used in this study. The amplitude enhancement near caustics on the other hand are not addressed in this study. Following Hertzog et al. [2002], the implications for the amplitude are assumed to be small enough to be negligible, especially in the full horizontal ensembles considered here. Below critical levels, this could in theory become a problem, in practice, however, the ray tracer stops reasonably far below these to describe the GWs only before the caustics occur. GROGRAT is inherently linear and a rephrasing to weakly non-linear GW packets would be a very interesting endeavor. It is however beyond the scope of this work.

28. *- general - The lower boundary condition of the ray tracer is not described here. It thus remains unclear how the ray tracing is launching rays in space and time. Only through the analysis one may guess that there is one ray ray per ridge (launched in the center?) with unknown distribution in time.*

Very good point and needs mentioning. The launching process is now detailed in the text (see comments below). Indeed a single GW is launched for every mountain ridge at every time step of the background data. This corresponds to hourly for the Southern Andes test case and 6-hourly for the global considerations.

29. *- L177f - "[...] the ground based frequency of all our waves is assumed to be zero [...]" This contradicts later considerations and the notion that the used ray tracing supposedly considers temporally changing background fields. Maybe the author refer to the lower boundary condition? If not this may be a major contradiction.*

This section describes the initialization of MWs within the ray tracer and therefore the (surface) boundary condition. This has been made clearer in the text as follows:

"Since we are considering mountain waves, the ground based frequency of all our waves is assumed to be zero at launch, $\omega_{gb} = 0$, which in turn leads to the intrinsic frequency.."

30. *- L185, Eq. 2 - While the formulation is rather standard, the authors use a tuning constant of 0.32 but do not explain reason for the choice or even the fact that it is a tunin constant. Please clarify.*

This tuning parameter stems form the cited literature (0.64 as the conversion factor between kinetic and potential energy). This has been made more explicit in the text:

"To account for surface friction of the low level wind and potential blocking at low speeds, a reduced, effective displacement amplitude is calculated following the discussion in Barry [2008, pp. 72-82], who states that the conversion factor between kinetic and potential energy due to surface friction effects is about 0.64. In addition, the amplitude of the displacement excited by air forced vertically over the Gaussian mountain is assumed to be about half the air parcels total vertical displacement:"

31. *- L194, Eq. 4 - The coordinate system of the equation system is not clear. The absence of metric terms (c.f. Hasha et al. [2008]) suggest a local Cartesian system but no procedure for projection is mentioned. This nmeeds clarification. If the integration is done on the sphere it needs revision.*

This is performed in spherical coordinates, where the correction terms for the geometry are taken from Hasha et al. [2008]. To clarify, the text has been altered to:

"The ray-tracing of the excited GWs itself is performed by (a modified version of) GROGRAT [Marks and Eckermann, 1995], which implements the ray equations derived in Lighthill [1978] including corrections for spherical geometry as derived by Hasha et al. [2008]."

32. - *L198ff - It is unclear how the ray tracing is generating time series of wave eave perturbation quantities. Do the authors continuously launch rays at the source location taking into account the changing background? If not, how do the authors justify temporatl changes in the background winds and overlapping structures given the transient nature of IGWs (c.f. Bölöni et al. [2021])? This problem propagates into the understanding of the time series in the analysis as the meaning is unclear.*

    MWs are launched at every time step of the background data, i.e. every 6 hours in the global case. The corresponding launch parameters, $m$ and $\zeta$ are calculated from the given conditions. This has been expanded on by including the following sentence in the text:

    "In the specific simulations, a single MW is launched at the center of each mountain ridge at every simulated time step (i.e. every 6 hours in the global and every hour in the Southern Andes case) with launch parameters derived from the corresponding atmospheric conditions."

33. - *L203 - Do the authors really use the Boussinesq approximation? If so, how do they justify neglecting the anelastic amlification effect?*

    As mentioned above, this is phrased in a bad way. The Boussinesq approximation is only in use for the derivation of the ray tracing equation as is common. In the explicit ray tracing, however, the amplitude grows with decreasing density as usual.

34. - *L207 - This is misleading. GROGRAT, according to Marks and Eckermann (1995), does not solve this equation but a prognostic equation for the wave action flux instead (c.f. Marks and Eckermann, 1995, Eq. 4). Moreover the wave action is not conserved along the path. Moreover Eq. (6) is a flux equation rather than a transport equation and in this formulation the wave action density, A, is not conserved along the path even when there is no turbulent damping. Only when expanding into a phase space wave action density this may be the case. In that case, however, the physical extend of the wave packet surrounding the carrier ray (in other words the phase space ray volume) may change its shape along the ray trace (c.f. Muraschko et al. [2015]).*

    This is true and was changed in the revised version. The wave action conservation is not used in GROGRAT amplitudes, and thus they underlie an intrinsic source of uncertainty. The corresponding section has been changed to:

    "For the prediction of GW amplitudes along the ray path, GROGRAT considers the vertical flux of wave action, $F = c_{g,z}A$, where $A$ is the wave action and $c_{g,z}$ the vertical component of the group velocity. The corresponding equation is given by [Marks and Eckermann, 1995, , Eq. 4]:

    $$\frac{dF}{dt} = -\frac{2}{\tau}F - Fc_{g,z}\nabla \cdot \vec{j},$$

    where $\vec{c_g} = c_{g,z}\vec{j}$ is the wave's group velocity and $\tau$ the parameterized damping time scale. The last term on the right-hand side is neglected since it would need evaluation using a "ray tube" technique, which is not implemented in GROGRAT [c.f. Marks and Eckermann, 1995, Lighthill, 1978]. A more precise consideration using conserved wave action along the path requires a much more involved description of the wave packet in full phase space [c.f. Muraschko et al., 2015]."

35. *Section 3.3 Representation of ray-tracing data*

    - *general - As before this section suffers from missing information on the methods used. In particular it is unclear when the traces, reconstructed at a specific time and position, were started in time. Moreover, as before the projection remains unclear.*

    The description of the ray initialization has been added in scope of the MWM description and as by the following comments. In the same way a note on the local Cartesian projection has been added.

36. - *L221 - L224 - As the temporal scheme is unclear from the description of the ray tracing algorithm it remains unclear what traces would be taken into account for the reconstruction. This needs clarification.*

    A note on this has been added: "In particular, for the selected time $t$ all rays that are still propagating (i.e. launched before but not yet terminated) at this time are considered." Combined with the above addition of the ray launching frequency (at every time step) this point has been made more clear.

37. - *L226, Eq. 7 - To reconstruct the wave phase the authors utilize a second order Taylor expansion but neglect several second order terms. In particular both terms with horizontal gradients of both the horizontal and vertical wavenumbers are not taken into account. 3D oblique porpagation would suggest that as long as the horizontal gradients act on the scales of the wave packets these terms are non-zero. The fact that the vertical term is non-zero reflects on the very same argument. Based on L292ff the ridge lengths are of the size between 75 and 500km. While the approximation that the horizontal wavenumber may not change throughout a wave packet might hold for small horizontal distances it breaks down for larger packets. If so, also the assumption that the wave packet extend does not shear in the horizontal but stays a rectangle breaks down as well. This simplification potentially poses an important error source and therefore needs mentioning and explanation in the text.*

    This is a good point, there are a lot of known uncertainties that should have been pointed out. The following paragraph has been added to the revised version:

    "There are a few uncertainties introduced by the simplifications that have been made in this reconstruction of temperature perturbations. For one, in Eq. 7, the horizontal change of wavenumbers $k$, $l$ and $m$ is neglected, since only the vertical change can be calculated reliably from a single ray path. In addition, the vertical change of horizontal wavenumbers has been neglected, because the vertical change in vertical wavenumber dominates here (especially when approaching critical levels [e.g. Nappo, 2012]). In addition, the change of the shape of wave packets is not considered here as their footprint is assumed to be rectangular at all times. Due to the horizontal extent of some wave packets and the horizontal shear of group velocities, this is only correct in a first approximation."

38. - *L229ff - The authors mention that the chirp rate is reconstructed from the closest time steps near the target altitude. Does that mean that the chirp rate is calculated on the characteristic of the ray (total derivative in time) rather than in the vertical (partial derivative in the vertical)? What is the error of that? How would the results change if only the linear term in the vertical would be considered?*

    The results do not change significantly if only the linear term is considered, however, this can lead to inconsistent phases in vertical cross sections of the temperature field (which are not considered here but in other studies). So for pure horizontal considerations, there is almost no gain from this term.

    Although it is true, that the derivative is approximated by the total Lagrangian derivative along the ray, this is fine in the considered slowly changing background fields, which are more or less static while the single ray is traced from one altitude level to another. The timescales to consider here are about 10-40 min between levels and 6-hourly background data fields (interpolated to a spline). The GW could however end up in a different background due to oblique propagation, if it is very fast, which is not much of a concern when looking at the scales at which the background has been separated. This estimation is, however, the only reliable way to get a chirp rate from the singular ray trace data and is useful for reconstruction of overlying GW packets from the same source.

39. - *L237, Eq. 8 - As before the reconstruction of the temperature field is statically coupled to the exciting ridge in terms of physical extent and and shape. The assumption of a static wave packet shape is inconsistent with a WKB theory assuming slowly varying background fields and thus needs justification.*

    The exciting ridges considered in this study are of limited length and therefore the constant width of the reconstructed wave packets is a fair first approximation. A more complex

consideration is beyond the scope of this study and would outbalance the Mountain Wave Model, which is by construction a simplified approach.

The following note has been added to the text: "In addition, the change of the shape of wave packets is not considered here as their footprint is assumed to be rectangular at all times. Due to the horizontal extent of some wave packets and the horizontal shear of group velocities, this is only correct in a first approximation, but since the exciting mountain ridges, in general, are of limited length this approximation is reasonable."

40. *- L248 - The authors mention that the (pseudo?) momentum flux scales with the square of the temperature. It would be important for the reader to know what formulation the authors use to calculate the momentum fluxes. If, in accordance with linear theory, they calculate the pseudo-momentum flux the terminology should be adapted accordingly (c.f. Achatz et al. [2017] and Wei et al. [2019]).*

This has been changed to state that the pseudo momentum flux is calculated as in previous studies. The explicit formulation of the GWMF has been added and the reference of the temperature-GWMF relation [Ern et al., 2004] has been added:

"The spatial distribution of the pseudo momentum flux (further also referred to as GWMF) is performed across the specified data grid using the same wave packet assumption as for temperature perturbations in Eq. 8. The maximum pseudo momentum flux of the wave, $F_{\max}$, is given by the raytracer [Marks and Eckermann, 1995], but can also be calculated from the temperature amplitude following the relation given by Ern et al. [2004]. They state GWMF $\sim T_{\mathrm{amp}}^2$ and therefore the GWMF of a wave packet, $F$, has to decay faster than the temperature perturbation (by a factor of 2). The oscillating term, $\cos \phi_{\mathrm{tot}}$, has to be dropped because $F$ depends only on the amplitude and not the phase. As in Sec. 3.3.1, the edges of the wave packet are smoothed with the same 6th-order Butterworth function. The resulting equation in analogy to Eq. 8 is then given by:

$$F = F_{\max} \frac{1}{\sqrt{1 + \left(\frac{2d_{\mathrm{perp}}}{l}\right)^{12}}} \exp\left(-2\left(\frac{2d_{\mathrm{along}}}{\lambda_{\mathrm{hor}}}\right)^2 - 2\left(\frac{2d_z}{\lambda_z}\right)^2\right). \qquad (2)$$

"

41. *- L255, Eq. 9 - The authors explain that they compute an integral average of the momentum flux from the Lagrangian reconstruction over the target grid cell. While they mention integrating over cells in a sub grid it remains unclear how the integral is actually computed. I am guessing that the authors approximate the integral as a sum of the cell centered values multiplied with the cell areas. In that case I would like to see a statement on the convergence of the area integral with only 9 values (3x3 sub grid). How large is the error of the integration scheme and how does it compare to the error introduced by assuing a static geometry of the reconstructed field?*

The expected error due to limiting to 3x3 sub points is sufficient for the accuracy, that we aim at in this study. Fig. R3 shows the error that might be expected for a GW randomly distributed in the grid cell with grid spacing 1.5°, that has been used in the global considerations (as added in the revised text). The gray dashed line gives a 10% threshold. In x-direction, the scale dependence of the error is shown. For 3 subsamplings, the expected error for scales, that we consider (i.e. 80 km and above) is below 9%, for scales above 100 km, which is the majority of MWs in this study, it is below 3%.

Therefore error introduced by the sampling can be neglected for the GWs considered in this study, especially when considering the MWM as a simplified model.

42. *Section 4.1 Detected structures and scales in the MWM*

*- general - I think this part is important as the misrepresentation of the mountains in terms of spectral power can lead to both - over and underestimates of the excited gravity wave energy. The chosen comparison, does however, leave a couple of questions open. It would also be*

[Figure]

Figure R3: Expected (relative) error due to subsampling compared to the true integral over the grid cell for 1 to 9 sampled points (per dimension). The scale dependence is given in x-direction and the dashed gray line shows the 10% threshold.

*nice to show the Tibetan plateau as well as the orography in southern Africa as these are mentioned explicitly in the analysis.*

This section has received a rework along the comments given below. In addition there was a figure added in the appendix that shows the orography reconstruction of southern Africa and Mongolian Plateau as requested by the reviewer.

In particular, the following section was added:

**"Himalaya and southern Africa**

The same topography reconstruction separated in small and large scale contributions as in Fig. 2 for the Mongolia region and southern Africa are shown in Fig. R4. Similar comments as for the Southern Andes and Rocky Mountain region are applicable here. The very large scale plateaus, especially the Tibetan Plateau as a whole, are not described due to the limitation in terms of horizontal scales. Small scale features, on the other hand, are approximated well in terms of orientation and location. The total height for the larger, i.e. 200 km and above, scales are, as for the Rocky Mountains region, over-represented due to multiple ridges contributing to the same feature in different directions.

43. - *L265ff - The comparison in Fig. 2 is made between the raw dataset and the reconstruction isolated by scales. It would be nice to reorder the comparison so that it shows the raw orography, the filtered orography as used for the ridge construction and sum of the constributions of small and large scales. Then, in a second step, the large and small scales could be analyzed. This procedure would have the advantage that the reader wold get a much better feeling for included and filtered scales of the ridge construction algorithm. Moreover, I am wondering: Does the ridge reconstruction into a full orography field demand a maximum of all overlaying detected ridges rather than a sum? This could possibly deal with the representation problem the authors mention in L268ff.*

There are multiple spectral bands used in the reconstruction of the full topography on which the ridge-finding has been performed individually. To give a better overview what happens at each given scale, a corresponding section has been added to the Appendix.

[Figure]

Figure R4: Same as Fig. 2 but for the Mongolia region (a – c) and southern Africa (d – f). The underlying topographic data is shown in panels a and d, respectively, while the reconstruction from the identified idealized Gaussian mountain ridges are shown in panels b, c, e and f. The reconstruction is separated into small scales ($\leq 150\,\mathrm{km}$, panels b and e) and large scales ($>150\,\mathrm{km}$, panels c and f).

The reconstruction using a maximum instead of the sum has been tested but leads to much too small elevations. It is necessary to superpose the detected scales by a sum to get the true elevation height. However, a combination of both might prove fruitful, where for each spectral interval a maximum reconstruction is performed individually and these contributions are summed afterwards. This would limit the effect of isotropic features which are sampled in multiple directions.

The added section of the Appendix reads as follows:

**"Southern Andes ridge-finding**

The ridge-finding algorithm operates on scale intervals and detects the ridges in the bandpass filtered topography data. This is illustrated in Fig. R5, where the topography after application of the bandpass filter is shown in the left column and the corresponding reconstruction of detected ridges in the right column. Note that each spectral band was reconstructed by taking the maximum of all ridges that cover the same spot. The four largest scale intervals given in Tab. 1 yielded no ridges in this region and are therefore not shown.

Figure R5 shows that each individual contribution to the full spectrum of elevation features in the original elevation data is detected and reconstructed in a good way. Features agree in orientation height and length."

44. - L290ff - *The presentation of the statistics of the detected ridges (Fig. 3) is hard to read and therefore the description is difficult to understandFor instance the statement that longer ridges are associated to large scales and small amplitudes (L294) is not visible from the figure. I therefore suggest to make a scatter plot for the horizontal scales (L, $\lambda\_hor$) and color code the dots with the feature height, h. This would give the reader a much better understanding of the statistics and the relationships between the different detected parameters.*

Indeed a scatter plot could show the relation between horizontal scales and ridge length better. Therefore the corresponding figure was adapted according to the comment:

45. - L301ff - *"[...] the MWM does a good job in representing features on variaous scales." It seems odd to testify a "good job" while several shortcomings are described leading to very large descripancies in the reconstructions as shown in the example plots (c.f. Fig. 2a and c, at ~46°S). This might either be mended by an improved representation / reconstruction (see comments above) or by focusing on whether the detection is appropriate to determine orographic waves.*

[Figure]

Figure R5: Reconstruction from the different detected idealized two-dimensional ridges in the given bandpass filtered elevation data. The left column show the result of the bandpass filter, the right column the corresponding detected ridges. The spectral band varies with row from smallest to largest scales. For the reconstruction, if multiple ridges cover the same spot, the maximum height was taken. Note that only the four smallest scale intervals yielded ridges and are therefore shown here.

[Figure]

Figure R6: Scales of detected ridges that contribute to the approximation of the Southern Andes region (Fig. 2b and c) and western North America (Fig. 2d and e). The scatter shows the length of the ridge versus the detected wavelength in km (cf. Sect. 3.1) and the corresponding height in color shading. Note the logarithmic scales on both axes.

The corresponding paragraph was changed to emphasize that the detection of ridges by the presented algorithm is appropriate for a wide range of scales and the main MW sources in the orography. The text now reads:

"In conclusion, the MWM detects and represents orographic features of the elevation of various scales under consideration that the representation of elevation data by two-dimensional ridges has some intrinsic problems (especially with isotropic and plateau-like features). Although this is no indication of whether we cover all relevant scales, it provides confidence in the underlying ridge detection algorithms as a tool to extract the main MW sources and the corresponding parameters."

46. *Section 4.2 Residual temperature as compared to ECMWF operational analysis data*

*- general - At some parts of the comparisons the authors seem to mix up the comparison and evaluate the IFS rather than evaluating the MWM agains the IFS. Being more clear about it would make the point the authors want to make a lot stronger. Also, how do the ampülitudes compare well if the MWM uses Boussinesq dynamics and neglects the anelastic amplification? Is it negligible or do the authors actually take the amplification into account? The amplitues suggest the latter, this needs clarification.*

The section has been partially rewritten to set the focus on the evaluation of the MWM. E.g.:

"At 30 km altitude, the polar vortex turns from mainly eastward to a more north-eastward direction above the Andes. This leads to a change in the horizontal wind gradients, and in turn to GWs refracting and turning as well (Fig. 4g and h). Both data sets show gravity waves mainly facing to the south-west instead of westward, as a consequence of horizontal refraction towards the stronger winds in the south-east [Krasauskas et al., 2022]. The MWM predicts significant MW activity above the Atlantic Ocean, which is (mostly) stemming from the main Andes mountain ridge and has propagated to the east. These patterns are confirmed by the IFS data, which shows a large connected band of GWs above this region. The MWM does not show such high continuity due to the (short) mountain ridge approximation and reconstruction from single wave packets. Nevertheless, the phases of different wave packets fit well together and a coherent structure is seen. Propagation to the west is only faintly predicted by the MWM, while the IFS data shows strong GWs far above the ocean."

As mentioned in a previous comment, only the derivation of the ray tracing equations is performed in Boussinesq approximation. The anelastic amplification is taken into account for

the wave amplitudes upon explicit ray-tracing. This has been clarified at the corresponding location in the text.

47. - L321 - "[...] which can be associated with a similar pattern in the IFS data." I do not understand how the patterns are associated, In particular as there are patterns all over the oceans in the IFS data. I suggest to connect the observation of the pattern above the Atlantic at 8km altitude observed by the MWM with the leading argument on structures above the ocean in L315ff.

This part has been rephrased and mentioned after talking about the IFS features above the ocean. In particular, the text now reads:

"..completely different tropospheric processes, that our scale separation anomalously picks up. However, the MWM also shows a large scale pattern to the east of the continent, which might indicate that orographic GWs of large scale (and therefore higher horizontal group velocity) might also add to the patterns seen in the IFS. Above the Andes.."

48. - L326f - "The IFS, however, seemingly does a better job of resolving very small scales." I was under the impression that the IFS data was considered the "truth" to be compared against based on the data assimilation it is based on. With this sentence the authors seem to evaluate the IFS based on the MWM, which is however the model which is to be evaluated against the proven IFS results. Please reformulate.

This sentence has been rephrased to "Features of very short scales (i.e. below about 60 km) are not as well represented by the MWM temperature reconstruction as by the IFS, which is partly due to the lower scale limit set to 80 km."

49. - L332ff - "We see significant GW activity over both the Pacifi Ocean as well as the Atlantic Ocean in the IFS data, which can be mostly explained by oblique propagation of MWs from the Southern Andes as indicated by the MWM data." Again, it would be better to strictly seperate arguments here. Explaining the patterns of the IFS with the MWM is not the objective here. Rather the MWM needs evaluation when comparing to the IFS data.

This has been addressed in the revised version. The corresponding section now reads: "The MWM predicts significant MW activity above the Atlantic Ocean, which is (mostly) stemming from the main Andes mountain ridge and has propagated to the east. These patterns are confirmed by the IFS data, which shows a large connected band of GWs above this region. The MWM does not show such high continuity due to the (short) mountain ridge approximation and reconstruction from single wave packets. Nevertheless, the phases of different wave packets fit well together and a coherent structure is seen. Propagation to the west is only faintly predicted by the MWM, while the IFS data shows strong GWs far above the ocean."

50. - L336ff - For which height is the inverse ray tracing done? Does that refer to the patterns at 30km height?

This refers to Fig. 4g, which has been clarified in the text: "We analyzed the spectral characteristics of the waves around 45°-50°S, 60°-66°W in Fig. 4g at 30 km altitude using the S3D technique (3D wave fitting in a subdomain, previously used by e.g. Preusse et al. [2014], Geldenhuys et al. [2021], Krasauskas et al. [2022]."

51. - L340 - The location is missing.

This should have been "∼49° - 51°S", which the revised text now reads.

52. - L345 - The authors refer to Fig. 4 but mention Fig. 5.

This has been changed accordingly in the revised text.

53. - L347, Fig. 5 - For which are region at what time is the analysis of the momentum fluxes done?

This information has been added to the text and the caption: "The turning of GWs in the Southern Andes region (same as in Fig. 5) within the MWM can be seen in Fig. 5, which

accounts for all GWs launched between 20.09.19 00:00 and 21.09.19 23:00 with each mountain ridge launching a single GW every hour."

54. - *Fig. 5 - What does the ray count entail? Related to the question concerning the temporal dependency of the lower boundary condition (see discussion above) of the MWM it is not exactly clear what the ray count represents. Are those rays from strictly different ridges, or are these overlays from dleayed times due to the transient propagation in a slowly varying medium?*

The ray count entails all rays launched in between 20.09.19 00:00 and 21.09.19 23:00, where rays are launched every hour. Thus, this includes both, rays from different ridges as well as rays from the same ridge launched at different times. This has been made more specific in the text by addition of the following sentence in the caption: "Direction vs. altitude distribution of MWs as modeled by the MWM in the Southern Andes region (same as in Fig. 4). This considers all GWs launched between 20.09.19 00:00 and 21.09.19 23:00 with each individual mountain ridge launching a single GW every hour. "

55. *Section 5.1 Global distributions of momentum flux*

   - *L365ff - The authors hint at errors in the HIRDLS data. This would need references or data showing the deficiency.*

The was not aimed at errors in HIRDLS data but at not understanding the comparatively low GW activity in satellite GW climatologies. A reference to Ern et al. [2018] has been added to give more context.

56. - *L379 - How is the MWM data "binned" after applying the observational filter? Does that mean the intefgal over the GWMF is sinply done over other cells (c.f. Eq. 9)?*

The GWMF distribution is generated on a $1.5°$ resolution from the filtered ray tracing data before the binning to the (rather coarse) HIRDLS data resolution is performed. A note to clarify this process has been added to the text:

"For a direct comparison of global GWMF distributions, we apply an observational filter that accounts for HIRDLS observation geometry [Trinh et al., 2015] to every ray of the MWM before calculating the GWMF distribution following Sect. 3.3.2 on a $1.5°$ resolution and bin the resulting GWMF in the same way as the HIRDLS profiles.."

57. *Section 5.1.1 January 2006*

   - *general - This section is partly difficult to read and understand due to three reasons. First, the meaning of a monthly mean is unclear as the time dependence of the MWM lower boundary is unclear (mentioned above). Second, some of the mentioned conclusions from the datasets seem speculative. Third, the representation of the data on logarithmic (nowhere mentioned) levelsets with very few levels is quite hard to read and interpolates over many details. I therefore recommend to streamline the section and change the data representation to pseudo color plots. Moreover I recommend linear color scales for the horizontal maps in Figs. 6 and 10 as the currently shown levels are barely covering 2 orders of magnitude and the features in HIRDLS are hard to distinguish in the flat color scaling.*

There have been major modifications of this section according to the comments and answers below. Especially the speculative touch of some parts of the text has been changed.

With respect to Figs. 6 and 10: the number of color levels have been increased, which makes the different features mentioned in the text more easily visible. Linear color scale is not used, as it reduces the visibility of the shape of features in the MWM data. A note on the logarithmic nature has been added to the caption.

58. - *L395f - "There are two possible reasons for the GWs missing in the satellite observations at higher altitudes." This claim needs justification / references of reported descripancies. Moreover*

The referred section might be indeed phrased a bit too certain about the mentioned reasons for discrepancies. This has been reduced in certainty in the found possible reasons.

The corresponding sentence was changed to "There are at least two possible reasons for these GW features missing in the satellite observations at higher altitudes that are considered here:"

59. - *L405f - "There is also a minor southward shift of GWMF towards California visible which is also picked up by the satellite data." This seems highly speculative. The supposed shift is located at the edge of the observations and thus it is unclear what influence structures in the not observed regions have. Moreover there are no structures visible beyond the global GMF band in the HIRDLS data.*

This feature is better visible (also at higher altitudes) in the updated plots with additional color levels. The text is now less certain about this feature to account for it being at the edge of the observations:

"There is also a minor southward shift of GWMF towards California visible which is also hinted at by the satellite data. This feature sits, however, right at the edge of the observation and is therefore not completely seen."

60. - *L409ff - "Therefore*

*the strong signature in the MWM data could be another hint at this or another process missing in our current understanding of GW physics." This, too seems speculative. The quantitative correctness of the MWM over the Rockies was not shown. Instead it is stated that the momentmum flux results of the MWM are too strong at lower altitudes. Moreover some known and cited physical mechanism (total breakdown of the waves) explaining the amplitude behavior not captured by the MWM are mentioned. This context rather suggests that some of the many simplifications of the MWM are not covering the dynamics here, rather than some unknown GW physics.*

This sentence was indeed a bit speculative. It was changed to refer more to the aspect, that the MWM does not model such a process as of now: "Therefore, the strong signature seen in the predictions could be an indication of a process that is not yet modeled within the MWM."

61. - *L414f - "Another strong feature predicted by the MWM are local maxima in the Southern Hemisphere above New Zealand and the southern Andes, which are matched by the observations." Again, this seems speculative. I cannot see any particular structure over New Zealand in the HIRDLS observations. While the HIRDLS signal over the southern Andes could be interpreted as originating from the mountain waves its magnitude is significantly lower than the momentum fluxes predicted by the MWM.*

The plots have been updated to see more structure in the HIRDLS observations. In addition, the corresponding section was altered to reflect and explain that the MWM shows stronger amplitudes in theses regions:

"Another strong feature predicted by the MWM are local maxima in the Southern Hemisphere above New Zealand and the southern Andes, which are matched partially by the observations. These maxima strongly decrease in higher altitudes and vanish completely at 25 km as expected, since MWs are filtered by the wind reversal at around 20 km in the summer hemisphere. The MWM prediction is stronger than the observations, which could, again, be related to the above-mentioned processes." [referring to complete breakdown of strong amplitude GWs]

This feature is also more evident once the plots are updated to show more color levels. The strong SA maximum has received another explanatory sentence: "The MWM prediction is stronger than the observations, which could, again, be related to the above-mentioned processes."

62. - *L418f - "However, their persistence at 25 km altitude in the observations is not consistent with them being MWs." I suggest a reformulation of this statement as the waves observed at higher altitudes and inside the polar vortex are most likely originating at different locations (possibly from other sources), given they are advected by very high wind speeds. These waves are, therefore, most likely not "persisting" to higher altitues as suggested.*

This has been rephrased as "The presence of this pattern at 25 km altitude in the observations is, however, not consistent with them being MWs as well. Instead, these are most likely of other origins."

63. *- L420 - 427 - This paragraph is somewhat confusing as it is not clea when the authors refer to HIRDLS and when to the MWM. Particularly the sentence in L425 was very confusing to me at first. I suggest a streamlining of the whole paragraph clearly laying out the differences.*

This paragraph has been rewritten for a better understanding and more clarity. The revised text reads:

"In the North Atlantic region, the MWM predicts GW sources in Newfoundland, southern Greenland, Iceland, and Scandinavia. Strong eastward propagation of MWs is seen especially above Iceland, where the pattern of GWMF merges with the feature above Scandinavia. In addition, the MWM predicts eastward propagation from Newfoundland towards southern Greenland at 25 km, even though the GWMF values predicted by MWM are, generally, suppressed at this altitude. The HIRDLS observations show similar, although more complex features in this region. The aforementioned MW sources are clearly visible but merge into a band of strong GW activity at 20 km and above. This band follows the path of the polar vortex and might therefore be related to local GW sources such as jet imbalances and fronts [Geldenhuys et al., 2021]. The occurrence of other sources than orography in the observations is strengthened by the (slight) GWMF increase between 20 km and 25 km."

64. *- L433 - As mentioned above. Beyond the coined name blocking diagrams I suggest to use the term critical layer filtering to distinguish flow blocking from wave filtering to clarify the refferred to physics to the reader.*

This has been altered in the revised version. The term "critical level filtering" is now used throughout (except for referencing a "blocking diagram").

65. *- L435, Eq. 11 - This equation is singular for the extrinsic frequency / the phase speed being nearly zero which is notably true for orographic waves near the surface and where the wind is approximately static throughout the propagation path. I would thus suggest to emphasize that by the curves for ω_intr=0 only occur where ω_gb is non-zero.*

This is a good point, a note on this has been added after the derivation: "Note that the derivation of this equation requires $\omega_{gb} \neq 0$, which is not true for the considered MWs close to the launch site or in a static background."

66. *- Fig. 9 - It would be helpful if the considered time interval (Jan 2006) was added to the caption of the figure.*

This has been added in the revised version: "Blocking diagrams as introduced in Taylor et al. [1993] for the four regions shown in Fig. 8 for the time period of January 2006:.."

67. *- L443 - The authors note that the blocking diagrams only show critical layer filtering for ω_gb=0. I suggest to add a statement that this is true where the wind profile is approximately constant and near the topography where the refraction is not strong enough, yet.*

This is a note that should be kept in mind and is therefore added in the text as follows:

"Note, however, that these diagrams are only an indication of critical levels for MWs near their launch location and in an approximately constant wind profile, where $\omega_{gb} \approx 0$. In these conditions, critical levels are encountered wherever the horizontal wind projected onto the horizontal wave-vector becomes zero."

68. *- L456 - "This total breakdown of saturating waves is not represented in GROGRAT simulations, but could be a reason why HIRDLS sees less activity above the Himalaya (Fig. 6)" I suggest to reformulate this statement as it is not the MWM leading to patterns being observed in HIRDLS. Rather than that the observations hint at missing effects in the MWM, which I suspect the authors meant to say.*

Indeed, this text should be phrased more clearly. The revised text reads now:

"The lack of this effect in the GROGRAT simulations could be one reason for the enhanced GW activity above the Himalaya predicted by the MWM (Fig. 6)."

69. - L474 - There are no yellow lines in Fig. 9d anf f? I suppose the authors refer to the red lines?

Yes, it should have been "red". This mistake was fixed in the revised version.

70. - L475 - The authors refer to the wind reversal, I suppose at very low altitudes. This is difficult to see so I suggest the altotude is mentioned in the text so that the reader will not have to search for the detail in the plots.

A note on this has been added to the text: "The blocking diagram has a pronounced dumbbell shape from a wind reversal, which is confirmed by the wind profiles at low altitudes (about 4 km)."

71. Section 5.1.2 July 2006

- L495ff - Here the authors state to see eastward propagation in the HIRDLS dataset. Albeit this being the likely reason there is no propagation information in the data. I thus suggest to reformulate the paragraph arguing that the MWM suggest this to be the reason for the patterns observed in the HIRDLS dataset. Moreover a hint at why the contribution from the Antarctic Peninsula is missing in the (filtered) MWM would be helpful and complement the argument.

This section has been rewritten for cleaner presentation and a better argumentation. The revised text reads: "The HIRDLS data set shows a strong eastward spread around these maxima up to about 30°W over the Atlantic. A comparison with the MWM, which shows a very similar pattern with a larger maximum above the southern Andes, shows that the observed spread of GWMF can be explained by eastward propagating MWs. In a textbook case of oblique propagation from a single source region, the extent should increase with altitude. This is true in the MWM predictions, where MWs reach as far as 30°W at 25 km. The satellite observations, however, shows a decrease at 20 km followed by an increase in spread at 25 km altitude. Although the maximum of the MWM prediction is not as precisely localized as in the observations, it shows the same northward shift with higher altitude."

In addition, there is a note on the relatively weak GWMF around the Antarctic Peninsula added further down in the text: "This strong filtering at higher latitudes is also a likely reason for the relatively weak GWMF predictions of the MWM around the Antarctic Peninsula."

72. - L511 - What is meant with the "assumption"? The reference needs to be fixed, maybe be more specific.

This "assumption" refers to the scales that are filtered using the observational filter [Trinh et al., 2015]. This has been made clearer in the text. The reference should have been "Fig. 13", which is fixed in the revised version.

"the horizontal spectrum than we assume in the observational filter (c.f. Fig.13, Trinh et al. [2015])."

73. - L517ff - The attribution of the momentum flux to spontaneous from jet imbalances or katabatic flows seems somewhat ad hoc. That would be generally fine but has to be formulated witout a clear attribution. On a site note: If the ray tracing does not contain the necesseary metric terms it will have particularly strong errors near the pole. Should this be the case (as in the equations of Marks and Eckermann, 1995) it is a possible candidate for the missing northward propagation.

This is a very strong claim indeed. This was therefore changed to: "This feature in the observations might be related to MWs due to katabatic flow [Watanabe et al., 2006], or other non-orographic processes like imbalances of the polar jet or frontal systems. Partly, this lack of GWMF might also be related to the strong filtering in high latitudes by the observational filter."

Geometric corrections within GRORGAT were taken from Hasha et al. [2008] and should therefore describe the propagation correctly. Fair northward propagation is also seen in the unfiltered data (Fig. 13), which hints at the observational filter in combination with the coarse binning as one reason behind the lack of GWMF in this comparison.

74. *- L522f - "The location and strength of local maxima fits nicely between the two." Does that refer to the comparison between the MWM and the observations by HIRDLS? If so it would be contradictive as the following sentence states (and I agree), that the effect is not seen in HIRDLS as it cannot be separated from the background. Since no definite statement can be made about the considered structures in the observations I suggest to leave them out entirely.*

Since the Southern Alps is a hotspot for orographic gravity waves, I would like to briefly discuss the prediction of the MWM. Nevertheless, the comparison of exact features was too optimistic and was therefore removed. The revised section reads:

"Enhanced GW activity is seen in the MWM prediction above the Southern Alps and the Great Dividing Range/Tasmania, which has been shown by Eckermann and Wu [2012] to be a strong MW source. The MWs from Australia and Tasmania show a relatively localized pattern with increasing altitude, while the MWs from the Southern Alps show strong south-eastward propagation especially at higher altitudes. These features are hard to separate from the background in the HIRDLS data and can therefore not be entirely validated. The observations, however, shows some enhanced GWMF around the Southern Alps, which stretches to the south-east and merges with the background at around 170°W. A look into unfiltered MWM data (see Fig. 13) shows that there is strong north-eastward propagation from eastern Antarctica, which can partly explain enhanced fluxes in this region."

75. *- L534ff - "Weak GW activity above the northern Rocky Mountains, Greenland and the Japanese Sea can be assigned to structurally agreeing counterparts in HIRDLS. Note however that the baseline of HIRDLS is much higher than the predicted GWMF of the MWM and these features might as well not be visible in the observations at all" This statement is contradictive. If the structure cannot be seen in the HIRDLS data there is no attribution of any structure in the HIRDLS data with respect to orographic waves. I suggest to remove the comparison entirely.*

This is very true, this section does not add any value and is therefore removed.

76. *Section 5.2 Zonal mean momentum flux distributions*

*- L548 - Unclear. Black contour lines in which plot (Fig. 12 I suppose) and from which dataset?*

This refers to Fig. 12 and the wind data is taken from ERA5 data. This has been made clearer:

"The black contour lines in Fig. 12 show the monthly mean zonal wind as taken from ERA5 data."

77. *- L552f - "Another low altitude maximum at 30°S in the observations is probably not robust, [...]" This statement is not backed up and needs an argument. The HIRDLS data quality threshold permitted the structure and so the fact that it lies inbetween data that was flagged invalid is not enough to make that claim.*

This is indeed not enough to claim that this data point is invalid, however, since it stems from a very localized region in the horizontal distribution (the small spot west of Australia), it is not representative for a zonal mean but more like the local situation in this region. Drawing conclusions from this to the zonal mean is not possible. The text has been altered to present this problem in a better way: "Another low altitude maximum at 30°S in the observations is probably not representative of the zonal mean, as Fig. 6 shows, that it stems from a very small region with few data points west of Australia."

78. *- L557-567 - In this paragraph it is not always clear which dataset the authors refer to. Some streamlining would help understand the points better.*

This section was rewritten to give a better idea of which data set is considered at any point. The revised text reads:

"Corresponding data for July 2006 is shown in Fig. 12b and d for HIRDLS observations and MWM predictions, respectively. As in Sect. 5.1.2, the dominant feature in both data sets is the GWMF above the southern Andes and the Southern Ocean around 40°-50°S. The prediction from the MWM shows enhanced GWMF around the neck region at roughly 16 km, 30°S. A quick check against the MWM without the observational filter applied (Fig. 12f) shows that this is most likely not due to (northward) propagation, but due to the shift of MW parameters towards values that are better observable by the instrument, i.e. towards longer vertical wavelengths, and therefore less suppressed after filtering. The observations do not show southward propagation towards 60°S explicitly due to the strong band of GWMF obscuring individual features in the zonal mean. The predicted zonal mean from the MWM, on the other hand, does show neither strong southward propagation from the Southern Andes nor northward propagation of MWs from Antarctica, which could lead to the enhanced fluxes in the southernmost observations at highest altitudes (about 22–25 km). As mentioned in Sect. 5.1.2, this feature is, therefore, most likely of non-orographic origin and generated by sources higher up in the atmosphere like jet fronts and spontaneous adjustment. In the Northern Hemisphere, the strongest MW activity is seen above 65°N in the MWM predictions can be attributed to Greenland as a source, followed by another local maximum above the Himalaya around 40°N. Both features are confirmed by the observations. The MW activity is well confined by the wind reversal in the summer hemisphere in the simulation, as would be expected for MWs."

79. - L560f - "Below about 20km, there is also southward propagation towards 60°S." The presentation of zonal mean absolute momentum flux does not show any meridional propagation. Can this claim be backed up (for instance by showing meridional fluxes)? If not, remove it.

Indeed, this is very hard to distinguish. Therefore, this comment has been removed from the text.

80. - L570 - observational -¿ observational filter

This mistake has been fixed in the revised version.

81. - L585f - "Since the observations are limited due to clouds, this is only seen in the model data, but since this feature almost completely vanishes after application of the observational filter, HIRDLS would probably not have observed this." This is very speculative claim which adds little to the discussion. I suggest to remove the second part of the sentence and write: "Since the observations are limited due to clouds, this is only seen in the model data."

Agreed, this is rather speculative and cannot be shown. Thus this sentence was replaced by your suggested phrasing instead.

82. Section 5.3 Time evolution of GWMF distributions

- L596 - "this is usually happening" Needs a reference.

This sentence was changed to reflect, that this is the general case in our ray-tracing experiments:

"in our raytracing experiments, the strongest horizontal propagation takes place"

83. - L609 - "zonal propagation" -¿ "zonal propagation and advection"

This has been changed as suggested.

84. Section 6 - Conclusions

- L651 - "a study of blocking and wind filtering" Here it is unclear what is meant by blocking. Is it flow blocking in the PBL or critical layer filtering? See note on consistent terminology concerning blocking.

This terminology has been changed throughout the article to "critical level filtering". The specific section was changed to "..a study of critical level filtering due to the wind profiles.."

85. *- L666f - "Another finding is that it could be worthwhile to implement katabatic MWs in order to obtain increased fluxes northward of Antarctica and southward of Greenland." This formulation is somewhat misleading. I suppose the authors would like to improve the predictions at these locations rather than just obtain an increase.*

This is true, the aim would be a better representation of all orographic GW sources. The corresponding sentence was changed to: "..worthwhile to implement katabatic MWs in order to obtain improved GWMF predictions northward of Antarctica and southward of Greenland."

86. *- L691 - When you suggest ray-tracing as a gravity wave parametrization for GCMs it would be useful to reference works that have shown first implementations, strengthening the feasability of the idea. In particular I suggest to refer to the recent works of Bölöni et al. [2021] and Kim et al. [2021].*

Although this aimed at simple parametrization developed on static transport matrices that are one-time generated and used as a look-up for propagation predictions, not online ray-tracing parametrizations, it is still important to give a hint at other approaches using ray-tracing for GW parametrizations. Therefore the suggested references are cited in the conclusion as follows:

"Bölöni et al. [2021] and Kim et al. [2021] have already shown that ray-tracing can be used to improve the representation of subgrid-scale GWs in atmospheric models and that this is a path worth investigating."

87. *Appendix A2 - Mountain wave fit*

*- L711 - Do I understand correctly that the function R is equal to the function f from Eq. 1?*

This is the same ridge as given in Eq. 1. This section of the appendix has been rewritten and expanded in order to provide a better overview of the used methodology. This includes a renaming of this ridge $R$ to $f$, which was used in the text before.

88. *- L716 - How many parameters are fitted to the ridge? I suppose it is the mountain height as well as the half width? Please clarify what R depends on.*

This has been addressed in the reworked section. The ridges have the two parameters $h$ and $a$ which correspond to the height and width of the ridge, therefore $R = R(x, y; h, a)$.

89. *Appendix B - The (Probabilistic) Hough Transformation*

*- general - This appendix is very interesting but also brief. Given the scope of the manuscript that is to be expected, however, it urgently needs references for further reading and understanding.*

This section has also received a significant update and expansion and should now give a better presentation of the method itself. In addition, references to the first applications and occurrence of the Hough transformation are given by Duda and Hart [1972], Herman [2009].

90. *Appendix B2 - Sensitivity of the probabilistic Hough Transform*

*- general - The Hough transform seems to be done on a local Cartesian coordinate system, however the details about the projection are not mentioned. I suggest expanding on that.*

A note on this has been added to the overview of Sect. B: "The line detection assumes equidistant grid points and is therefore performed on a local Cartesian grid."

91. *- L752 and Fig. B1 - This is interesting and a nice overview but needs some consistency. It would be helpful to order the subplot along the changes in the two length scales (so that one row or column has one constant parameter) and also show the optimal values used. Finally: Why does Fig. 1d look very different from all figures here? Aren't they both the southern Andes region?*

This section and the figure was reworked to account for this comment. As suggested, the figure now shows different lengths with constant gaps in the same column and vice versa for rows. The reason for the skeleton looking different is a slightly different bandpass filter interval, that was used, this has been adjusted such that they are closer in scale and are more similar.

92. *Appendix C - Alternative representation of ridges as used in previous studies*

   *- general - This section does not seem to add much to the manuscript and could be removed.*

   Agreed, this is not as useful for comparisons against the mentioned studies and therefore this part is removed.

   ?

**References**

U. Achatz, B. Ribstein, F. Senf, and R. Klein. The interaction between synoptic-scale balanced flow and a finite-amplitude mesoscale wave field throughout all atmospheric layers: weak and moderately strong stratification. *Quarterly Journal of the Royal Meteorological Society*, 143(702):342–361, 2017. doi: https://doi.org/10.1002/qj.2926. URL https://rmets.onlinelibrary.wiley.com/doi/abs/10.1002/qj.2926.

S. P. Alexander, K. Sato, S. Watanabe, Y. Kawatani, and D. J. Murphy. Southern hemisphere extratropical gravity wave sources and intermittency revealed by a middle-atmosphere general circulation model. *Journal of the Atmospheric Sciences*, 73(3):1335 – 1349, 2016. doi: https://doi.org/10.1175/JAS-D-15-0149.1. URL https://journals.ametsoc.org/view/journals/atsc/73/3/jas-d-15-0149.1.xml.

C. Amante and B. Eakins. ETOPO1 1 arc-minute global relief model: Procedures, data sources and analysis, 2009. last access: 20 February 2020.

D. G. Andrews, J. R. Holton, and C. B. Leovy. *Middle Atmosphere Dynamics*, volume 40 of *International Geophysics Series*. Academic Press, 1987. ISBN 0-12-058576-6.

R. G. Barry. *Mountain Weather and Climate*. Cambridge University Press, Cambridge, UK, third edition, 2008. ISBN 978-0511754753. doi: 10.1017/CBO9780511754753.

G. Bölöni, Y.-H. Kim, S. Borchert, and U. Achatz. Toward transient subgrid-scale gravity wave representation in atmospheric models. part i: Propagation model including nondissipative wave–mean-flow interactions. *Journal of the Atmospheric Sciences*, 78(4):1317 – 1338, 2021. doi: https://doi.org/10.1175/JAS-D-20-0065.1. URL https://journals.ametsoc.org/view/journals/atsc/78/4/JAS-D-20-0065.1.xml.

A. de la Camara and F. Lott. A parameterization of gravity waves emitted by fronts and jets. *GEOPHYSICAL RESEARCH LETTERS*, 42(6):2071–2078, MAR 28 2015. ISSN 0094-8276. doi: 10.1002/2015GL063298.

R. O. Duda and P. E. Hart. Use of the hough transformation to detect lines and curves in pictures. *Communications of the ACM*, 15(1):11–15, 1972.

S. D. Eckermann and D. L. Wu. Satellite detection of orographic gravity-wave activity in the winter subtropical stratosphere over Australia and Africa. *Geophys. Res. Lett.*, 39(21), 2012. doi: https://doi.org/10.1029/2012GL053791. URL https://agupubs.onlinelibrary.wiley.com/doi/abs/10.1029/2012GL053791.

M. Ern, P. Preusse, M. J. Alexander, and C. D. Warner. Absolute values of gravity wave momentum flux derived from satellite data. *J. Geophys. Res. Atmos.*, 109(D20), 2004. ISSN 2156-2202. doi: 10.1029/2004JD004752.

M. Ern, Q. T. Trinh, P. Preusse, J. C. Gille, M. G. Mlynczak, J. M. Russell III, and M. Riese. GRACILE: A comprehensive climatology of atmospheric gravity wave parameters based on satellite limb soundings. *Earth Syst. Sci. Dat.*, 10:857–892, 2018. doi: 10.5194/essd-10-857-2018. URL https://www.earth-syst-sci-data.net/10/857/2018/.

D. Fritts and M. Alexander. Gravity wave dynamics and effects in the middle atmosphere. *Rev. Geophys.*, 41(1), APR 16 2003. ISSN 8755-1209. doi: 10.1029/2001RG000106.

M. Geldenhuys, P. Preusse, I. Krisch, C. Zülicke, J. Ungermann, M. Ern, F. Friedl-Vallon, and M. Riese. Orographically induced spontaneous imbalance within the jet causing a large-scale gravity wave event. *Atmos. Chem. Phys.*, 2021. doi: 10.5194/acp-21-10393-2021.

A. Hasha, O. Bühler, and J. Scinocca. Gravity wave refraction by three-dimensionally varying winds and the global transport of angular momentum. *J. Atmos. Sci.*, 65:2892–2906, 2008.

G. Herman. *Fundamentals of Computerized Tomography: Image Reconstruction from Projections.* Advances in Computer Vision and Pattern Recognition. Springer London, 2009. ISBN 9781852336172. URL https://books.google.de/books?id=hF68xAEACAAJ.

H. Hersbach, B. Bell, P. Berrisford, G. Biavati, A. Horányi, J. Muñoz Sabater, J. Nicolas, C. Peubey, R. Radu, I. Rozum, D. Schepers, A. Simmons, C. Soci, D. Dee, and J.-N. Thépaut. ERA5 hourly data on pressure levels from 1979 to present. copernicus climate change service (C3S) climate data store (CDS), 2018. accessed February 17, 2022.

H. Hersbach, B. Bell, P. Berrisford, S. Hirahara, A. Horanyi, J. Munoz-Sabater, J. Nicolas, C. Peubey, R. Radu, D. Schepers, A. Simmons, C. Soci, S. Abdalla, X. Abellan, G. Balsamo, P. Bechtold, G. Biavati, J. Bidlot, M. Bonavita, G. De Chiara, P. Dahlgren, D. Dee, M. Diamantakis, R. Dragani, J. Flemming, R. Forbes, M. Fuentes, A. Geer, L. Haimberger, S. Healy, R. J. Hogan, E. Holm, M. Janiskova, S. Keeley, P. Laloyaux, P. Lopez, C. Lupu, G. Radnoti, P. de Rosnay, I. Rozum, F. Vamborg, S. Villaume, and J.-N. Thepaut. The ERA5 global reanalysis. *Quart. J. Roy. Meteorol. Soc.*, 146(730):1999–2049, JUL 2020. ISSN 0035-9009. doi: 10.1002/qj.3803.

A. Hertzog, C. Souprayen, and A. Hauchecorne. Eikonal simulations for the formation and the maintenance of atmospheric gravity wave spectra. *Journal of Geophysical Research: Atmospheres*, 107(D12):ACL 4–1–ACL 4–14, 2002. doi: https://doi.org/10.1029/2001JD000815. URL https://agupubs.onlinelibrary.wiley.com/doi/abs/10.1029/2001JD000815.

J. R. Holton. The influence of gravity wave breaking on the general circulation of the middle atmosphere. *J. Atmos. Sci.*, 40(10):2497–2507, 1983. doi: 10.1175/1520-0469(1983)040¡2497:TIOGWB¿2.0.CO;2.

B. Kaifler, N. Kaifler, B. Ehard, A. Doernbrack, M. Rapp, and D. C. Fritts. Influences of source conditions on mountain wave penetration into the stratosphere and mesosphere. *Geophys. Res. Lett.*, 42(21):9488–9494, NOV 16 2015. ISSN 0094-8276. doi: 10.1002/2015GL066465.

C.-W. Kang, R.-H. Park, and K.-H. Lee. Extraction of straight line segments using rotation transformation: generalized hough transformation. *Pattern Recognition*, 24(7):633–641, 1991. ISSN 0031-3203. doi: https://doi.org/10.1016/0031-3203(91)90030-9. URL https://www.sciencedirect.com/science/article/pii/0031320391900309.

Y.-H. Kim, G. Bölöni, S. Borchert, H.-Y. Chun, and U. Achatz. Toward transient subgrid-scale gravity wave representation in atmospheric models. part ii: Wave intermittency simulated with convective sources. *Journal of the Atmospheric Sciences*, 78(4):1339 – 1357, 2021. doi: https://doi.org/10.1175/JAS-D-20-0066.1. URL https://journals.ametsoc.org/view/journals/atsc/78/4/JAS-D-20-0066.1.xml.

L. Krasauskas, B. Kaifler, S. Rhode, J. Ungermann, W. Woiwode, and P. Preusse. Oblique propagation of mountain waves to the upwind side of the andes observed by gloria and alima during the southtrac campaign. *Earth and Space Science Open Archive*, page 37, 2022. doi: 10.1002/essoar.10512325.1. URL https://doi.org/10.1002/essoar.10512325.1.

M. J. Lighthill. Waves in fluids. *Cambridge University Press*, page 504pp, 1978.

C. J. Marks and S. D. Eckermann. A three-dimensional nonhydrostatic ray-tracing model for gravity waves: Formulation and preliminary results for the middle atmosphere. *J. Atmos. Sci.*, 52(11):1959–1984, 1995. doi: 10.1175/1520-0469(1995)052¡1959:ATDNRT¿2.0.CO;2.

J. Muraschko, M. D. Fruman, U. Achatz, S. Hickel, and Y. Toledo. On the application of Wentzel-Kramer-Brillouin theory for the simulation of the weakly nonlinear dynamics of gravity waves (vol 141, pg 3446, 2015). *Quart. J. Roy. Meteorol. Soc.*, 141(693, B):3446, OCT 2015. ISSN 0035-9009. doi: 10.1002/qj.2719.

C. J. Nappo. *An Introduction to Atmospheric Gravity Waves.* Academic Press, second edition, 2012. ISBN 978-0-12-385223-6.

P. Preusse, M. Ern, P. Bechtold, S. D. Eckermann, S. Kalisch, Q. T. Trinh, and M. Riese. Characteristics of gravity waves resolved by ECMWF. *Atmos. Chem. Phys.*, 14(19):10483–10508, 2014. doi: 10.5194/acp-14-10483-2014.

S. Saha, K. Niranjan Kumar, S. Sharma, P. Kumar, and V. Joshi. Can quasi-periodic gravity waves influence the shape of ice crystals in cirrus clouds? *Geophysical Research Letters*, 47(11):e2020GL087909, 2020. doi: https://doi.org/10.1029/2020GL087909. URL https://agupubs.onlinelibrary.wiley.com/doi/abs/10.1029/2020GL087909. e2020GL087909 2020GL087909.

W. C. Skamarock. Evaluating mesoscale NWP models using kinetic energy spectra. *Mon. Weath. Rev.*, 132:3019–3032, 2004.

M. J. Taylor, E. H. Ryan, T. F. Tuan, and R. Edwards. Evidence of preferential directions for gravity wave propagation due to wind filtering in the middle atmosphere. *J. Geophys. Res.*, 98: 6047–6057, 1993. doi: 10.1029/92JA02604.

J. P. Thayer, M. Rapp, A. J. Gerrard, E. Gudmundsson, and T. J. Kane. Gravity-wave influences on arctic mesospheric clouds as determined by a rayleigh lidar at sondrestrom, greenland. *Journal of Geophysical Research: Atmospheres*, 108(D8), 2003. doi: https://doi.org/10.1029/2002JD002363. URL https://agupubs.onlinelibrary.wiley.com/doi/abs/10.1029/2002JD002363.

Q. T. Trinh, S. Kalisch, P. Preusse, H.-Y. Chun, S. D. Eckermann, M. Ern, and M. Riese. A comprehensive observational filter for satellite infrared limb sounding of gravity waves. *Atmos. Meas. Tech.*, 8:1491–1517, 2015. doi: 10.5194/amt-8-1491-2015.

S. Watanabe, K. Sato, and M. Takahashi. A general circulation model study of the orographic gravity waves over antarctica excited by katabatic winds. *J. Geophys. Res.*, 111, 2006. doi: 10.1029/2005JD006851.

J. Wei, G. Bölöni, and U. Achatz. Efficient modeling of the interaction of mesoscale gravity waves with unbalanced large-scale flows: Pseudomomentum-flux convergence versus direct approach. *Journal of the Atmospheric Sciences*, 76(9):2715 – 2738, 2019. doi: https://doi.org/10.1175/JAS-D-18-0337.1. URL https://journals.ametsoc.org/view/journals/atsc/76/9/jas-d-18-0337.1.xml.

P. D. Williams, P. L. Read, and T. W. N. Haine. Spontaneous generation and impact of inertia-gravity waves in a stratified, two-layer shear flow. *Geophysical Research Letters*, 30(24), 2003. doi: https://doi.org/10.1029/2003GL018498. URL https://agupubs.onlinelibrary.wiley.com/doi/abs/10.1029/2003GL018498.

D. L. Wu and S. D. Eckermann. Global gravity wave variances from Aura MLS: Characteristics and interpretation. *J. Atmos. Sci.*, 65(12):3695–3718, DEC 2008. ISSN 0022-4928. doi: 10.1175/2008JAS2489.1.

---

## Author Response (AR2)

**1 General Reply**

Thank you very much for your second review of our article. As with the previous review, your detailed comments helped us to write a better, more comprehensive paper.

We have considered all your comments and included many of your suggested changes in the revised manuscript.

Find below your original comments (in italics), the specific responses, and the changes made to the manuscript (in blue).

Sebastian Rhode, on behalf of all authors

**1.1 General comment**

*I would like to thank the authors of the manuscript "A mountain ridge model for quantifying oblique mountain wave propagation and distribution" for their thorough review, explanations, and friendly response. I think the manuscript has significantly improved but still requires some minor changes. Besides some minor comments, I figured out that the used blocking diagrams are violating the authors' assumptions and need replacement. A possible strategy is suggested below in the individual comments. With these suggested changes added to the manuscript, I will happily recommend it for publication. Please note that all line numbers refer to the manuscript version 2.*

Thank you again for your thorough review. We have addressed your specific comments below. In addition, we want to thank you for your constructive suggestion of a different kind of blocking diagram. We are happy to include this in our article as another analysis tool. The original blocking diagrams, as done in Taylor et al. [1993], are still kept in the manuscript to grant an investigation of the propagation conditions for non-orographic GWs.

The manuscript has received another overhaul, thanks to your suggestions.

**1.2 Specific comments**

1. *Response #14 and #15 are convincingly explained in the response but not reflected by a hint in the text. It would be nice if the authors could add corresponding comments.*

   We assume this comment refers to #13 and #15 since we added a new section in response to #14.

   Regarding the response to #13, we added another paragraph clarifying the difference in scale separation for the HIRDLS retrieval and the ray-tracing background generation: "Note that the scale-separation methodology of the ERA5 data used for background removal differs from the generation of the ray-tracing backgrounds. Since we are interested in the GW content of the measurements, we need to carefully remove the larger-scale dynamics, such as Rossby waves, from the background field. In the lower stratosphere, these can reach zonal wavenumbers as high as 6, but considerably higher in the troposphere, which is why the filter is designed with a linear decrease of cutoff wavenumber with altitude. The ray-tracing simulations are not as sensitive to small remnants of smaller-scale dynamics, and, therefore, the scale separation described in Strube et al. [2020] is used there."

   Regarding the response to #15, we added the following sentences to the manuscript: "The use of overlapping bins introduces spatially dependent auto-correlations to some extent, which leads to smearing out of the global distribution. The advantage is, however, an increase in statistics for each bin."

2. *L6 - location -¿ locations*

   This mistake is corrected in the revised version.

3. *L144 - the projection is still unclear - projecting onto an equidistant grid from 10° slices is not clear*

   We resample the longitude grid onto a 1' latitude equivalent at the center of the topography slice. In this way, the grid is equidistant at the center. The longitude spacing, however, is still increasing (decreasing) equator-ward (pole-ward).

The text in the manuscript was changed to:

"The longitudes of the topography slice are resampled onto a 1' latitude, or about 1.85 km, equivalent grid (at the meridional center), such that the resulting grid is equidistant in the center. The grid distance is, however, still increasing equator-ward (decreasing pole-ward). This resampling mitigates possible errors in the scale separation and fitting of ridge widths and lengths due to differently scaled dimensions at high latitudes."

4. *38 - the authors explain their usage of the chirp rate, maybe mentioning in the text a little clearer would be great*

   We edited the text to describe the effect of the approximation and which particular term is approximated by the chirp rate $c_m$ in a better way.

   The corresponding section has been reworked to: "The last term accounts for a linear approximation of the change in vertical wavelength along the vertical with chirp rate $c_{\mathrm{m}} = \frac{\Delta_{\mathrm{m}}}{\Delta_z}$ and $m(z) \approx m(z_0) + c_{\mathrm{m}} d_z + \mathcal{O}(d_z^2)$. The chirp rate is calculated as the finite difference derivative of $m$ for the closest time steps around target altitude $z$. The linear approximation of the dependence of the vertical wavelength on altitude increases the reconstruction performance significantly where it changes rapidly, e.g. below critical layers [e.g. Nappo, 2012]. In testing, we found that considering only the leading order, i.e., $m(z) \approx m(z_0)$, leads to inconsistent phase transitions between wave packets excited by the same mountain ridge at different times."

5. *Sec. 3.2 - It would be great to remind the reader of the lower boundary condition in the description of the raytracer.*

   We added a brief recapitulation of the lower boundary condition as required for GROGRAT at the beginning of section 3.2: "In this framework, the lower boundary condition for each individual GW is given by the location (longitude, latitude, and altitude) and launch time, the horizontal wave vector $(k, l)$, the initial amplitude, and the ground-based frequency."

6. *L239 - The amplitude correction was added to the appendix by the authors, it would be great if they would also mention some results and reason why they do not go forward with the calculated correction in the main body. In particular, the nice results raise the question of why this has not been taken into account throughout the whole study.*

   A few sentences to the general findings were added below the reference to App. D: "In our specific investigation, this correction leads to unchanged GW amplitudes (horizontally) close to the sources and enhanced amplitudes for GWs propagating far from the sources. Since horizontal deformation of the wave packet is caused by refraction, turning, and changing backgrounds, laterally far propagating GWs are especially prone to this effect."

   Our main reason for not including the correction in this study is to be consistent with previous studies focusing on ray-tracing in GROGRAT. The currently-used amplitude calculation has been used in a number of studies with validation to observations [Preusse et al., 2009, Krisch et al., 2017, Krasauskas et al., 2023]. Compared to this the approximated ray-tube method is less well validated and deserves a dedicated study to be properly introduced and tested. We here give first results in App. D to demonstrate the size of the effect.

   The following reasoning was added to the text: "Although the results in App. D are reasonable, for consistency with previous studies we are not taking the correction into account here, and all following simulations of this study are performed with the standard GROGRAT amplitude calculation.

   Although the results in App. D are reasonable for consistency with previous studies, we are not taking the correction into account here, and all following simulations of this study are performed with the standard GROGRAT amplitude calculation. The currently-used method of ignoring the last term in Eq. 6 has been used in a number of studies with validation to observations [Preusse et al., 2009, Krisch et al., 2017, Krasauskas et al., 2023]. Compared to this, the approximated ray-tube method is less well-validated and deserves a dedicated study to be properly introduced and tested. Therefore, we give only first results in App. D for a demonstration of the size of the effect."

7. *L303f - The authors provide a nice justification for the 3x3 choice in the response which is, however, not reflected in the manuscript. I would suggest adding a sentence that the expected errors given the subsampling are of the order of a couple of percents as quantified in the response.*

That is a good point. The following sentence was added to the manuscript after the description of the calculation: "We estimated the error of this approximation to be below $\sim 5\%$ for randomly oriented GWs of $100\,\mathrm{km}$ horizontal wavelength and decreasing for longer GWs."

8. *Sec. 4.1 - The new figures for the other regions are indeed a nice addition to the manuscript. It would be nice if the analysis in Appendix C was referenced in Sec. 4.1.*

Indeed, there was little advertising for this appendix. This was fixed by including of the following sentence:

"More detail on the reconstruction of the southern Andes topography for different scales and a similar analysis for the Himalaya and South Africa region, considered later on in this study, is given in App. C."

9. *L370ff - "The IFS data shows high activity of all scales above the oceans as well at this height, which indicates either wave sources other than orography (e.g. convection, jet fronts, and geostrophic (or spontaneous) adjustment, where an out-of-balance jet radiates excessive energy as inertia-gravity waves (e.g. Fritts and Alexander, 2003; Williams et al., 2003; de la Camara and Lott, 2015)) or completely different tropospheric processes, that our scale separation anomalously picks up. However, the MWM also shows a large-scale pattern to the east of the continent, which might indicate that orographic GWs of large-scale (and therefore higher horizontal group velocity) might also add to the patterns seen in the IFS."*

*- Here the line of argument needs to be disentangled. The point of comparison to the IFS is the validation of the MWM. However, the authors rather interpret the IFS and suggest short-comings to the scale separation in postprocessing the IFS data instead. This interpretation would be premature for a non-validated dataset. I suggest the authors reframe the argument into the interpretation of the differences in terms of the MWM rather than evaluating the IFS based on the MWM.*

This section was framed the wrong way around and was therefore changed corresponding to your comment to: "Nevertheless, the MWM data exhibits a large-scale pattern to the east of the continent above the Pacific Ocean, indicating that MWs of comparatively large scales (and thus high horizontal group velocity) are strongly propagating below the tropopause. And indeed, we see similar structures in the IFS model data. Note, however, that the IFS models GWs of all sources, and therefore, the seen features could be (partly) due to convection, jet fronts, and geostrophic (or spontaneous) adjustment (where an out-of-balance jet radiates excessive energy as inertia-gravity waves [e.g. Fritts and Alexander, 2003, Williams et al., 2003, de la Camara and Lott, 2015]) or even other tropospheric processes, that are not filtered out well in our scale separation."

10. *LL450-454 - The authors describe the patterns found in the HIRDLS dataset, but do not mention the latter. I suggest the authors clarify which dataset the described observation is associated with.*

We tried to use the phrasing "observation" only for the HIRDLS measurements, but indeed this is not very clear in general. Thus the text was changed according to your comment:

"The strongest pattern in the HIRDLS observations is found above..."

11. *L496, Fig. 6 - blocking diagrams Surprisingly I have not seen some inconsistencies concerning the blocking diagrams in the first round of reviews. I think, however, that the blocking diagrams are not applicable in the here presented form. First of all, equation 12 is generally singular for mountain waves. Dividing by $\omega_{\mathrm{gb}}$ is only permissive where it is non-zero and additionally contradicts the interpretation of the blocking diagram where it is assumed to be approximately zero (c.f. lines 506 and 515). Moreover, the rotation was neglected by assuming a minimum intrinsic frequency equal to zero. Finally, I am under the impression*

*that the authors misinterpret the figures of Taylor et al. 1993. Their blocking diagrams are plotted in terms of the background wind (U, V) (admittedly their axes lack labels) rather than phase speeds. Moreover, description in terms of ground base phase speeds (as done here) for orographic waves would be trivial as they are always approximately zero. The present interpretation of Fig. 9 thus directly violates the quasi-stationary nature of mountain waves. I thus suggest the following procedure instead:# Rewriting equation 11 - under the assumption of an approximately zero ground-based frequency and a quasi-steady wind - one may find that*

$$f < \omega_{intr} = -|\vec{U}||\vec{k}_{hor}|\cos(\alpha), \tag{1}$$

*where $\alpha$ is the angle between the horizontal wind and wave vectors. One may thus find all regions where the above relation is violated (corresponding to the critical filtering) as a function of the zonal and meridional wavenumbers within the target area and altitude range. The probability then depends on the observed wind directions (relative to the wave vector) as well as the wind amplitude. Visualizations similar to Fig. 9 but depending on the wavenumbers (k, l) instead would then give a probability of critical layer filtering given the initial wavenumber pair.*

Yes, it is correct that these blocking diagrams give only a hint at the filtering of mountain waves (with $\omega_{gb} \approx 0$). However, they give a good impression of regions within the phase speed that are critical level filtered for GWs of other origin (or even MWs far from their source, where the transient background field has shifted gb away from 0).

The diagrams in Taylor et al. [1993], however, do show phase speed spectrum and the restricted regions due to the external wind. Although the restricted areas depend on the background wind speeds, the diagram axis are given as the phase speed in zonal and meridional direction. Indeed the missing description and axes labeling are misleading, the corresponding text in Taylor et al. [1993], however, clarifies that phase speeds are considered.

Our main aim in using the blocking diagrams is to look at the restrictions for non-orographic waves and thereby explain the different change in GWMF in the HIRDLS observations and the MWM predictions, since HIRDLS observes much more than we model within the MWM. In the revised version, we tried to make the focus on the non-orographic GWs more clear.

Indeed, the Coriolis frequency has been left out for this consideration similar to the work of Taylor et al. [1993]. The framework of blocking diagrams gives nevertheless a good first look at the restrictions in phase speed space. To clarify that $f$ is neglected, we added the following sentence to the manuscript: "In addition, the Coriolis parameter is neglected within this consideration, which would restrict the intrinsic frequency even more ($|f| < \omega_{intr}$) and hence leads to a stronger restriction of phase speeds."

[revised manuscript text omitted]

12. *L519 - blocking -> critical layer filtering*

   This was changed accordingly in the revised version.

13. *L726 - "Using the ray-tracer GROGRAT [...]" -¿ Using a modified version of the ray-tracer GROGRAT...*

    This was changed accordingly in the revised version.